# DIYHEALTH SUITE: DATASET, MODEL, AND BENCHMARK FOR HEALTH MANAGEMENT AT HOME

## ABSTRACT

Generative AI is reshaping healthcare by enhancing multimodal data interpretation, clinical insight generation, and personalized decision support. However, existing advances remain tightly coupled with hospital-grade devices, restricting accessibility and use for anytime, anywhere health management in non-clinical settings. With the proliferation of wearables, mobile sensors, and telemedicine, healthcare is shifting toward the home, giving rise to the emerging field of **Diagnosis-It-Yourself (DIY) at home**, *i.e.*, home care. Despite this promise, several distinctive challenges remain: (i) home-collected data are heterogeneous, exacerbated by the absence of standardized large-scale datasets; (ii) models require adaptation to highly variable task demands and dynamically evolving individual conditions; (iii) the broad spectrum of home care tasks lacks a unified benchmark for systematic evaluation. In this paper, we present **DIYHealth Suite**, a comprehensive framework designed to address these challenges through a tailored dataset, model, and benchmark. We first curate **DIYHealth-900K**, a large-scale multimodal dataset capturing diverse real-world home care scenarios. Building on this, we propose **DIYHealthGPT**, an adaptive foundation model for home-based health management, powered by the novel Hybrid Hyper Low-Rank Adaptation technique, which integrates expert mixtures with hypernetwork-driven modulation to balance cross-task generalization and instance-level personalization. Finally, we establish **DIYHealthBench**, the first benchmark to evaluate foundation models on home care tasks. Extensive experiments demonstrate that DIYHealthGPT delivers state-of-the-art performance over both general-purpose and medical-specific baselines on 11 home care tasks in both open-QA and closed-QA settings, laying the groundwork for the next generation of AI-driven, personalized, and scalable health management at home.

## 1 INTRODUCTION

Generative AI has progressed at an unprecedented rate, evolving from large language models (LLMs) (Brown et al., 2020; OpenAI, 2023) to multimodal architectures (Radford et al., 2021), driving broad impact across diverse domains. Within healthcare, these advances have enabled the interpretation of complex clinical data, the integration of heterogeneous modalities, and the provision of decision support, opening new opportunities to enhance diagnosis, treatment, and patient care (Wang et al., 2015; Lin et al., 2025; Qiu et al., 2024).

Recent studies have begun to investigate medical foundation models that leverage large-scale clinical data to improve tasks such as radiology report generation (Thawkar et al., 2023), medical image interpretation (Rajpurkar & Lungren, 2023), and clinical question answering (Li et al., 2025; Xie et al., 2025). These models achieve strong performance by capturing domain-specific knowledge and adapting to a broad range of medical tasks, representing a significant step toward AI-assisted healthcare. Nevertheless, current efforts remain predominantly clinical-centric: they are trained on hospital-grade data, tailored for professional use, and optimized for environments where high-quality imaging, electronic health records (EHRs), and expert annotations are readily available.

Although foundation models have achieved considerable success in clinical contexts, their application to the home care setting has remained largely **unexplored** to date. The home care setting holds significant promise for collecting multimodal health data and enabling convenient inference beyond clinical environments, facilitated by the widespread adoption of smartphones, wearables, and home

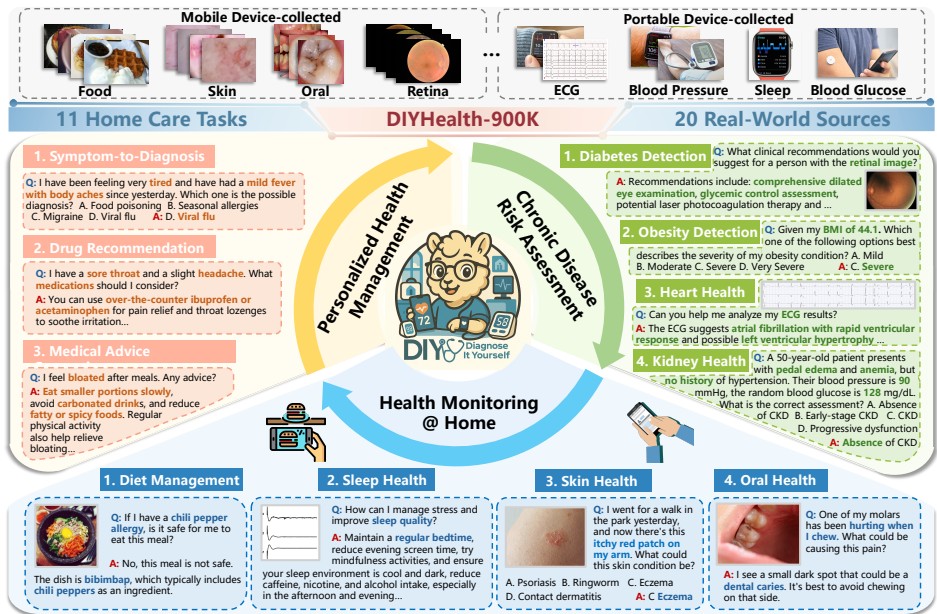

Figure 1: Overview of DIYHealth Suite, integrating DIYHealth-900K, DIYHealthGPT, and DIY-HealthBench across 11 home care tasks towards health management at home.

sensors (Zaidan et al., 2018; Kruzan et al., 2023). Despite this opportunity, home care introduces distinct challenges that differ fundamentally from those encountered in clinical settings. To begin with, data collected at home often originates from consumer-grade devices and self-reported inputs, resulting in heterogeneous and lower-quality signals; the absence of standardized large-scale datasets further constrains the systematic development of foundation models in this context. Further, whereas population-level models may suffice in hospital settings, home care demands adaptation to highly variable personal health baselines and evolving individual conditions. Finally, home care spans a wide spectrum of tasks, from personalized health management to daily health monitoring, yet currently lacks a unified benchmark for evaluating model performance across such diverse applications.

To systematically unlock the potential for health management at home while addressing the aforementioned key challenges of data accessibility, personalization, and task diversity, we introduce **DIYHealth Suite**, a comprehensive ecosystem that integrates a large-scale multimodal dataset curated for home care, an adaptive foundation model designed to accommodate individual variability, and a unified benchmark spanning diverse tasks in everyday health management and monitoring.

Within the ecosystem, we propose three core components. We first construct **DIYHealth-900K**, a multimodal dataset curated via an LLM-powered data engine to aggregate heterogeneous inputs from home environments under rigorous quality control. To address variability across individuals, we propose **DIYHealthGPT**, an adaptive foundation model that employs a novel Parameter-Efficient Fine-Tuning (PEFT) technique, Hybrid Hyper Low-Rank Adaptation ($H^2$LoRA), which combines low-rank expert mixtures for efficient cross-task knowledge sharing with hypernetwork-driven adaptation for instance-aware personalization. Finally, to enable systematic evaluation, we establish **DIYHealthBench**, a unified benchmark guided by a multi-dimensional evaluation protocol spanning both open-QA and closed-QA, thereby capturing the dual requirements of home care: adaptive dialogue for personalized advice and structured reasoning for decision support. Collectively, these components lay the groundwork for accessible, intelligent health management beyond clinical settings, advancing inclusive AI for everyday well-being. Our key contributions are summarized below:

- We curate DIYHealth-900K, a comprehensive multimodal dataset collected from everyday devices to reflect the complexity and variability of real-world home care scenarios.
- We propose DIYHealthGPT, an adaptive foundation model for home-based health management, powered by the innovative $H^2$LoRA mechanism that enables personalized representations while maintaining robust generalization.
- We introduce DIYHealthBench, the first unified benchmark for evaluating foundation models in non-clinical settings, spanning tasks from daily health monitoring to chronic disease risk assessment and personalized health management, reflecting the diverse needs of home care.

- Extensive experiments on DIYHealthBench demonstrate that DIYHealthGPT consistently outperforms both state-of-the-art generalist and medical-specific baselines across diverse home care tasks, affirming its effectiveness in facilitating personalized health management at home.

## 2 RELATED WORK

**LLMs for General Reasoning and Dialogue.** LLMs have advanced natural language understanding and generation, enabling general-purpose reasoning and coherent dialogue (Chowdhery et al., 2023; Touvron et al., 2023). Models such as GPT-3 (Brown et al., 2020) and GPT-4 (OpenAI, 2023), trained on large-scale corpora, exhibit remarkable zero-shot and few-shot generalization across tasks. These capabilities arise from their scale—with billions of parameters—and sophisticated pre-training and alignment techniques (Wei et al., 2022; Fedus et al., 2022). However, their use in specialized domains such as healthcare is limited by the lack of domain knowledge and grounding in real-world physiological data, particularly in low-resource and home-based settings.

**Medical Foundation Models.** To bridge the domain gap between general-purpose LLMs and medical applications, recent efforts have introduced medical foundation models, including text-only Med-LLMs and multimodal Med-LVLMs. Text-based models such as BiomedGPT (Luo et al., 2024b) and HuatuoGPT (Chen et al., 2024a) achieve strong performance on clinical question answering (QA) benchmarks by leveraging curated biomedical corpora and synthetic datasets. Med-LVLMs extend this paradigm to multimodal reasoning: LLaVA-Med (Li et al., 2023a), Med-Flamingo (Moor et al., 2023), and MedVLM-R1 (Pan et al., 2025) align visual encoders with LLMs for diagnostic tasks. More recently, HealthGPT (Lin et al., 2025) supports multimodal comprehension and generation on diverse medical tasks, while EyecareGPT (Li et al., 2025) devises specialized mechanisms for ophthalmic analysis. These models demonstrate potential in image captioning, visual question answering (VQA), and differential diagnosis (see Appendix B for a comprehensive review).

Despite these advances, existing Med-LVLMs rely primarily on data from professional medical devices (e.g., radiographic images, pathology slides), limiting their applicability in everyday contexts. Their lack of portability, weak adaptability to informal data, and insufficient support for personalized inference further constrain their use in home-based health management. To overcome these challenges, we propose DIYHealth Suite, an innovative solution with three core components—DIYHealth-900K, DIYHealthBench, and DIYHealthGPT—designed for effective operation on consumer-grade devices such as smartphones, wearables, and smart home sensors in home care settings.

## 3 DIYHEALTH-900K

### 3.1 TASK LANDSCAPE AND DATA CURATION IN HOME CARE

To reflect the diversity of real-world home care scenarios, we categorize the tasks in the DIYHealth-900K dataset into three groups: (i) *Personalized Health Management*, which includes core tasks such as symptom-based diagnosis, drug recommendation, and medical advice generation; (ii) *Chronic Disease Risk Assessment*, which targets conditions such as diabetes, obesity, cardiovascular, and kidney health through self-reported symptoms and home-acquired signals; and (iii) *Daily Health Monitoring*, which encompasses dietary intake, sleep, skin, and oral assessments. All tasks are curated or adapted to incorporate multimodal input, real-world variability, and nonclinical supervision, ensuring their relevance to home care environments.

To support these tasks, we construct DIYHealth-900K by collecting and integrating data from 20 publicly available data sources, such as Kaggle, PhysioNet, and Figshare. Each dataset focuses on a specific medical task and contains data ranging from patient demographics, vital signs, and laboratory test results to medical conversations, questionnaires, and other clinically relevant records. Each task is aligned with one or more source datasets that have been adapted to the home care setting. The selection prioritizes modalities commonly observed in practice, including natural language symptom descriptions, wearable-derived signals (for heart and sleep health), and mobile-captured images (for dietary intake, skin conditions, and oral health). Further details regarding task design and dataset construction are provided in Appendix C.

## 3.2 DIYHEALTH DATA ENGINE

To construct a reliable data resource for home care, we develop an LLM-powered data engine with human-in-the-loop verification, consisting of four components: (i) *Linguistic and Signal Normalizer*: We standardize heterogeneous raw data by addressing common issues such as medical text abbreviations and preprocessing of physiological signals (e.g., ECG). (ii) *Prompt and Template Library*: We design a principled library of prompts and templates to guide QA pair generation. The library spans both text-only QA and VQA tasks, supports open-QA (free-form answers) and closed-

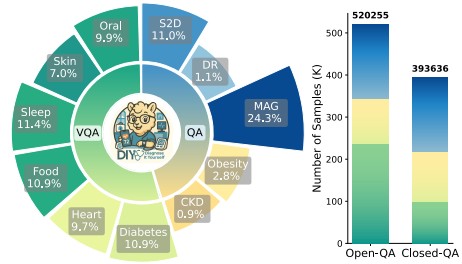

Figure 2: Data statistics of DIYHealth-900K

QA (multiple-choice), and incorporates diverse user perspectives (first-person and third-person), simulating realistic home care interactions. (iii) *Semantic QA Synthesizer*: We employ Claude 3 Haiku to automatically synthesize large-scale QA pairs, guided by the prompt schema to maintain semantic consistency across tasks. (iv) *Human-in-the-Loop Validator*: To guarantee quality, we randomly sample 10% of the automatically generated QA pairs for inspection by human reviewers. Medical professionals focus on semantic validity, medical consistency, and format standardization. Each entry undergoes two rounds of independent review, providing fine-grained data quality control and ensuring reliability. Data statistics are summarized in Figure 2, with task abbreviations defined in Table 6.

# 4 DIYHEALTHGPT

## 4.1 MULTIMODAL PERCEPTION UNIFICATION

Home care scenarios naturally involve heterogeneous data sources, including food images, skin images, and textual symptom descriptions, among others. To enable consistent reasoning across such diverse modalities, DIYHealthGPT designs a multimodal perception unification mechanism that projects both visual and textual inputs into a shared semantic embedding space.

**Visual Encoding.** Given an input image $\mathcal{I} \in \mathbb{R}^{H \times W \times 3}$, we employ a pretrained vision encoder $\mathcal{E}_v(\cdot)$ to extract a sequence of patch-level representations:

$$\mathcal{V} = \mathcal{E}_v(\mathcal{I}) \in \mathbb{R}^{L_v \times d_v} \tag{1}$$

where $L_v$ denotes the number of visual tokens and $d_v$ represents the visual embedding dimension.

**Textual Encoding.** For a textual symptom description $\mathcal{T} = \{t_1, \ldots, t_{L_t}\}$, where $t_i \in \mathcal{V}_{txt}$, and $\mathcal{V}_{txt}$ represents the vocabulary of the backbone language model, we use a pretrained tokenizer and embedding layer $\mathcal{E}_t(\cdot)$. This yields:

$$\mathcal{U} = \mathcal{E}_t(\mathcal{T}) \in \mathbb{R}^{L_t \times d} \tag{2}$$

where $L_t$ is the text length and $d$ is the language embedding dimension.

**Modality Projection and Unification.** To align heterogeneous modalities within a shared semantic space, we introduce a learnable projection function $\mathcal{P}_v : \mathbb{R}^{d_v} \to \mathbb{R}^d$ that maps visual embeddings into the language embedding space. The unified multimodal representation is then formulated as:

$$\mathcal{Z} = \left[\mathcal{P}_v(\mathcal{V}); \mathcal{U}\right] \in \mathbb{R}^{(L_v + L_t) \times d} \tag{3}$$

where $[\,;\,]$ denotes token concatenation and $d$ is the shared embedding dimension in DIYHealthGPT.

This process defines a heterogeneous-to-homogeneous interface: $\Phi : (\mathcal{I}, \mathcal{T}) \mapsto \mathcal{Z}$, which ensures that both visual and textual home care signals are embedded within a coherent semantic space. The resulting unified representation $\mathcal{Z}$ is subsequently provided as input to the backbone language model $\mathcal{M}_{LLM}$ for downstream adaptation and generation.

## 4.2 HYBRID HYPER LOW-RANK ADAPTATION

While LLMs provide strong general reasoning capabilities, they generally lack the domain- and task-specific specialization required in home care scenarios. PEFT techniques (Ding et al., 2023),

Figure 3: Model architecture of DIYHealthGPT, where H²LoRA integrates Low-Rank Expert Mixture with Hyper LoRA Adaptation to balance task generalization and instance-level personalization.

particularly Low-Rank Adaptation (LoRA) (Hu et al., 2022), offer a scalable approach by introducing trainable low-rank adapters into otherwise frozen pretrained weights. However, conventional LoRA strategies face inherent limitations: (i) allocating a distinct adapter for each task restricts cross-task knowledge sharing, whereas (ii) relying on a fully shared adapter overlooks the fine-grained task-specific distinctions. To overcome these limitations, we propose Hybrid Hyper Low-Rank Adaptation (H²LoRA), a novel mechanism designed to enable efficient parameter sharing while retaining adaptive task-specific specialization.

Given the unified multimodal embedding $\mathcal{Z}$, the backbone language model $\mathcal{M}_{\text{LLM}}$ produces task-aware outputs $\mathcal{O}$ by combining frozen parameters $\Theta$ with task-adaptive parameters $\Theta_{H^2L}$ introduced through our H²LoRA mechanism:

$$\mathcal{O}_{H^2LoRA} = \mathcal{M}_{\text{LLM}}(\mathcal{Z}; \Theta, \Theta_{H^2L}), \quad \Theta_{H^2L} = \{\mathcal{A}, \mathcal{B}, \mathcal{R}\} \tag{4}$$

where $\mathcal{A} = \{\mathbf{A}^t, \Delta\mathbf{A}^t\}_{t=1}^N$ and $\mathcal{B} = \{\mathbf{B}^t, \Delta\mathbf{B}^t\}_{t=1}^N$ denote the task-specific low-rank parameters associated with the home care tasks $T = \{1, \cdots, N\}$. The routing parameters $\mathcal{R}$ further integrate task-level outputs, as detailed in Section 4.3. To achieve cross-task knowledge sharing and task-level specialization simultaneously, the task-specific pair $(\mathbf{A}^t, \mathbf{B}^t)$ and $(\Delta\mathbf{A}^t, \Delta\mathbf{B}^t)$, $t \in T$ are internally structured through two complementary mechanisms: *Low-Rank Expert Mixture* and *Hyper LoRA Adaptation*.

**Low-Rank Expert Mixture.** At the task level, consider a weight matrix $\Theta \in \mathbb{R}^{d_{\text{out}} \times d_{\text{in}}}$ in the backbone model $\mathcal{M}_{\text{LLM}}$. H²LoRA augments this parameter with a shared low-rank projection $\mathbf{A}^t \in \mathbb{R}^{d_{\text{out}} \times r}$ and a set of $K$ expert matrices $\{\mathbf{B}_1^t, \ldots, \mathbf{B}_K^t\}$, where $r \ll \min(d_{\text{out}}, d_{\text{in}})$. Inspired by the Mixture of Experts (MoE) paradigm, a task embedding $\mathcal{Z} \in \mathbb{R}^d$ is processed by a routing layer to generate expert weights $\mathcal{W}^t \in \mathbb{R}^K$, typically normalized with a softmax function. To interface with the low-rank structure, these weights are expanded along the rank dimension as

$$\hat{\mathcal{W}}^t = K\mathcal{W}^t/r \otimes \mathbf{1}_r \tag{5}$$

where $\otimes$ denotes the replication operation. The task-adaptive projection is then obtained as a convex combination of experts:

$$\mathbf{B}^t = \hat{\mathcal{W}}^t \odot \text{Concat}(\mathbf{B}_1^t, \ldots, \mathbf{B}_K^t) \tag{6}$$

where $\odot$ denotes element-wise multiplication. This design employs $\mathbf{A}^t$ as a shared anchor across $K$ expert matrices $\mathbf{B}_k^t$, encouraging subspace alignment. Meanwhile, the MoE-driven mixture of $\mathbf{B}^t$ matrices provides the flexibility required for task-level specialization. During the subsequent multi-task training phase (Sec. 4.4 Stage 4), this structure further facilitates cross-task knowledge sharing. By integrating LoRA with expert routing, the low-rank expert mixture strikes a principled balance between efficiency and expressivity, providing greater adaptation capacity than either fully shared or fully task-isolated alternatives.

**Hyper LoRA Adaptation.** H²LoRA further incorporates hyper LoRA adaptation to capture the task-aware, instance-specific variations that frequently arise in home care scenarios, such as personalization

across patients. In this design, both the shared projection $\mathbf{A}^t$ and the expert matrices $\mathbf{B}_k^t$ are equipped with instance-dependent offsets, dynamically generated by dedicated hypernetworks:

$$\Delta\mathbf{A}^t = \mathcal{H}_A(\mathcal{Z}), \quad \Delta\mathbf{B}_k^t = \mathcal{H}_B(\mathcal{Z}), \quad k = 1, \ldots, K \tag{7}$$

The offset for MoE-driven mixture $\Delta\mathbf{B}^t$ is then computed via Eq.(6). Thus, the task-level output of H²LoRA is $\mathcal{O}_{H^2LoRA}^t = \mathcal{Z}\mathbf{A}^t\mathbf{B}^t + \mathcal{Z}\Delta\mathbf{A}^t\Delta\mathbf{B}^t$. By conditioning the hypernetworks on $\mathcal{Z}$, this formulation renders the adaptation instance-aware, enabling flexible personalization that extends beyond coarse task-level specialization and better reflects the heterogeneity of home care contexts.

### 4.3 INTER-TASK KNOWLEDGE FUSION

While H²LoRA equips each task with a specialized adapter, healthcare tasks in home care scenarios are generally strongly correlated. For instance, dietary patterns directly affect the risks of diabetes and obesity, while early symptom recognition provides essential context for subsequent drug recommendation and personalized advice generation (Hu, 2011; Mozaffarian, 2016). To exploit such inter-dependencies, we introduce an inter-task knowledge fusion mechanism, where a global soft-MoE router $\mathcal{R}$ dynamically integrates the outputs from the $N$ task-specific H²LoRA blocks. Specifically, the router $\mathcal{R}$ assigns mixture weights $\beta$ conditioned on the shared embedding $\mathcal{Z}$:

$$\beta = (\beta^1, \ldots, \beta^N) = \mathcal{R}(\mathcal{Z}), \quad \beta^t \geq 0, \sum_{t=1}^{N} \beta^t = 1 \tag{8}$$

The overall H²LoRA update is then expressed as: $\mathcal{O}_{H^2LoRA} = \mathcal{Z}\Theta + \sum_{t=1}^{N} \beta^t \mathcal{O}_{H^2LoRA}^t$. This design achieves a principled balance among efficiency, specialization, and personalization. Within each task, the shared projection $\mathbf{A}^t$ promotes common representation learning, the expert matrices $\mathbf{B}_k^t$ capture task-specific signals, and hypernetwork-driven offsets provide instance-aware modulation for fine-grained variation. Beyond task-level adaptation, the inter-task fusion layer treats each H²LoRA block as an expert and leverages the global soft router $\mathcal{R}$ to integrate them contextually, thereby exploiting correlations across healthcare tasks. Prior approaches account for only partial aspects: MoELoRA (Luo et al., 2024a) emphasizes feature-level expert diversity, whereas HyperLoRA (Lv et al., 2024) directly generates task-specific LoRA weights via a hypernetwork but suffers from optimization instability. In contrast, H²LoRA broadens the design space by integrating expert mixtures, a residual hypernetwork-driven modulation, and cross-task soft fusion. As shown in Figure 3, this integration enables coherent, contextually grounded, and personalized health responses across diverse scenarios while adhering to strict parameter budgets.

### 4.4 TRAINING PIPELINE

To optimize DIYHealthGPT for diverse home care tasks, we design a four-stage training pipeline that progressively aligns modalities, adapts the backbone to the medical domain, specializes task-level experts, and integrates them into a unified framework.

**Stage 1: Cross-Modal Alignment.** We begin by training the projector $\mathcal{P}_v$, which maps visual embeddings into the shared semantic space, using LLaVA-558k (Liu et al., 2024) and PubMedVision (Chen et al., 2024a). This stage establishes robust alignment between visual and language representations.

**Stage 2: Medical Domain Adaptation.** We then perform supervised fine-tuning by jointly training the projector $\mathcal{P}_v$ and the backbone $\mathcal{M}_{\text{LLM}}(\cdot; \Theta)$ on our curated DIYHealth-900K dataset covering 11 tasks. To balance efficiency and coverage, we sample 10% of training data per task, allowing the backbone to acquire medical knowledge while reducing overfitting risk. The sampling is performed once prior to training to construct a fixed subset, which remains unchanged throughout the training process.

**Stage 3: Task-Specific Expert Training.** For each task, we train a dedicated H²LoRA block, parameterized by $(\mathbf{A}^t, \Delta\mathbf{A}^t, \mathbf{B}^t, \Delta\mathbf{B}^t)$, on its respective subset of DIYHealth-900K, while keeping all other parameters fixed. After individual training, we introduce a hard-MoE layer that activates a single expert at a time, and jointly optimize all experts on the full multi-task training set of DIYHealth-900K, ensuring integration under shared supervision.

**Stage 4: Cross-Task Knowledge Transfer.** Finally, we fine-tune the task-specific H²LoRA expert blocks while replacing the hard-MoE layer with a global soft-MoE router $\mathcal{R}$ on the multi-task training set of DIYHealth-900K. This stage facilitates cross-task knowledge transfer, enabling the model to exploit inter-task correlations while maintaining parameter efficiency.

## 5 DIYHEALTHBENCH

**Standardized Benchmark for Home Care AI.** To enable a rigorous and fair evaluation of foundation models in non-clinical settings, we introduce DIYHealthBench, a benchmark dedicated to 11 real-world home care tasks defined in DIYHealth-900K. These tasks span three major categories: personalized health management, chronic disease risk assessment, and daily health monitoring. The tasks are formulated in both open-QA and closed-QA formats, reflecting the dual requirements of home care: naturalistic dialogue for adaptive advice and structured reasoning for actionable decision support. DIYHealthBench is derived from the designated test set of DIYHealth-900K and comprises 12,167 examples in total. For each task, we randomly sample 1% of the data for evaluation. For small datasets, a minimum of 1,000 samples is used to ensure statistical reliability and sufficient task coverage. To ensure representativeness, the samples are balanced across task types, input modalities, and disease categories, providing a standardized basis for evaluation in home-based health management. By establishing the first benchmark tailored for home care AI, DIYHealthBench bridges the gap left by existing hospital-centric evaluations and lays a foundation for developing and assessing personalized health assistance beyond clinical settings.

**Multi-Dimensional Evaluation Protocol.** We evaluate both general-domain LVLMs and medical-specific LVLMs as baselines, establishing reference performance through a multi-dimensional evaluation suite. For closed-QA, we adopt accuracy (ACC) as the primary measure of diagnostic precision and complement it with Matthews Correlation Coefficient (MCC). For open-QA, we employ two complementary groups of metrics. *Content-level* metrics, including F1-RadGraph (F1-Rad) (Yu et al., 2023) and F1-BioBERT (F1-Bio) (Lee et al., 2020), quantify the semantic and biomedical fidelity of generated responses. *Language-level* metrics, namely BLEU (Papineni et al., 2002) and ROUGE-L (RL) (Lin & Hovy, 2003), assess surface-level fluency and textual overlap with ground-truth answers, providing a balanced assessment for medical applications. Please refer to Appendix D.5 for details of evaluation metrics.

## 6 EXPERIMENTS

### 6.1 EXPERIMENTAL SETUP

**Data Details.** Following the DIYHealthBench protocol, we evaluate DIYHealthGPT on DIYHealth-900K, a multimodal QA dataset of approximately 900K samples spanning 11 home care tasks, split into training and test sets at a 99:1 ratio with strict user-level separation to prevent data leakage.

**Baselines.** We evaluate DIYHealthGPT against a broad set of baselines, including state-of-the-art generalist models (e.g., LLaVA-1.5 (Liu et al., 2023), InstructBLIP (Dai et al., 2023), Llama 3.2 (Dubey et al., 2024), Yi-VL (Young et al., 2024), InternVL3 (Zhu et al., 2025), Qwen2.5-VL (Bai et al., 2025), Gemma 3 (Team et al., 2025), Claude 3 Haiku (Anthropic, 2024) and GPT-4o Mini (Achiam et al., 2023)) and medical-specific models (e.g., LLaVA-Med v1.5 (Li et al., 2023a), Med-Flamingo (Moor et al., 2023), HuatuoGPT-Vision (Chen et al., 2024b), MedGemma (Sellergren et al., 2025), HealthGPT (Lin et al., 2025), Med-R1 (Lai et al., 2025), Lingshu (Xu et al., 2025), and MedVLM-R1 (Pan et al., 2025)). Experiments are conducted on 11 home care tasks under open-QA settings, and 10 tasks are included for closed-QA, as the drug recommendation task is inherently multi-label and thus better suited to open-QA format. Please refer to Appendices C and D for details on the construction of DIYHealth-900K and the implementation of DIYHealthGPT.

### 6.2 MAIN RESULTS

The experimental results in closed- and open-QA settings are in Tables 1 and 2, with supplementary results of F1-Rad and BLEU-1 provided in Appendix E.1. From these results, we derive several key observations. **(i) State-of-the-art performance.** Despite its compact model size, DIYHealthGPT delivers the best performance across all tasks, with an average improvement of 22.7% in ACC under closed-QA and 16.7% in F1-Bio under open-QA, substantially surpassing both general and medical domain models. These results highlight DIYHealthGPT's strong capability in producing accurate choices and generating faithful, coherent responses. **(ii) Narrow margin on MAG under closed-QA.** DIYHealthGPT's gain over the runner-up (i.e., InstructBLIP) is marginal on MAG task, as MAG is inherently general and depends more on wide medical knowledge than on specialized

cues. Thus, models pretrained on large corpora transfer well, raising baselines and narrowing gaps. Yet, DIYHealthGPT's advantage becomes pronounced on the open-QA setting of MAG, where broad knowledge alone is insufficient and robust medical reasoning is required. **(iii) Limitations of existing Med-LVLMs.** Current models, though pretrained on extensive medical corpora, largely overlook home care scenarios, where learning and prediction depend exclusively on data available outside clinical settings. This omission results in suboptimal performance on home care tasks. In contrast, we propose DIYHealth Suite, which integrates DIYHealth-900K, DIYHealthGPT, and DIYHealthBench, offering a unified resource and a strong baseline for advancing health management at home.

Table 1: Comparison of DIYHealthGPT with baselines under *closed-QA* settings in DIYHealthBench.

| Model | S2D | | MAG | | Diabetes | | Obesity | | Heart | | CKD | | Food | | Sleep | | Skin | | Oral | | Avg | |
|---|---|---|---|---|---|---|---|---|---|---|---|---|---|---|---|---|---|---|---|---|---|---|
| | ACC | MCC | ACC | MCC | ACC | MCC | ACC | MCC | ACC | MCC | ACC | MCC | ACC | MCC | ACC | MCC | ACC | MCC | ACC | MCC | ACC | MCC |
| *General Domain Models* | | | | | | | | | | | | | | | | | | | | | | |
| LLaVA-1.5-7B | 52.36 | 39.22 | 36.74 | 19.13 | 58.90 | 39.72 | 61.90 | 47.97 | 48.94 | 32.79 | 81.44 | 75.27 | 80.69 | 74.29 | 21.35 | -0.56 | 40.21 | 16.68 | 52.35 | 32.29 | 52.87 | 35.39 |
| InstructBLIP-7B | 25.67 | 4.44 | 4.87 | 0.89 | 5.52 | 5.38 | 15.40 | 6.52 | 0.18 | 2.60 | 5.15 | 4.01 | 42.92 | 35.26 | 9.42 | 0.65 | 14.17 | 2.89 | 21.37 | 5.43 | 14.35 | 8.54 |
| Llama 3.2-11B | 60.99 | 49.89 | 46.48 | 33.66 | 64.23 | 47.03 | 54.12 | 39.58 | 25.35 | 6.96 | 56.91 | 48.42 | 74.03 | 66.96 | 20.96 | 2.53 | 38.54 | 17.55 | 41.03 | 25.97 | 47.42 | 33.12 |
| Yi-VL-6B | 75.56 | 67.39 | 48.49 | 33.49 | 75.27 | 59.01 | 65.47 | 52.78 | 48.59 | 31.71 | 73.20 | 64.38 | 76.39 | 68.63 | 25.38 | 7.65 | 45.83 | 19.12 | 69.23 | 51.75 | 59.46 | 46.08 |
| InternVL3-8B | 85.42 | 80.54 | 59.73 | 48.16 | 88.61 | 79.94 | 70.92 | 59.61 | 37.15 | 16.29 | 72.81 | 91.88 | 85.41 | 80.53 | 21.54 | 2.10 | 74.38 | 51.45 | 91.88 | 85.44 | 68.79 | 59.59 |
| Qwen2.5-VL-7B | 46.82 | 29.56 | 57.89 | 46.82 | 70.28 | 52.03 | 58.01 | 43.28 | 33.80 | 12.69 | 64.74 | 55.10 | 73.18 | 64.58 | 20.38 | 3.78 | 54.79 | 27.87 | 47.44 | 26.94 | 51.77 | 35.50 |
| Gemma 3-4B | 66.94 | 56.32 | 46.14 | 30.47 | 67.08 | 47.6 | 65.94 | 53.18 | 34.68 | 12.56 | 72.81 | 66.45 | 86.05 | 81.41 | 15.96 | -4.25 | 48.13 | 27.06 | 66.45 | 54.28 | 57.02 | 42.51 |
| Claude 3 Haiku | 29.57 | 6.12 | 40.77 | 24.67 | 73.67 | 56.17 | 60.65 | 46.16 | 43.84 | 25.98 | 79.38 | 72.72 | 54.72 | 41.65 | 33.65 | 17.77 | 54.79 | 28.02 | 78.21 | 65.01 | 54.26 | 38.98 |
| GPT-4o Mini | 72.07 | 62.85 | 59.23 | 47.93 | 75.98 | 60.63 | 62.36 | 50.36 | 13.03 | -5.17 | 70.97 | 68.59 | 86.48 | 82.34 | 26.73 | 9.03 | 60.63 | 32.67 | 68.59 | 50.26 | 59.61 | 45.95 |
| *Medical Domain Models* | | | | | | | | | | | | | | | | | | | | | | |
| LLaVA-Med v1.5-7B | 68.17 | 58.36 | 32.55 | 15.28 | 35.59 | 27.00 | 58.48 | 45.85 | 7.22 | -6.96 | 48.04 | 39.72 | 37.12 | 22.08 | 20.19 | 0.08 | 41.88 | 21.82 | 46.37 | 30.96 | 38.73 | 26.08 |
| Med-Flamingo-7B | 28.54 | 6.04 | 16.78 | 0.97 | 11.74 | 5.47 | 17.42 | 3.60 | 20.42 | -3.50 | 10.84 | 13.46 | 28.11 | 6.10 | 18.85 | -0.48 | 3.33 | -2.03 | 13.46 | 4.00 | 16.95 | 3.36 |
| HuatuoGPT-Vision-7B | 81.11 | 74.80 | 53.69 | 40.53 | 87.19 | 77.80 | 67.96 | 56.04 | 42.78 | 25.39 | 80.00 | 73.48 | 85.19 | 80.28 | 22.12 | 1.71 | 76.46 | 56.42 | 90.81 | 83.79 | 68.08 | 58.76 |
| MedGemma-4B | 61.40 | 49.26 | 53.69 | 40.35 | 76.16 | 61.71 | 70.30 | 58.68 | 48.94 | 32.66 | 70.00 | 86.32 | 70.82 | 61.03 | 15.58 | -4.46 | 58.13 | 31.85 | 86.32 | 76.09 | 61.13 | 49.35 |
| HealthGPT-3.8B | 77.41 | 70.13 | 54.70 | 41.55 | 89.50 | 81.14 | 71.85 | 61.08 | 40.14 | 20.20 | 82.68 | 77.06 | 75.11 | 66.79 | 25.96 | 9.45 | 85.42 | 66.88 | 87.82 | 78.79 | 68.50 | 58.38 |
| Med-R1-2B | 77.41 | 69.87 | 48.83 | 33.88 | 84.16 | 72.20 | 65.94 | 52.23 | 37.32 | 17.19 | 84.33 | 79.21 | 83.26 | 77.63 | 46.73 | 34.00 | 64.58 | 38.06 | 90.60 | 83.78 | 67.80 | 56.94 |
| Lingshu-7B | 80.08 | 73.79 | 57.72 | 46.21 | 89.86 | 82.02 | 72.63 | 61.63 | 40.49 | 20.39 | 78.44 | 91.24 | 82.62 | 76.85 | 30.96 | 13.64 | 83.33 | 63.93 | 91.24 | 84.57 | 70.74 | 61.43 |
| MedVLM-R1-2B | 75.36 | 67.45 | 37.75 | 21.22 | 64.95 | 44.69 | 59.72 | 43.65 | 27.64 | 9.50 | 69.82 | 80.34 | 84.33 | 79.37 | 25.96 | 9.00 | 48.75 | 22.22 | 80.34 | 66.47 | 57.46 | 44.39 |
| DIYHealthGPT-3.8B | **97.74** | **96.98** | **59.73** | **48.26** | **95.02** | **90.76** | **85.23** | **78.89** | **83.10** | **77.57** | **99.73** | **99.57** | **97.85** | **97.14** | **51.90** | **40.02** | **98.13** | **95.21** | **99.57** | **99.19** | **86.80** | **82.36** |

Table 2: Comparison of DIYHealthGPT with baselines under *open-QA* settings in DIYHealthBench.

| Model | S2D | | DR | | MAG | | Diabetes | | Obesity | | Heart | | CKD | | Food | | Sleep | | Skin | | Oral | | Avg | |
|---|---|---|---|---|---|---|---|---|---|---|---|---|---|---|---|---|---|---|---|---|---|---|---|---|
| | F1-Bio | RL | F1-Bio | RL | F1-Bio | RL | F1-Bio | RL | F1-Bio | RL | F1-Bio | RL | F1-Bio | RL | F1-Bio | RL | F1-Bio | RL | F1-Bio | RL | F1-Bio | RL | F1-Bio | RL |
| *General Domain Models* | | | | | | | | | | | | | | | | | | | | | | | | |
| LLaVA-1.5-7B | 68.43 | 5.25 | 63.29 | 3.97 | 78.04 | 34.92 | 76.83 | 14.84 | 72.23 | 11.58 | 72.65 | 11.22 | 74.98 | 11.40 | 66.29 | 3.32 | 73.88 | 11.58 | 77.41 | 23.81 | 79.29 | 22.75 | 73.03 | 14.06 |
| InstructBLIP-7B | 58.24 | 2.28 | 58.96 | 4.02 | 69.21 | 5.91 | 66.46 | 8.75 | 60.68 | 4.48 | 69.02 | 2.65 | 70.33 | 17.02 | 65.45 | 8.35 | 45.91 | 8.73 | 63.37 | 11.42 | | | 60.77 | 7.75 |
| Llama 3.2-11B | 66.22 | 4.67 | 62.17 | 4.07 | 79.94 | 40.98 | 73.20 | 12.09 | 70.51 | 11.56 | 67.53 | 7.71 | 73.56 | 12.19 | 68.87 | 6.55 | 69.37 | 7.27 | 73.25 | 16.46 | 73.85 | 15.89 | 70.77 | 12.68 |
| Yi-VL-6B | 66.06 | 4.33 | 62.02 | 4.34 | 83.99 | 34.86 | 77.49 | 16.62 | 71.49 | 11.19 | 72.69 | 12.84 | 76.58 | 14.70 | 76.53 | 13.51 | 72.15 | 13.01 | 72.78 | 12.66 | 88.82 | 22.40 | 73.83 | 15.16 |
| InternVL3-8B | 63.63 | 2.85 | 60.97 | 3.03 | 76.97 | 33.01 | 75.89 | 17.78 | 67.9 | 8.24 | 68.91 | 9.26 | 73.01 | 8.63 | 67.64 | 4.92 | 66.46 | 7.14 | 74.06 | 21.78 | 74.61 | 20.09 | 70.00 | 12.43 |
| Qwen2.5-VL-7B | 64.90 | 3.14 | 65.01 | 6.76 | 76.36 | 23.86 | 76.06 | 15.31 | 68.90 | 8.40 | 69.77 | 5.00 | 70.57 | 7.00 | 78.00 | 26.61 | 71.03 | 7.00 | 76.57 | 24.09 | 77.66 | 22.32 | 72.28 | 13.93 |
| Gemma 3-4B | 62.48 | 1.84 | 60.57 | 2.14 | 71.17 | 26.14 | 69.55 | 8.14 | 65.87 | 6.18 | 64.31 | 5.43 | 64.38 | 3.17 | 67.66 | 5.44 | 65.14 | 5.06 | 68.28 | 10.09 | 68.74 | 10.30 | 66.20 | 7.63 |
| Claude 3 Haiku | 67.24 | 4.81 | 62.15 | 4.41 | 79.39 | 35.84 | 78.02 | 17.87 | 72.23 | 11.48 | 72.06 | 12.36 | 72.31 | 11.05 | 67.04 | 3.77 | 73.81 | 10.90 | 77.30 | 22.14 | 78.66 | 22.92 | 72.86 | 14.94 |
| GPT-4o Mini | 66.19 | 4.24 | 62.15 | 4.02 | 80.92 | 45.17 | 75.58 | 15.09 | 71.01 | 11.90 | 68.57 | 8.73 | 75.51 | 12.64 | 70.66 | 10.14 | 70.84 | 8.44 | 74.92 | 22.87 | 75.08 | 21.09 | 71.95 | 14.94 |
| *Medical Domain Models* | | | | | | | | | | | | | | | | | | | | | | | | |
| LLaVA-Med v1.5-7B | 72.11 | 9.09 | 65.98 | 7.93 | 76.33 | 24.31 | 77.93 | 17.77 | 74.94 | 15.95 | 74.66 | 18.76 | 77.27 | 17.14 | 70.75 | 4.97 | 75.72 | 19.08 | 78.58 | 26.32 | 79.12 | 24.82 | 74.82 | 16.88 |
| Med-Flamingo-7B | 59.70 | 2.51 | 56.57 | 3.36 | 59.98 | 7.80 | 65.47 | 8.70 | 57.95 | 5.80 | 56.98 | 2.44 | 62.64 | 5.24 | 51.91 | 0.45 | 54.83 | 3.71 | 61.87 | 8.14 | 62.43 | 7.49 | 59.12 | 5.06 |
| HuatuoGPT-Vision-7B | 67.77 | 4.13 | 66.49 | 4.41 | 71.86 | 13.03 | 77.76 | 15.18 | 72.56 | 10.80 | 70.14 | 8.61 | 73.27 | 8.61 | 65.08 | 2.14 | 72.55 | 8.23 | 77.58 | 21.53 | 80.39 | 26.83 | 72.12 | 11.17 |
| MedGemma-4B | 79.55 | 34.99 | 68.90 | 8.42 | 71.17 | 15.84 | 72.27 | 11.20 | 61.48 | 4.67 | 64.91 | 3.64 | 67.47 | 8.83 | 69.7 | 13.33 | 68.68 | 7.99 | 65.29 | 5.00 | 68.22 | 9.83 | 68.88 | 11.25 |
| HealthGPT-3.8B | 67.87 | 4.77 | 63.59 | 4.33 | 77.53 | 25.52 | 79.47 | 18.56 | 71.81 | 11.71 | 70.18 | 7.78 | 76.05 | 13.27 | 67.36 | 4.15 | 73.29 | 10.66 | 80.34 | 29.39 | 82.92 | 34.47 | 73.67 | 14.96 |
| Med-R1-2B | 64.42 | 3.53 | 61.54 | 3.20 | 76.02 | 27.94 | 74.94 | 14.22 | 68.68 | 9.43 | 63.15 | 5.45 | 74.72 | 10.99 | 66.30 | 3.26 | 65.26 | 5.15 | 75.11 | 20.71 | 77.25 | 20.32 | 69.76 | 11.29 |
| Lingshu-7B | 68.65 | 5.40 | 64.71 | 5.11 | 74.23 | 18.95 | 79.64 | 19.14 | 72.95 | 13.42 | 71.19 | 10.34 | 77.50 | 10.78 | 62.82 | 6.85 | 73.82 | 10.09 | 79.74 | 27.36 | 81.86 | 31.19 | 73.77 | 14.66 |
| MedVLM-R1-2B | 64.58 | 3.75 | 61.01 | 3.61 | 77.12 | 26.26 | 74.8 | 13.86 | 70.47 | 11.28 | 68.86 | 9.62 | 75.34 | 13.98 | 69.82 | 8.73 | 72.08 | 10.49 | 75.62 | 20.80 | 76.04 | 18.47 | 71.43 | 12.80 |
| DIYHealthGPT-3.8B | **86.45** | **45.41** | **75.47** | **22.10** | **89.62** | **60.20** | **87.45** | **44.64** | **84.44** | **42.97** | **82.26** | **33.01** | **90.57** | **64.80** | **97.92** | **90.28** | **85.55** | **43.33** | **88.31** | **55.79** | **92.68** | **70.65** | **87.34** | **52.11** |

### 6.3 Ablation Study and In-Depth Analysis

**Ablation Study of H$^2$LoRA.** We propose H$^2$LoRA, which integrates a LoRA expert mixture with Hyper LoRA adaptation to enable DIYHealthGPT to acquire multi-task knowledge while retaining instance-aware personalization. To assess the contribution of each component, we conduct a detailed ablation study, with results shown in Table 3. Relative to w/o H$^2$LoRA, incorporating either the Expert Mixture or Hyper LoRA leads to notable improvements across all metrics, confirming the effectiveness of both mechanisms. Further, combining the two within H$^2$LoRA yields consistent gains, showing that the mechanisms are both effective individually and complementary when integrated. Hyperparameter study on the number of experts is presented in Appendix E.3.

**Effectiveness of H$^2$LoRA against Counterparts.** H$^2$LoRA endows an LLM with the ability to achieve cross-task generalization while retaining instance-level personalization. We comprehensively evaluate H$^2$LoRA in comparison with existing PEFT methods, including LoRA (Hu et al., 2022), MoELoRA (Luo et al., 2024a), and HyperLoRA (Lv et al., 2024), with results presented in Figure 4. The compared LoRA variants yield only modest performance, with MoELoRA slightly leading on F1-bio and HyperLoRA marginally outperforming on ACC. In contrast, H$^2$LoRA consistently surpasses these LoRA methods, highlighting its superior effectiveness.

**Investigation of Training Stages.** We design a unified four-stage training pipeline that equips DIYHealthGPT with multi-task learning in home-care settings and facilitates effective knowledge

Table 3: Ablation study of H$^2$LoRA within DIYHealthGPT.

| | S2D | | Heart | | Food | | Avg. | | S2D | | Heart | | Food | | Avg. | |
|---|---|---|---|---|---|---|---|---|---|---|---|---|---|---|---|---|
| | ACC | MCC | ACC | MCC | ACC | MCC | ACC | MCC | F1-Bio | RL | F1-Bio | RL | F1-Bio | RL | F1-Bio | RL |
| w/o H$^2$LoRA | 95.28 | 93.71 | 64.79 | 53.02 | 94.42 | 92.65 | 84.83 | 79.79 | 85.14 | 40.30 | 81.01 | 30.75 | 96.41 | 83.17 | 87.52 | 51.41 |
| w/o Expert Mixture | 97.33 | 96.43 | 80.28 | 73.80 | 97.00 | 96.01 | 91.54 | 88.75 | 85.74 | 43.22 | 81.68 | 31.26 | 97.84 | 89.79 | 88.42 | 54.76 |
| w/o Hyper LoRA | 97.33 | 96.43 | 79.93 | 73.30 | 97.42 | 96.57 | 91.56 | 88.77 | 85.73 | 43.27 | 81.63 | 31.40 | 97.71 | 89.13 | 88.36 | 54.60 |
| w/ H$^2$LoRA | **97.74** | **96.98** | **83.10** | **77.57** | **97.85** | **97.14** | **92.90** | **90.56** | **86.45** | **45.41** | **82.26** | **33.01** | **97.92** | **90.28** | **88.88** | **56.23** |

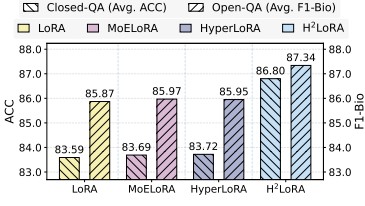

Figure 4: Comparison of H$^2$LoRA with existing LoRA methods.

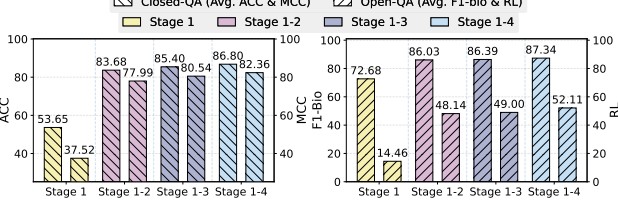

Figure 5: Performance across training stages.

transfer. To assess the contribution of each stage, we add them sequentially and report the results in Figure 5. As outlined in Section 4.4, the stages correspond to Cross-Modal Alignment → Medical Domain Adaptation → Task-Specific Expert Training → Cross-Task Knowledge Transfer, with each stage introducing a distinct capability. The progressive performance gains highlight both the necessity and complementarity of all four stages. In particular, Stage 2 achieves marked improvement through effective medical adaptation, Stage 3 specializes in task-specific patterns, and Stage 4 further enables knowledge sharing across experts rather than focusing on isolated tasks.

**DIYHealthGPT's Representation Visualization.** We visualize the representations derived by DIYHealthGPT in open-QA in Figure 6 using t-SNE (Maaten & Hinton, 2008). Distinct clusters emerge for most tasks, reflecting low intra-task variance and large inter-task margins, which indicates that DIYHealthGPT captures task-aware features while preserving individual-level nuances. An overlap is observed between DR and MAG, which is reasonable since MAG often involves advice related to drug management. This aligns with the performance gains in both closed- and open-QA, underscoring that DIYHealthGPT yields both task-structured and personalized representations.

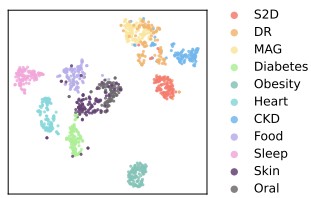

Figure 6: Visualization of representations learned by DIYHealthGPT.

**Clinical Expert Review.** We conduct a clinical expert review by randomly sampling 500 open-QA pairs from DIYHealthBench and assigning them to clinical experts for assessment of clinical significance. Each answer was evaluated according to three criteria: (i) Conciseness: providing direct and succinct answers for non-expert readers while avoiding unnecessary details; (ii) Correctness: ensuring factual accuracy of the response; (iii) Relevance: evaluating the degree to which the response avoids introducing irrelevant content. The answers are rated on a 1–6 scale, with 1 indicating the best and 6 the worst. The results are shown in Figure 7. As illustrated in Figure 7(a), DIYHealthGPT is the clear first preference of clinical experts, demonstrating the highest clinical utility. Further, Figure 7(b) shows that DIYHealthGPT con-

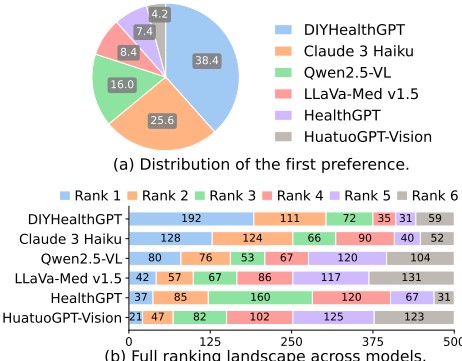

Figure 7: Results of the clinical expert review.

centrates mass in the top ranks, whereas other models exhibit heavier tail ranks. Among competitors, Claude-3 Haiku is the strongest, achieving the largest share of Rank 2 answers, yet it trails DIYHealthGPT in decisive first-preference counts. Overall, clinical experts consistently judged open-QA answers of DIYHealthGPT to be the most faithful and clinically meaningful. In addition to clinical expert review, evaluations conducted with GPT-5 are provided in Appendix E.2.

**Comparison With Fine-tuned Baselines.** To examine whether the performance gain of DIYHealthGPT is solely due to training on DIYHealth-900K, we conduct fine-tuning experiments on two representative models, Gemma 3-4B and LLaVA-Med v1.5-7B, using exactly the same training data with our limited computational resources. As shown in Table 4, DIYHealthGPT-3.8B consistently

Table 4: Comparison with fine-tuned baselines.

| Model | Closed-QA | | Open-QA | |
|---|---|---|---|---|
| | ACC | MCC | F1-Bio | RL |
| Gemma 3-4B | 80.96 | 74.42 | 84.72 | 41.59 |
| LLaVA-Med v1.5-7B | 77.58 | 69.51 | 86.28 | 49.63 |
| DIYHealthGPT-3.8B | **86.80** | **82.36** | **87.34** | **52.11** |

outperforms Gemma 3-4B and LLaVA-Med v1.5-7B by a substantial margin on the four evaluation metrics, despite LLaVA-Med having a larger model size. These results indicate that the observed improvements cannot be attributed only to the dataset. Instead, the joint effect of DIYHealth-900K, the training strategy, and the $H^2$LoRA architecture contributes to the model's effectiveness.

## 7    CONCLUSIONS

The blossoming of generative AI signals new opportunities for accessible and personalized health-care beyond traditional clinical settings. In response, we propose DIYHealth Suite, a home-based health management framework comprising DIYHealth-900K, DIYHealthBench, and DIYHealthGPT. DIYHealth-900K curates diverse multimodal inputs from consumer-grade devices, enabling AI systems to operate in everyday home contexts. At the core, DIYHealthGPT leverages our proposed $H^2$LoRA technique to balance cross-task generalization with instance-level personalization, deliver-ing conversational and personalized support across health domains. DIYHealthBench establishes the first dedicated evaluation protocol for dynamic, non-clinical home environments, ensuring stan-dardized and fair comparison. Comprehensive experiments on 11 home care tasks demonstrate that DIYHealthGPT consistently surpasses state-of-the-art in both open-QA and closed-QA. Notably, tasks including Sleep, Heart, Obesity, Diabetes, and DR continue to pose significant challenges for both general and medical VLMs. We highlight these tasks as priority areas for future research on home care reasoning and multimodal health understanding. Collectively, these components define a new paradigm for non-clinical healthcare AI, rooted in real-world scenarios and designed for extensibility, usability, and rigorous evaluation.

ETHICS STATEMENT

This study is based exclusively on publicly available datasets collected from 20 open sources. No hospital-held or proprietary clinical data are used. All datasets are distributed under their respective licenses, and their use strictly adheres to the corresponding terms and conditions. The datasets are de-identified by the original providers, and no personally identifiable information is accessible to the authors. The data are used solely for the purposes of this research.

REPRODUCIBILITY STATEMENT

The code and data for our project are available at `https://anonymous.4open.science/r/DIYHealthGPT-codes-E71A`. Detailed descriptions of hyperparameters and experimental settings are provided in Appendix D.

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

APPENDIX

# A    NOTATION TABLE

To provide a comprehensive overview of the notations used throughout the paper, we present a summary of notations in Table 5 as a quick reference to facilitate the understanding and recall of each symbol.

Table 5: Notations.

| Notation | Description |
|---|---|
| $\mathcal{I} \in \mathbb{R}^{H \times W \times 3}$ | Input image of height $H$, width $W$, and three RGB channels. |
| $\mathcal{T} = \{t_1, \ldots, t_{L_t}\}$ | Textual symptom description consisting of $L_t$ tokens. |
| $\mathcal{V}_{txt}$ | Vocabulary of the backbone language model. |
| $\mathcal{E}_v(\cdot)$ | Pretrained vision encoder. |
| $\mathcal{E}_t(\cdot)$ | Pretrained textual encoder. |
| $\mathcal{V} \in \mathbb{R}^{L_v \times d_v}$ | Visual embeddings with $L_v$ tokens, each of dimension $d_v$. |
| $\mathcal{U} \in \mathbb{R}^{L_t \times d}$ | Textual embeddings with $L_t$ tokens, each of dimension $d$. |
| $\mathcal{P}_v : \mathbb{R}^{d_v} \to \mathbb{R}^d$ | Learnable projection function aligning visual and textual embeddings. |
| $\mathcal{Z} \in \mathbb{R}^{(L_v + L_t) \times d}$ | Unified multimodal representation combining visual and textual embeddings. |
| $\mathcal{M}_{LLM}$ | Backbone large language model. |
| $\Theta$ | Frozen pretrained parameters of $\mathcal{M}_{LLM}$. |
| $\Theta_{H^2L} = \{\mathcal{A}, \mathcal{B}, \mathcal{R}\}$ | Task-adaptive parameters introduced in H$^2$LoRA. |
| $N$ | Total number of home care tasks. |
| $T = \{1, \cdots, N\}$ | Set of home care tasks. |
| $\mathcal{A} = \{\mathbf{A}^t, \Delta\mathbf{A}^t\}_{t=1}^N$ | Set of shared projection matrices in H$^2$LoRA. |
| $\mathcal{B} = \{\mathbf{B}^t, \Delta\mathbf{B}^t\}_{t=1}^N$ | Set of expert matrices in H$^2$LoRA. |
| $\mathcal{R}$ | Routing parameters for integrating task-level outputs. |
| $\mathbf{B}_k^t$ | The $k$-th expert mixture matrix of task $t$. |
| $K$ | Number of expert mixture matrices per task. |
| $\mathcal{W}^t \in \mathbb{R}^K$ | Mixture weights over the $K$ expert matrices $\{\mathbf{B}_1^t, \ldots, \mathbf{B}_K^t\}$ for task $t$. |
| $\hat{\mathcal{W}}^t$ | Expanded expert weight vector for task $t$. |
| $\mathcal{H}_A(\cdot)$ | Hypernetwork of the shared projection. |
| $\mathcal{H}_B(\cdot)$ | Hypernetwork of the expert mixture matrix. |
| $\Delta\mathbf{A}^t$ | Instance-aware offset for the shared projection of task $t$. |
| $\Delta\mathbf{B}_k^t$ | Instance-aware offset for the $k$-th expert mixture matrix $\mathbf{B}_k^t$ of task $t$. |
| $\Delta\mathbf{B}^t$ | Instance-aware offset for MoE-driven expert mixture matrix $\mathbf{B}^t$ of task $t$. |
| $\beta = (\beta^1, \ldots, \beta^N)$ | Inter-task fusion mixture weights across $N$ tasks. |
| $\mathcal{O}_{H^2LoRA}^t$ | Output of H$^2$LoRA for task $t$. |
| $\mathcal{O}_{H^2LoRA}$ | Overall output of H$^2$LoRA aggregated across all tasks. |

# B    EXTENDED RELATED WORK

The development of DIYHealth Suite is grounded on a broad and evolving body of research spanning pre-LLM healthcare AI, general-purpose LLMs, and emerging medical foundation models. In this section, we review key advances across these interconnected areas to contextualize our contributions.

**Pre-LLM Healthcare Techniques.** Before the advent of LLMs, deep learning rapidly advanced healthcare applications, particularly in medical imaging and EHR analysis. Early work—largely driven by progress in computer vision—centered on interpreting medical images such as MRI, ultrasound, and fundus photography for disease diagnosis and risk assessment. Prominent examples include automated detection of Alzheimer's disease from brain MRI (Brosch et al., 2013; Helaly et al., 2022), segmentation of knee cartilage in osteoarthritis (Prasoon et al., 2013), and lesion analysis for conditions such as multiple sclerosis and breast nodules (Yoo et al., 2014; Cheng et al., 2016a). Convolutional neural networks (CNNs) achieved diagnostic performance comparable to that of clinical

experts, including dermatologists classifying skin lesions (Esteva et al., 2017) and ophthalmologists screening for diabetic retinopathy (Gulshan et al., 2016). In parallel, deep learning began reshaping EHR analysis by enabling predictive modeling over both structured inputs (e.g., diagnoses, laboratory tests) and unstructured clinical narratives. Supervised models, such as CNNs and Recurrent Neural Networks (RNNs) equipped with Long Short-Term Memory (LSTM) or Gated Recurrent Unit (GRU) architectures, demonstrated superior performance in tasks including disease onset prediction for congestive heart failure (Cheng et al., 2016b), disease progression modeling (Pham et al., 2016), and automated diagnosis and medication recommendation (Choi et al., 2016a). Unsupervised approaches, including stacked denoising autoencoders (Miotto et al., 2016), restricted Boltzmann machines (Liang et al., 2014), and neural embedding methods (Choi et al., 2016b), facilitated the extraction of latent patient representations for downstream applications such as disease risk stratification and phenotype discovery. Although these innovations established a solid foundation for data-driven healthcare, they are typically limited by narrowly scoped tasks, institution-specific datasets, and the absence of standardized benchmarks for evaluation.

**LLMs for General Reasoning and Dialogue.** LLMs have become central to recent advances in natural language understanding and generation, fundamentally advancing the capabilities of machines to perform general-purpose reasoning and engage in coherent, human-like dialogue (Chowdhery et al., 2023; Touvron et al., 2023). Representative models such as GPT-3 (Brown et al., 2020) and GPT-4 (OpenAI, 2023), trained on large-scale corpora, exhibit remarkable zero-shot and few-shot generalization across diverse natural language tasks. Beyond text-only models, the emergence of multimodal LLMs (MLLMs) such as LLaVA-1.5 (Liu et al., 2023), Llama 3.2 (Dubey et al., 2024), Qwen2.5-VL (Bai et al., 2025), and GPT-4o (Achiam et al., 2023) further extends these capabilities by enabling joint reasoning across language and visual modalities. These capabilities stem from their architectural scale—often comprising hundreds of millions to hundreds of billions of parameters—as well as sophisticated pre-training and alignment techniques (Wei et al., 2022; Fedus et al., 2022). Despite such strengths, the application of general-purpose LLMs in specialized domains such as healthcare remains constrained by the absence of domain-specific knowledge and the lack of grounding in real-world physiological data, particularly in low-resource and home-based healthcare contexts.

**Medical Foundation Models.** To bridge the domain gap between general-purpose LLMs and medical applications, a new generation of medical foundation models has emerged. These include both text-only architectures and multimodal frameworks, referred to as Med-LLMs and Med-LVLMs. Text-based models such as Med-PaLM (Singhal et al., 2023), BiomedGPT (Luo et al., 2024b), and HuatuoGPT (Chen et al., 2024a) have shown strong performance on clinical question answering (QA) benchmarks by leveraging curated biomedical corpora and large-scale synthetic datasets. BiomedGPT, in particular, achieves a compact yet competitive architecture through cross-modal pre-training and decoder alignment techniques. In parallel, Med-LVLMs have advanced multimodal medical understanding. Models such as LLaVA-Med (Li et al., 2023a), Med-Flamingo (Moor et al., 2023), MedGemma (Sellergren et al., 2025), Med-R1 (Lai et al., 2025), Lingshu (Xu et al., 2025), and MedVLM-R1 (Pan et al., 2025) align visual encoders with textual LLMs to enable diagnostic reasoning over imaging data. HealthGPT (Lin et al., 2025) extends this direction by supporting multimodal comprehension and generation on multiple medical tasks, while EyecareGPT (Li et al., 2025) devises a resolution mechanism and a layer-wise dense connector to improve ophthalmic visual understanding. These systems demonstrate potential for tasks such as image captioning, visual question answering (VQA), and differential diagnosis.

Despite these advances, the existing Med-LVLMs are trained and evaluated primarily on data from professional medical devices, such as radiographic images and pathology slides, which are not accessible in daily life scenarios. Consequently, their applicability to home-based health management remains limited. The absence of portability, limited adaptability to informal data, and insufficient support for personalized inference underscore the need to redesign medical AI architectures that can operate effectively with consumer-grade devices such as smartphones, wearables, and smart home sensors. To this end, we propose DIYHealth Suite as an innovative solution for home care health management, comprising three core components—DIYHealth-900K, DIYHealthBench, and DIYHealthGPT—which are elaborated in Sections 3, 4, and 5, respectively.

## C  CONSTRUCTION DETAILS OF DIYHEALTH-900K

### C.1  TASK DESIGN AND FUNCTIONAL DESCRIPTIONS

A detailed overview of the tasks included in the DIYHealth-900K dataset is provided in Table 6. To reflect the complexity and diversity of real-world home care scenarios, we categorize the tasks into three major groups: personalized health management, chronic disease risk assessment, and daily health monitoring. The first category, personalized health management, encompasses core tasks such as symptom-based diagnosis, drug recommendation, and tailored medical advice generation, which are essential for supporting early clinical decision-making. In the context of chronic conditions, DIYHealth-900K includes risk assessments for diabetes, obesity, cardiovascular disease, and kidney health, drawing on both self-reported symptoms and home-acquired signals. Daily health monitoring tasks address routine wellness dimensions, such as dietary intake analysis, sleep quality evaluation, skin condition assessment, and oral health screening. For each task, we curate or adapt relevant datasets to align with the characteristics of home care, with particular emphasis on multimodal inputs, real-world variability, and nonclinical supervision, to ensure their applicability to AI models deployed in home environments.

Table 6: Home care task design in DIYHealth-900K with corresponding functional descriptions.

| Task Category | Home Care Task | Task Description |
|---|---|---|
| Personalized Health Management | Symptom-to-Diagnosis (S2D) | Predict potential diagnoses based on symptom descriptions |
| | Drug Recommendation (DR) | Suggest appropriate medications based on symptoms and suspected conditions |
| | Medical Advice Generation (MAG) | Generate personalized medical advice tailored to individual health concerns |
| Chronic Disease Risk Assessment | Diabetes Detection (Diabetes) | Assess for potential diabetes risks based on retinal fundus images |
| | Obesity Detection (Obesity) | Identify obesity risk through lifestyle, dietary, and physical measurements |
| | Heart Health (Heart) | Monitor cardiovascular health and detect early signs of heart conditions |
| | Kidney Health (CKD) | Detect risks of chronic kidney disease through symptoms and home test inputs |
| Daily Health Monitoring | Diet Management (Food) | Manage dietary habits by analyzing intake and providing dietary recommendations |
| | Sleep Health (Sleep) | Evaluate sleep quality and patterns to provide improvement suggestions |
| | Skin Health (Skin) | Assess skin conditions and detect abnormalities from patient inputs or images |
| | Oral Health (Oral) | Screen for common oral issues and provide oral health recommendations |

### C.2  DATA SOURCES AND DETAILED STATISTICS

The task-specific data sources and corresponding statistics of DIYHealth-900K are summarized in Table 7. To fully leverage real-world data, we curate DIYHealth-900K from 20 publicly available data sources selected for their alignment with the target QA tasks. All included data are readily obtainable in home settings, ensuring that the dataset reflects scenarios accessible to end users without reliance on specialized clinical equipment. For instance, in VQA tasks such as diet management, oral health, skin health, and diabetes detection (Naz et al., 2024; Rogers et al., 2021), the images consist of everyday photographs that can be captured using mobile devices, rather than specialized medical imaging. In heart health and sleep health tasks, the physiological signals are collected from portable, home-use devices, ensuring that the data reflect measurements users can realistically obtain outside clinical environments. Specifically, we develop a translation script to convert the raw physiological signals into images suitable for VQA tasks. For the heart health task, the data are derived from ECG recordings. Due to the hardware limitations of portable devices (e.g., Apple Watch) Li et al. (2022), only Lead-I signals are available; therefore, we extract the Lead-I channel from the original 12-lead ECG data. The extracted signals are subsequently processed using NeuroKit2's [1] clean function to perform noise filtering, artifact removal, and baseline correction. For recordings exceeding 10 seconds, a high-quality 10-second segment is selected for downstream processing (Wagner et al., 2020). For the sleep health task, which involves triaxial accelerometry data, we first extract a 30-second segment for each sample (Wang et al., 2024). Each segment is then subjected to a cleaning procedure to reduce noise and eliminate artifacts (Moscato et al., 2022).

For text-only QA tasks, including drug recommendation, symptom-to-diagnosis reasoning, medical advice generation, obesity detection, and kidney health, we construct home-care–oriented datasets through a carefully designed multi-stage pipeline. In the first stage, we extract candidate samples

---

[1] https://github.com/neuropsychology/NeuroKit

from a variety of publicly available medical and health-related datasets. We then apply a filtering procedure to exclude records, descriptions, or cases that are clearly irrelevant or impractical in a home setting (e.g., those requiring hospital-grade imaging or laboratory tests). After curating this subset, we generate corresponding question–answer pairs. Because many of the original data sources, such as questionnaires, are not naturally phrased as questions, we employ an LLM to rephrase them into fluent, conversational question formats. For VQA tasks covering diabetes detection, diet management, skin health, and oral health, we utilize publicly available image datasets and derive questions from their associated labels. These questions are likewise rephrased into fluent natural language by an LLM. Finally, human experts validate both generated questions and their associated answers to ensure correctness.

Table 7: Data sources and corresponding statistics for each task in DIYHealth-900K.

| Task | Total | Open-QA | Closed-QA | Type | Data Source |
|---|---|---|---|---|---|
| Symptom-to-Diagnosis (S2D) | 100861 | 50441 | 50420 | QA | DDXPlus [2] |
| Drug Recommendation (DR) | 9949 | 9949 | - | QA | MIMIC-III [3] and MIMIC-IV [4] |
| Medical Advice Generation (MAG) | 222190 | 180554 | 41636 | QA | MedQuAD [5], MedQA-USMLE [6], PubMedQA [7], MedAlpaca [8], and MedMCQA [9] |
| Diabetes Detection (Diabetes) | 100000 | 42687 | 57313 | VQA | Messidor Diabetic Retinopathy [10] |
| Obesity Detection (Obesity) | 25482 | 9367 | 16115 | QA | ObesityDataSet [11] |
| Heart Health (Heart) | 88454 | 42926 | 45528 | VQA | PTB [12], PTB-XL [13], and ECG-arrhythmia [14] |
| Kidney Health (CKD) | 8236 | 4118 | 4118 | QA | CKD Source Dataset1 [15] and CKD Source Dataset2 [16] |
| Diet Management (Food) | 100000 | 49851 | 50149 | VQA | Food-101 [17] |
| Sleep Health (Sleep) | 103902 | 51952 | 51950 | VQA | Dreamt [18] and Applewatch [19] |
| Skin Health (Skin) | 64010 | 32065 | 31945 | VQA | Fitzpatrick17k [20] |
| Oral Health (Oral) | 90807 | 46355 | 44452 | VQA | Oral Diseases[21] |
| Total | 913891 | 520255 | 393636 | QA&VQA | 20 Publicly Available Data Sources |

## C.3 PROMPT DESIGN

Claude 3 Haiku is employed to rephrase the source data of each dataset into first-person or third-person patient statements and descriptions. As an example, in the S2D open-QA setting, we illustrate the prompt design in Figure 8. To ensure reliability, we incorporate a human-in-the-loop validator to monitor and guarantee the quality of the rephrased data.

[2] https://figshare.com/articles/dataset/DDXPlus_Dataset_English_/22687585
[3] https://physionet.org/content/mimiciii/1.4/
[4] https://physionet.org/content/mimiciv/3.0/
[5] https://www.kaggle.com/datasets/pythonafroz/medquad-medical-question-answer-for-ai-research
[6] https://www.kaggle.com/datasets/moaaztameer/medqa-usmle
[7] https://pubmedqa.github.io/
[8] https://github.com/kbressem/medAlpaca
[9] https://medmcqa.github.io/
[10] https://www.kaggle.com/datasets/ascanipek/eyepacs-aptos-messidor-diabetic-retinopathy/data
[11] https://www.kaggle.com/datasets/aravindpcoder/obesity-or-cvd-risk-classifyregressorcluster/data
[12] https://www.physionet.org/content/ptbdb/1.0.0/
[13] https://physionet.org/content/ptb-xl/1.0.3/
[14] https://physionet.org/content/ecg-arrhythmia/1.0.0/
[15] https://www.kaggle.com/datasets/rabieelkharoua/chronic-kidney-disease-dataset-analysis?resource=download
[16] https://archive.ics.uci.edu/dataset/336/chronic+kidney+disease
[17] http://data.vision.ee.ethz.ch/cvl/food-101.tar.gz
[18] https://physionet.org/content/dreamt/2.1.0/
[19] https://physionet.org/content/sleep-accel/1.0.0/
[20] https://github.com/mattgroh/fitzpatrick17k
[21] https://www.kaggle.com/datasets/salmansajid05/oral-diseases

> **Question**
>
> Task: you are a medical expert, I am giving you a description of the symptoms:
>
> I have pain somewhere, related to my reason for consulting. I have pain somewhere, related to my reason for consulting. Characterize my pain: haunting. I have pain somewhere, related to my reason for consulting. Characterize my pain: sensitive. I have pain somewhere, related to my reason for consulting. Characterize my pain: tugging. I have pain somewhere, related to my reason for consulting. Characterize my pain: burning. I have pain somewhere, related to my reason for consulting. I feel pain somewhere: lower chest. I have pain somewhere, related to my reason for consulting. I feel pain somewhere: upper chest. I have pain somewhere, related to my reason for consulting. I feel pain somewhere: hypochondrium(R). I have pain somewhere, related to my reason for consulting. How intense is the pain? Eleven levels, lower values mean less intense pain. 6. I have pain somewhere, related to my reason for consulting. The pain radiate to another location: lower chest. I have pain somewhere, related to my reason for consulting. The pain radiate to another location: upper chest. I have pain somewhere, related to my reason for consulting. How precisely is the pain located? Eleven levels, lower values mean less precisely. 3. I have pain somewhere, related to my reason for consulting. The speed of pain onset. Eleven levels, lower values indicate slower. 2. I am significantly overweight compared to people of the same height as me. I drink alcohol excessively or I have an addiction to alcohol. I have a hiatal hernia. I have recently had stools that were black (like coal). I think I am pregnant or I am currently pregnant. I have a burning sensation that starts in my stomach then goes up into my throat, and can be associated with a bitter taste in my mouth. I have a cough. My symptoms are worse when lying down and alleviated while sitting up
>
> I want you to understand the symptoms and help me refine the description using a paragraph with the first or third-person symptom statement and follow the format <An individual has <symptoms>>.
> .

> **Claude 3 Haiku**
>
> An individual has pain that is haunting, sensitive, tugging, and burning, located in the lower chest, upper chest, and right hypochondrium, with an intensity level of 6 out of 11. The pain radiates to the lower chest and upper chest, and is moderately precisely located. The pain has a slow onset, and the individual is significantly overweight, has an alcohol addiction, a hiatal hernia, has recently had black stools, is possibly pregnant, experiences a burning sensation that starts in the stomach and goes up into the throat, has a bitter taste in the mouth, and a cough. The symptoms are worse when lying down and alleviated while sitting up.

Figure 8: An example prompt to rephrase raw data into natural human conversational expressions for the S2D task in the open-QA setting.

# D  IMPLEMENTATION DETAILS

## D.1  MODEL DETAILS

**Details on DIYHealthGPT**. DIYHealthGPT is built based on Phi-3-mini (Abdin et al., 2024), a publicly available, pre-trained, lightweight LLM with 3.8B parameters. We employ CLIP-L/14 (Radford et al., 2021) as the visual feature extractor to encode image representations. The extracted visual features are projected into a shared semantic space through two-layer MLPs, aligning them with text tokens. This alignment bridges the modality gap between vision and language, and the fused representations are subsequently fed into the LLM to support multimodal understanding and coherent language generation. The number of expert matrices per task $K$ is 4 in the experiments. The ranks of Low-Rank Expert and Hyper LoRA are 16 and 8, respectively. The model is optimized using a next-token prediction objective with the cross-entropy loss (Liu et al., 2023).

**Implementations on Training Stage 3**. During task-specific expert training in Stage 3, each task $t$ is assigned an $H^2$LoRA block with parameters $(\mathbf{A}^t, \Delta\mathbf{A}^t, \mathbf{B}^t, \Delta\mathbf{B}^t)$, which are trained independently using task-specific data, while other parameters remain frozen. Each block is optimized using the same hyperparameter settings across tasks. The Low-Rank Expert Mixture and Hyper LoRA Adaptation modules within the $H^2$LoRA block are inserted into the linear layers of the backbone LLM as lightweight training layers. Low-Rank Expert Mixture is implemented by a shared matrix $\mathbf{A}^t$ and multiple matrices $\mathbf{B}_k^t$, with a router that aggregates the output of matrices $\mathbf{B}_k^t$. Hyper LoRA follows a similar structure, but its parameters are dynamically generated by hypernetworks $\mathcal{H}_A$ and $\mathcal{H}_B$ rather than learned directly through backward propagation. The hypernetworks learn how to generate parameters conditioned on the instance input. Overall, this implementation allows the model

to acquire task-level specialization and parameter-efficient adaptation before cross-task fusion in Stage 4.

## D.2 BASELINE DETAILS

We provide detailed descriptions of all baseline models evaluated in this study below.

- LLaVA-1.5 (Liu et al., 2023) leverages GPT-4 to generate multimodal instruction-following data and tunes an end-to-end vision-language model, enabling general-purpose visual language understanding.
- InstructBLIP (Dai et al., 2023) enhances BLIP-2 (Li et al., 2023b) by tuning on 26 multimodal datasets. InstructBLIP further introduces an instruction-aware Query Transformer to capture informative features tailored to the provided instruction.
- Llama 3.2 (Dubey et al., 2024) is part of Meta's multimodal LLM series that integrates vision and language in efficient models for both edge deployment and general-purpose AI. Llama 3.2 Vision Instruct is used for comparison.
- Yi-VL (Young et al., 2024) is an open-source multimodal extension of the Yi LLM that integrates vision and language, supporting image understanding, text recognition, and multi-round visual question answering.
- InternVL3 (Zhu et al., 2025) is an open-source multimodal LLM natively pre-trained on both text and multimodal data in a unified framework, enhanced with variable visual position encoding, post-training, and test-time scaling.
- Qwen2.5-VL (Bai et al., 2025) is Alibaba's vision-language model that combines native-resolution visual understanding, object localization, document parsing, and long-video comprehension with general language capabilities.
- Gemma 3 (Team et al., 2025) is Google's open-weight multimodal LLM that supports multilingual, visual understanding, and advanced reasoning capabilities.
- Claude 3 Haiku (Anthropic, 2024) is Anthropic's fastest and most lightweight Claude 3 model, optimized for efficiency and low-latency reasoning while preserving strong language and comprehension abilities.
- GPT-4o Mini (Achiam et al., 2023) is a lightweight, cost-efficient variant of GPT-4o. GPT-4o Mini is optimized for speed while retaining multimodal reasoning across text, vision, and audio.
- LLaVA-Med v1.5 (Li et al., 2023a) is a biomedical LVLM trained on PubMed (Zhang et al., 2023) figure-caption pairs and GPT-4–generated instructions, enabling multimodal conversations.
- Med-Flamingo (Moor et al., 2023) is a multimodal few-shot learner built on OpenFlamingo-9B (Awadalla et al., 2023) and further pre-trained on medical image-text data, enabling generative medical VQA and few-shot adaptation.
- HuatuoGPT-Vision (Chen et al., 2024b) is a medical LVLM trained on refined PubMedVision, a denoised and reformatted medical VQA dataset curated with GPT-4V.
- MedGemma (Sellergren et al., 2025) is Google's domain-adapted variant of Gemma that incorporates biomedical knowledge to support medical text understanding and vision-language tasks.
- HealthGPT (Lin et al., 2025) is a unified medical vision-language model trained with the H-LoRA technique, hierarchical visual perception, and a three-stage learning strategy on the VL-Health dataset.
- Med-R1 (Lai et al., 2025) is an RL-enhanced medical vision-language model that employs Group Relative Policy Optimization to improve reasoning quality, generalization, and reliability across diverse medical imaging tasks.
- Lingshu (Xu et al., 2025) is a medical LVLM trained on a curated multimodal dataset with multi-stage learning and reinforcement learning for enhanced reasoning.
- MedVLM-R1 (Pan et al., 2025)) is a medical LVLM trained with reinforcement learning to generate explicit, human-interpretable reasoning paths.

Table 8: Summary of hyperparameter settings used for training DIYHealthGPT.

| Hyperparameter | Stage 1 | Stage 2 | Stage 3 | Stage 4 |
|---|---|---|---|---|
| Optimizer | AdamW | AdamW | AdamW | AdamW |
| Adapter LR | 1e-4 | 2e-5 | / | / |
| Learning Rate | / | 1e-5 | 1e-5 | 1e-5 |
| Global Batch Size | 64 | 32 | 32 | 32 |
| Weight Decay | 0.01 | 0.1 | 0.1 | 0.01 |
| Dropout Rate | 0 | 0.05 | 0.05 | 0.05 |
| LR Scheduler | Constant | Warm Up | Warm Up | Constant |
| Max Sequence Length | 2048 | 2048 | 2048 | 2048 |

## D.3 TRAINING DETAILS

We display the detailed hyperparameter configurations for DIYHealthGPT's four-stage training process. The specific settings used are listed in Table 8. These hyperparameters are set up following prior studies (Lin et al., 2025; Liu et al., 2023; Li et al., 2023a). The architectural parameters are listed in Table 9.

Table 9: Architectural parameters of DIYHealthGPT.

| Module | Parameter | Value / Shape |
|---|---|---|
| **Tokenizer** | Vocabulary size | 32,064 |
| **Backbone** | Hidden size
# Transformer blocks
Self-attn projections
MLP | 3,072
32
qkv: (3072, 9216), o: (3072, 3072 )
gate_up: (3072, 16384); down: (8192, 3072) |
| **CLIP** | Input resolution
Patch size
Embedding dim
# Transformer blocks | $336 \times 336$
$14 \times 14$
1,024
24 |
| **Expert Mixture of H$^2$LoRA** | LoRA rank
Shared matrix $\mathbf{A}$ (up / down)
Matrix $\mathbf{B}$ (up / down) | 16
(3072,16) / (8192,16)
(16,16384) / (16,3072) |
| **Hyper LoRA of H$^2$LoRA** | Rank
Input
Up-path generator (down / up)
Down-path generator (down / up) | 8
(3072, 8)
(8, 24576) / (8, 131072)
(8, 65536) / (8, 24576) |

## D.4 EXPERIMENTAL ENVIRONMENT

All experiments are conducted on a server equipped with an AMD EPYC 9334 CPU @ 2.7 GHz (32 cores), 128 GB of memory, and an NVIDIA H100 NVL with CUDA 12.9. The operating system is Ubuntu 24.04 running Linux kernel 6.8.0-79-generic.

## D.5 EVALUATION METRICS

### D.5.1 EVALUATION METRICS FOR CLOSED-QA

- **Accuracy.** Accuracy (ACC) measures the proportion of predictions that exactly match the ground-truth labels. It is a standard metric for classification tasks and provides a straightforward indicator of overall correctness.

$$ACC = \frac{TP + TN}{TP + TN + FP + FN} \tag{9}$$

where TP, TN, FP, and FN denote the numbers of true positives, true negatives, false positives, and false negatives, respectively.

- **Matthews Correlation Coefficient.** Matthews Correlation Coefficient (MCC) measures the correlation between the predicted and true labels across all classes by considering the full confusion matrix. Unlike accuracy, which may be inflated under class imbalance, MCC provides a balanced evaluation by jointly accounting for true positives, false positives, false negatives, and inter-class misclassifications. It can be interpreted as a generalized correlation coefficient that reflects how well the prediction distribution aligns with the true label distribution. A value of 1 indicates perfect prediction, 0 corresponds to no better than random guessing, and -1 indicates complete disagreement between predictions and ground truth.

$$\text{MCC} = \frac{\sum_k \sum_l \sum_m C_{kk} C_{lm} - C_{kl} C_{mk}}{\sqrt{\left(\sum_k T_k P_k\right) \left(\sum_k T_k^2 - \sum_k \sum_l C_{kl} C_{lk}\right) \left(\sum_k P_k^2 - \sum_k \sum_l C_{lk} C_{kl}\right)}} \quad (10)$$

where $C$ is the confusion matrix. $C_{kl}$ is the number of samples with ground truth class k and predicted class l. $T_k = \sum_l C_{kl}$ and $P_k = \sum_l C_{lk}$.

### D.5.2 Evaluation Metrics for Open-QA

- **F1-RadGraph.** F1-RadGraph computes entity-level F1 scores based on RadGraph (Jain et al., 2021) evaluation, evaluating the correctness of clinical entity extraction and relation grounding. It is widely used in medical language generation and assesses whether the model generates clinically valid content.

$$\text{F1-RadGraph} = \frac{2 \cdot \text{Precision} \cdot \text{Recall}}{\text{Precision} + \text{Recall}}, \quad (11)$$

where $\text{Precision} = \frac{TP}{TP+FP}, \text{Recall} = \frac{TP}{TP+FN}$. TP, FP, and FN are computed based on RadGraph entity matching.

- **F1-BioBERT.** F1-BioBERT measures the semantic alignment between generated answers and ground truth using BioBERT (Lee et al., 2020) embeddings. It captures domain-specific semantic similarity, making it suitable to evaluate models of medical open-QA tasks. The computation of F1-BioBERT is similar to F1-RadGraph, but TP, FP, and FN are computed by semantic matching using BioBERT similarity.

- **BLEU.** BLEU computes the precision of n-gram overlaps between predictions and ground truth to assess lexical fidelity.

$$\text{BLEU} = \text{BP} \cdot \exp\left(\sum_{n=1}^{N} W_n \log(P_n)\right) \quad (12)$$

where BP is brevity penalty. $P_n$ is the modified n-gram precision. $W_n$ is weight of n-gram precision and $N$ is the maximum n-gram order.

- **ROUGE-L.** ROUGE-L evaluates the longest common subsequence between predicted and ground truth answers, reflecting recall-oriented similarity. It captures global structural overlap and complements BLEU's precision-based evaluation.

$$\text{ROUGE-L} = \frac{(1 + \beta^2) \cdot R_{lcs} \cdot P_{lcs}}{R_{lcs} + \beta^2 \cdot P_{lcs}}, \quad (13)$$

where $R_{lcs} = \frac{\text{LCS}(X,Y)}{|X|}, P_{lcs} = \frac{\text{LCS}(X,Y)}{|Y|}$. $X$ is the ground truth, $Y$ is the predicted text. $\text{LCS}(X, Y)$ is the length of the longest common subsequence. $\beta$ controls the weighting of recall versus precision.

Together, these metrics, ACC, MCC, F1-RadGraph, F1-BioBERT, BLEU, and ROUGE-L, jointly assess correctness, robustness, clinical grounding, semantic fidelity, and textual similarity, offering a balanced and reliable evaluation across the diverse tasks in DIYHealthBench.

## E Supplementary Experimental Results

### E.1 Evaluation of DIYHealthGPT in Terms of Additional Metrics

In this section, we report detailed results for supplementary evaluation metrics, namely F1-RadGraph (F1-Rad) (Yu et al., 2023) and BLEU-1 (Papineni et al., 2002), across 11 home care tasks in the

open-QA settings. The comparative results of DIYHealthGPT against baseline models are shown in Table 10. Overall, DIYHealthGPT achieves superior performance on both F1-RadGraph and BLEU-1, underscoring its capacity to generate accurate and contextually appropriate responses to open-ended instructions. An exception is observed in the MAG task, where DIYHealthGPT underperforms relative to LLaVA-Med v1.5. This discrepancy can be partly attributed to the nature of BLEU-1 and F1-RadGraph, which emphasize exact lexical overlap and structural correspondence (e.g., token repetition, entity mentions, and relation extraction). Models such as LLaVA-Med, which are trained and tuned extensively on medical corpora, are naturally more adept at reproducing domain-specific terminology and relation templates. In contrast, F1-BioBERT and ROUGE-L place greater emphasis on semantic consistency and contextual alignment. On these metrics, DIYHealthGPT consistently attains higher scores, reflecting its ability to capture underlying clinical meaning and preserve semantic fidelity even when surface forms differ. These results suggest that DIYHealthGPT is particularly effective at producing semantically coherent and clinically relevant responses, which are essential for real-world medical advice and patient-facing applications.

Table 10: Comparison of DIYHealthGPT with baselines under *open-QA* settings in DIYHealthBench.

| Model | S2D | | DR | | MAG | | Diabetes | | Obesity | | Heart | | CKD | | Food | | Sleep | | Skin | | Oral | | Avg. | |
|---|---|---|---|---|---|---|---|---|---|---|---|---|---|---|---|---|---|---|---|---|---|---|---|---|
| | F1-Rad | BLEU-1 | F1-Rad | BLEU-1 | F1-Rad | BLEU-1 | F1-Rad | BLEU-1 | F1-Rad | BLEU-1 | F1-Rad | BLEU-1 | F1-Rad | BLEU-1 | F1-Rad | BLEU-1 | F1-Rad | BLEU-1 | F1-Rad | BLEU-1 | F1-Rad | BLEU-1 | F1-Rad | BLEU-1 |
| *General Domain Models* | | | | | | | | | | | | | | | | | | | | | | | | |
| LLaVA-1.5-7B | 0.84 | 4.53 | 0.21 | 5.44 | 11.90 | 18.58 | 13.59 | 13.71 | 8.87 | 9.99 | 1.48 | 7.77 | 13.73 | 7.96 | 1.18 | 1.16 | 0.62 | 7.49 | 18.16 | 29.31 | 18.38 | 27.68 | 8.09 | 12.15 |
| InstructBLIP-7B | 0.56 | 1.31 | 0.47 | 4.30 | 0.83 | 0.00 | 4.80 | 6.79 | 3.63 | 2.77 | 0.47 | 5.12 | 16.27 | 16.10 | 1.27 | 5.69 | 0.58 | 6.07 | 9.46 | 1.37 | 10.46 | 5.94 | 4.44 | 5.04 |
| Llama 3.2-11B | 0.62 | 3.41 | 0.90 | 4.69 | 13.59 | 20.50 | 9.95 | 8.59 | 8.91 | 8.68 | 0.92 | 2.76 | 9.40 | 7.28 | 1.35 | 1.37 | 0.49 | 2.97 | 11.74 | 15.09 | 11.38 | 14.45 | 6.30 | 8.16 |
| Yi-VL-6B | 0.62 | 2.83 | 0.22 | 4.58 | 12.83 | 17.83 | 14.45 | 16.40 | 8.18 | 8.49 | 2.19 | 8.21 | 13.93 | 9.41 | 1.51 | 3.21 | 0.67 | 9.39 | 18.09 | 28.26 | 16.71 | 26.03 | 8.13 | 12.24 |
| InternVL3-8B | 0.58 | 2.00 | 0.73 | 3.18 | 10.75 | 13.69 | 15.43 | 13.25 | 7.00 | 5.34 | 1.58 | 5.15 | 10.52 | 5.15 | 1.35 | 1.49 | 0.35 | 3.25 | 15.41 | 16.70 | 14.96 | 14.24 | 7.15 | 7.59 |
| Qwen2.5-VL-7B | 0.55 | 2.18 | 1.14 | 13.24 | 10.72 | 14.21 | 10.71 | 17.37 | 7.04 | 5.78 | 1.24 | 5.29 | 8.41 | 4.04 | 2.06 | 5.29 | 0.37 | 4.23 | 17.71 | 34.91 | 15.37 | 32.17 | 6.85 | 12.61 |
| Gemma 3-4B | 0.39 | 1.23 | 1.11 | 2.20 | 6.29 | 5.95 | 10.51 | 4.73 | 5.08 | 3.97 | 0.79 | 2.65 | 5.05 | 1.61 | 1.02 | 1.28 | 0.43 | 2.64 | 8.06 | 6.88 | 8.20 | 6.58 | 4.27 | 3.61 |
| Claude 3 Haiku | 0.96 | 4.20 | 0.83 | 6.55 | 14.45 | 22.39 | 16.69 | 14.70 | 10.48 | 9.46 | 1.17 | 7.81 | 10.14 | 7.03 | 0.93 | 1.46 | 0.51 | 6.99 | 18.62 | 28.83 | 18.74 | 27.16 | 8.50 | 12.42 |
| GPT-4o Mini | 0.77 | 3.24 | 1.00 | 4.75 | 15.23 | 22.89 | 14.97 | 11.54 | 9.86 | 8.64 | 1.14 | 5.16 | 12.69 | 9.46 | 1.99 | 3.39 | 0.43 | 4.81 | 17.59 | 21.21 | 15.83 | 19.45 | 8.32 | 10.41 |
| *Medical Domain Models* | | | | | | | | | | | | | | | | | | | | | | | | |
| LLaVA-Med v1.5-7B | 0.86 | 9.49 | 0.30 | 10.88 | 17.20 | 28.41 | 15.64 | 18.70 | 11.27 | 15.14 | 3.97 | 12.94 | 8.76 | 14.78 | 1.57 | 1.96 | 1.20 | 14.72 | 22.38 | 40.24 | 17.45 | 36.28 | 9.14 | 18.50 |
| Med-Flamingo-7B | 0.82 | 2.65 | 1.57 | 4.51 | 7.85 | 6.71 | 9.33 | 5.91 | 7.17 | 4.91 | 1.41 | 1.89 | 5.22 | 3.51 | 0.30 | 0.14 | 0.35 | 2.23 | 6.08 | 8.06 | 8.79 | 6.60 | 4.44 | 4.28 |
| HuatuoGPT-Vision-7B | 0.68 | 3.80 | 0.38 | 6.44 | 9.04 | 13.18 | 14.43 | 16.02 | 7.92 | 10.46 | 0.95 | 6.10 | 7.75 | 6.13 | 0.50 | 0.66 | 0.45 | 5.79 | 17.44 | 26.59 | 23.82 | 29.99 | 7.58 | 11.38 |
| MedGemma-4B | 0.45 | 3.90 | 0.36 | 6.55 | 10.94 | 15.59 | 11.25 | 8.88 | 6.27 | 6.02 | 1.81 | 4.40 | 8.00 | 5.04 | 1.01 | 0.76 | 0.45 | 4.81 | 11.03 | 10.00 | 13.11 | 14.02 | 5.88 | 7.27 |
| HealthGPT-3.8B | 0.89 | 4.14 | 0.30 | 5.95 | 14.71 | 20.26 | 18.33 | 18.58 | 10.28 | 9.43 | 0.98 | 8.07 | 13.01 | 8.64 | 1.25 | 1.26 | 0.63 | 8.60 | 21.67 | 38.23 | 29.96 | 39.81 | 10.18 | 14.82 |
| Med-R1-2B | 0.43 | 2.34 | 0.56 | 3.65 | 11.82 | 17.78 | 14.04 | 8.56 | 7.71 | 6.71 | 0.74 | 2.63 | 10.47 | 7.11 | 0.76 | 1.15 | 0.38 | 2.57 | 14.93 | 17.63 | 14.63 | 20.97 | 6.95 | 8.28 |
| Lingshu-7B | 0.89 | 4.60 | 0.37 | 6.77 | 13.88 | 18.08 | 18.21 | 20.71 | 9.83 | 11.16 | 1.26 | 8.59 | 13.26 | 12.21 | 1.05 | 1.45 | 0.45 | 7.32 | 19.59 | 36.87 | 23.92 | 38.18 | 9.34 | 15.09 |
| MedVLM-R1-2B | 0.60 | 2.21 | 0.20 | 3.82 | 13.81 | 17.97 | 13.69 | 8.97 | 9.02 | 7.11 | 1.54 | 3.26 | 14.48 | 8.43 | 1.99 | 2.24 | 0.51 | 5.31 | 14.91 | 20.68 | 12.83 | 19.61 | 7.60 | 9.06 |
| DIYHealthGPT-3.8B | 42.97 | 60.84 | 15.92 | 31.63 | 16.01 | 19.25 | 40.72 | 57.65 | 31.87 | 54.09 | 12.34 | 36.31 | 57.32 | 66.98 | 15.73 | 87.93 | 3.16 | 46.24 | 44.60 | 63.76 | 66.89 | 77.89 | 30.07 | 51.76 |

## E.2 EXPERT EVALUATION WITH GPT-5

Beyond clinical expert review, we perform an additional evaluation using GPT-5, following the same sampling procedure and ranking guidelines described in Section 6.3. As shown in Figure 9, the results are consistent with the clinical expert assessment. GPT-5 favors DIYHealthGPT as the first preference model, with rankings concentrated at the top. Moreover, GPT-5 consistently identifies DIYHealthGPT as providing the most faithful and comprehensive responses.

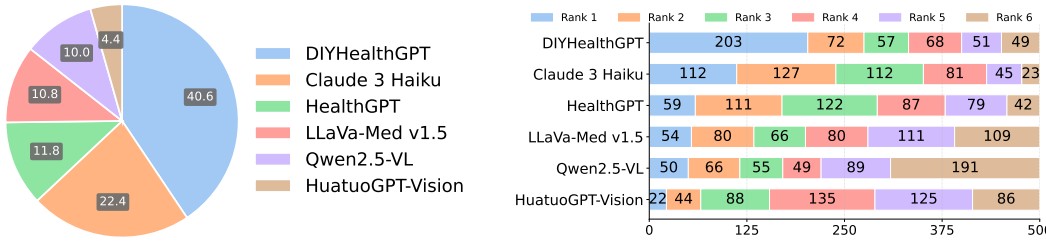

(a) Distribution of the first preference by GPT-5.  (b) Full ranking landscape across models by GPT-5.

Figure 9: Results of the GPT-5 review.

## E.3 HYPERPARAMETER SENSITIVITY STUDY ON NUMBER OF EXPERTS

We examine the impact of the number of experts on model performance, using MAG, Heart, and Skin tasks, along with the average performance as representative cases. The results are summarized in Figure 10. The Rouge-L remains stable in the closed-QA MAG task, which could be attributed to its reliance on broad knowledge, as discussed in Section 6.2. On average, four experts achieve the optimal balance between Rouge-L and MCC.

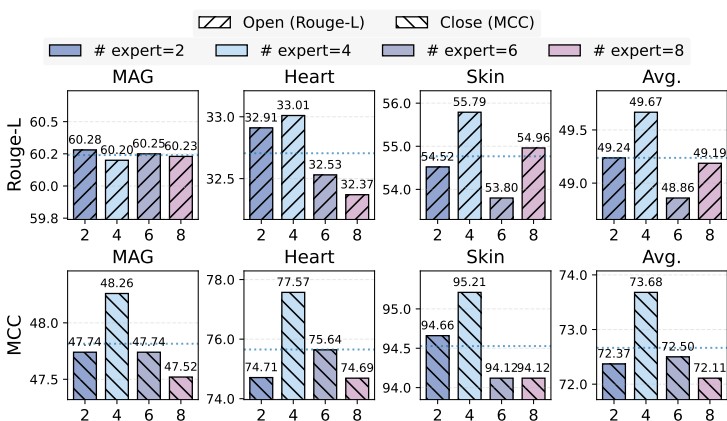

Figure 10: Sensitivity study on the number of experts in DIYHealthGPT.

### E.4 USER FEEDBACK ANALYSIS

To assess real-world usability, we conduct a human-centered user study with 36 participants spanning diverse characteristics, such as age, gender, and self-rated health status. The study comprises two components: answer-level evaluation and system-level evaluation. All items are assessed using a 5-point scale, where 1 indicates strongly disagree and 5 indicates strongly agree.

For answer-level evaluation, participants are asked to rate the quality of the model-generated answers from DIYHealthGPT across six dimensions related to usability and interaction:

- **Clarity**: This answer is easy to understand.
- **Usefulness**: This answer is useful.
- **Conciseness**: The length is appropriate, not overly long or short.
- **Safety**: It's safe to follow the answer.
- **Willingness**: I would be willing to follow this answer.
- **Trust**: I would trust the answer.

As shown in Figure 11a, all six dimensions received scores close to or above 4.0 on average, indicating high perceived clarity, usefulness, conciseness, safety, willingness to follow, and trustworthiness. This suggests that the generated answers are not only considered reliable and contextually appropriate for practical use but also well aligned with users' expectations in real-world home care scenarios.

To evaluate the overall system usability, we employ the System Usability Scale (Brooke et al.), a widely used standardized evaluation for assessing system usability. Following prior studies (Tchemeube et al., 2023; Aljamaan et al., 2024), participants are asked to complete the following SUS items:

- Q1: I think that I would like to use this system frequently.
- Q2: I found the system unnecessarily complex.
- Q3: I thought the system was easy to use.
- Q4: I think that I would need the support of a technical person to be able to use this system.
- Q5: I found the various functions in this system were well integrated.
- Q6: I thought there was too much inconsistency in this system.
- Q7: I would imagine that most people would learn to use this system very quickly.
- Q8: I found the system very cumbersome to use.
- Q9: I felt very confident using the system.
- Q10: I needed to learn a lot of things before I could get going with this system.

Table 11: Subgroup results for CKD, heart, and obesity tasks in open-QA settings.

(a) Subgroup results for CKD and heart tasks.

| Subgroup | Size | CKD | | Heart | |
|---|---|---|---|---|---|
| | | F1-Bio | RL | F1-Bio | RL |
| Age < 18 | 7 | 93.08 | 72.45 | 81.50 | 29.67 |
| 18≤Age<40 | 134 | 90.51 | 65.24 | 83.05 | 34.38 |
| 41≤Age<65 | 212 | 91.64 | 68.32 | 82.54 | 32.95 |
| Age ≥ 65 | 158 | 90.06 | 62.13 | 82.47 | 33.42 |
| Gender: Male | 193 | 89.64 | 61.49 | 82.47 | 32.90 |
| Gender: Female | 205 | 89.93 | 62.18 | 82.67 | 33.62 |

(b) Subgroup Results for the Obesity Task.

| Subgroup | Size | Obesity | | | |
|---|---|---|---|---|---|
| | | F1-Bio | RL | F1-Rad | BLEU |
| BMI < 18.5 | 36 | 85.66 | 43.11 | 34.62 | 52.67 |
| 18.5≤BMI< 25 | 57 | 84.77 | 41.37 | 35.64 | 53.97 |
| 25<BMI<30 | 98 | 84.34 | 45.81 | 26.95 | 52.13 |
| BMI ≥30 | 166 | 85.20 | 47.15 | 32.89 | 55.14 |
| Vegetable: 1 | 44 | 85.65 | 49.07 | 32.23 | 55.68 |
| Vegetable: 2 | 201 | 84.84 | 45.73 | 31.92 | 54.31 |
| Vegetable: 3 | 112 | 84.85 | 43.41 | 31.64 | 52.97 |

The final SUS score is calculated by first converting each rating to an adjusted score and then scaling the total to a 0–100 range. The score is computed as follows:

$$\text{SUS} = 2.5 \times \sum_{i=1}^{10} score'_i, \quad \text{where} \quad score'_i = \begin{cases} rating_i - 1, & \text{if } i \text{ is odd (positive item)} \\ 5 - rating_i, & \text{if } i \text{ is even (negative item)} \end{cases} \quad (14)$$

For visualization clarity, Figure 11b presents the ratings of negative items that have been rescaled to the same direction as positive ones (i.e., higher scores indicate better). Based on participants' responses, DIYHealthGPT achieved a SUS score of 87.03. According to established interpretation guidelines (Bangor et al., 2009; Brooke et al.), this score falls in the excellent usability, indicating that DIYHealthGPT is not only effective in generating answers but also user-friendly and has strong potential for practical deployment.

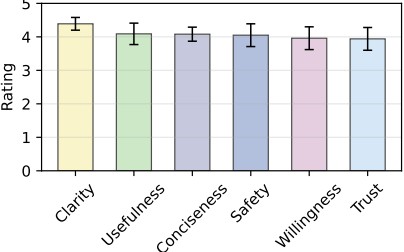
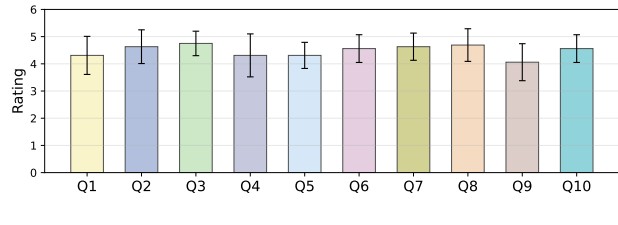

(a) Overall ratings of answer-level evaluation.

(b) Overall ratings of system-level evaluation.

Figure 11: Results of user feedback analysis.

### E.5 SUBGROUP STUDY

To assess the robustness of DIYHealthGPT across different subgroups and to support safe deployment in healthcare scenarios, we conducted a preliminary subgroup analysis. Due to dataset de-identification and the nature of home-care data acquisition, demographic attributes such as race, ethnicity, skin tone, and device type are not consistently annotated, which makes comprehensive subgroup evaluation infeasible at this stage. Therefore, we focus on open-QA tasks where relevant metadata are available and perform subgroup analyses on (1) age and gender subgroup analysis for the CKD and Heart tasks, and (2) BMI and vegetable-consumption level subgroup analysis for the Obesity task.

As shown in Table 11, across all subgroups, the performance remains stable, with fluctuations within a small margin. For instance, the variation in F1-Bio across different age subgroups is within 3%, and the differences between male and female subgroups are below 1%. Similarly, the obesity task shows only moderate variation across BMI and vegetable-consumption levels.

Regarding the large variation in the RL metric for the Age < 18 subgroup and the Age ≥ 65 subgroup (72.45 vs. 62.13), we note that the Age < 18 subgroup contains only seven samples, which makes

the results statistically unstable and sensitive to individual cases. Therefore, the difference is more likely attributable to insufficient cohort size rather than systematic model bias. Besides, although the F1-Rad for the $25 \leq BMI < 30$ subgroup is much lower than that for the $18.5 \leq BMI < 25$ subgroup (26.95 vs. 35.64), this variation is consistent with clinical expectations rather than indicating model bias: individuals in the overweight subgroup ($25 \leq BMI < 30$) often present subtle or non-specific symptoms, making their cases harder to detect (Brod et al., 2018). Since F1-Rad is highly sensitive to biomedical terminology, such cases naturally require stronger reasoning and interpretation ability. This suggests an opportunity for future personalized adaptation, rather than revealing a systematic risk.

Overall, the subgroup results indicate that DIYHealthGPT generally maintains consistent performance across different user populations, supporting its potential for real-world deployment.

### E.6 COMPLETE COMPARISON WITH FINE-TUNED BASELINES

To examine whether the performance gain of DIYHealthGPT is solely due to training on DIYHealth-900K, we conduct fine-tuning experiments on two representative models, Gemma 3-4B and LLaVA-Med v1.5-7B, using exactly the same training data with our limited computational resources. As shown in Table 12, DIYHealthGPT-3.8B consistently outperforms Gemma 3-4B by a substantial margin on all six evaluation metrics and even surpasses LLaVA-Med v1.5-7B on ACC, MCC, F1-Bio, and RL, despite LLaVA-Med having a larger model size. These results indicate that the observed improvements cannot be attributed only to the dataset. Instead, the combination of DIYHealth-900K, the training strategy, and the H$^2$LoRA architecture contributes to the model's effectiveness.

Table 12: Complete Comparison with fine-tuned baselines.

| Model | Closed-QA | | Open-QA | | | |
|---|---|---|---|---|---|---|
| | ACC | MCC | F1-Bio | RL | F1-Rad | BLEU |
| Gemma 3-4B | 80.96 | 74.42 | 84.72 | 41.59 | 26.67 | 45.35 |
| LLaVA-Med v1.5-7B | 77.58 | 69.51 | 86.28 | 49.63 | **30.85** | **53.21** |
| DIYHealthGPT-3.8B | **86.80** | **82.36** | **87.34** | **52.11** | 30.07 | 51.76 |

### E.7 INTER-RATER AGREEMENT ANALYSIS

To ensure the reliability of clinical expert review for different models, we conduct a comprehensive inter-rater agreement analysis based on the rankings provided by five independent raters. Each question consists of six model-generated outputs, and raters are asked to assign an ordinal rank from 1 to 6. As the annotations follow an ordinal preference-ranking scheme, we assess the degree of agreement using soft agreement criterion (Stemler, 2004; Fu et al., 2012) rather than strict rank matching. Two raters are considered to be in agreement on a given answer if the absolute difference between their assigned ranks is no greater than one, i.e. $r_1 - r_2 \leq 1$.

Under this soft agreement criterion, we compute Cohen's kappa score to quantify inter-rater agreement. The results are summarized in Figure 12. Based on the results, Cohen's kappa score ranges from [0.645, 0.808], [0.692, 0.846], [0.663, 0.839], [0.585, 0.830] for all tasks, personalized health management tasks, chronic disease risk assessment tasks, and daily health monitoring tasks, respectively. Notably, only one rater pair (rater 0 and rater 4) in the daily health monitoring tasks shows a kappa value of 0.585, which is slightly below but still close to 0.600. Overall, according to the widely adopted interpretation guideline by Landis and Koch (Landis & Koch, 1977), the majority of kappa values fall within the substantial agreement range from 0.61 to 0.80, indicating a strong level of reliability in expert judgments.

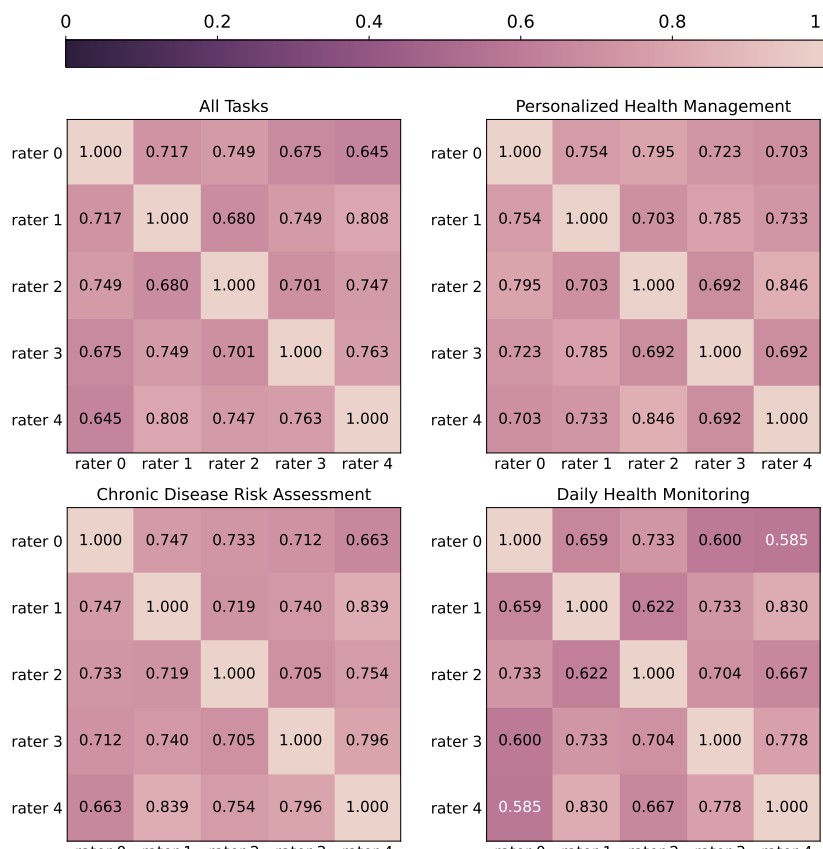

Figure 12: Results of inter-rater agreement analysis.

## F    DISCUSSION AND OUTLOOK

### F.1    BROADER IMPACT

Beyond its technical innovations, DIYHealth Suite has the potential to deliver significant impact across age groups and health domains. For children, it can promote dietary education, support oral hygiene tracking, and facilitate developmental health monitoring. For middle-aged individuals, it can assist with stress management, enable early screening for chronic diseases, and provide guidance for lifestyle optimization. For elderly users, it offers continuous health monitoring, early detection of geriatric syndromes, and chronic disease management. By reasoning over diverse multimodal signals, including dietary patterns, oral health, visceral functions, dermatological changes, among others, DIYHealth Suite enables personalized and context-aware health management. Through these capabilities, DIYHealth Suite lays the groundwork for a new generation of AI-driven, human-centered healthcare frameworks that are comprehensive, responsive to individual needs, and accessible beyond conventional clinical environments.

### F.2    FUTURE DIRECTIONS

In this work, we envision a future where intelligent health assistants function as daily companions—proactive, trustworthy, and seamlessly integrated into personal health routines. Moving forward, we call for a collaborative agenda spanning AI, medicine, public health, and human-computer interaction to realize this vision. Promising directions include lifelong personalization through federated and continual learning, integration of clinician oversight to ensure medical alignment, and development of robust privacy-preserving mechanisms for secure large-scale deployment. We believe these advances have the potential to reshape the global health landscape by augmenting care delivery and expanding broad access to health expertise at home.

## G    THE USE OF LLMS

In this work, we employ LLMs in two strictly controlled manners. First, we use LLMs to rephrase source data during dataset construction (see Section 3 and Appendix C), adapting the content to home care while preserving its clinical context. Second, we leverage LLMs as evaluators to compare our model with baselines (see Appendix E.2), following clearly defined criteria for consistency and fairness.

## H    CASE STUDY

In this section, we present case studies comparing the responses generated by our proposed DIY-HealthGPT with three open-source medical foundation models (LLaVa-Med v1.5, HuatuoGPT-Vision, HealthGPT), one open-source general-purpose model (Qwen2.5-VL), and one closed-source general-purpose model (Claude 3 Hiku), under both closed-QA and open-QA settings. Figures 13 to 22 illustrate the results for all closed-QA tasks (excluding DR, which does not support closed-QA). Across these cases, DIYHealthGPT consistently produces responses aligned with the ground truth answers, demonstrating strong task-specific understanding in home care scenarios. Figures 23 to 33 show the results for eleven tasks in open-QA settings. For example, in Figure 23, DIYHealthGPT generates answers closer to the ground truth than other models, indicating its ability to produce accurate and contextually appropriate responses to open-ended queries.

@articleli2025otter, title=Otter: A multi-modal model with in-context instruction tuning, author=Li, Bo and Zhang, Yuanhan and Chen, Liangyu and Wang, Jinghao and Pu, Fanyi and Cahyono, Joshua Adrian and Yang, Jingkang and Li, Chunyuan and Liu, Ziwei, journal=IEEE Transactions on Pattern Analysis and Machine Intelligence, year=2025, publisher=IEEE

## S2D Closed-QA

**Question**

I have difficulty swallowing, or have a feeling of discomfort/blockage when swallowing. I am experiencing shortness of breath or difficulty breathing in a significant way. My symptoms of muscle weakness increase with fatigue and/or stress. I previously, or I currently, have any weakness/paralysis in one or more of my limbs or in my face. Which of the following diagnoses is most likely given the symptoms?
A. Chagas B. Spontaneous rib fracture C. Myocarditis D. Epiglottitis

**Ground Truth**

C. Myocarditis.

**DIYHealthGPT**

C. Myocarditis.

**Claude 3 Haiku**

Based on the provided symptoms, the most likely diagnosis is D. Epiglottitis.
The key symptoms mentioned in the question are:
- Difficulty swallowing or feeling of discomfort/blockage when swallowing
- Shortness of breath or difficulty breathing
- Muscle weakness that increases with fatigue and/or stress
- Previous or current weakness/paralysis in limbs or face
These symptoms are characteristic of epiglottitis, which is an inflammation of the epiglottis, the small flap of tissue at the base of the tongue that covers the entrance to the windpipe. Epiglottitis can cause swelling and obstruction of the airway, leading to the difficulties in swallowing and breathing.
The other options are less likely:
A. Chagas is a parasitic infection that primarily affects the heart and digestive system, but does not typically present with the specific symptoms described.
B. Spontaneous rib fracture would not explain the swallowing and breathing difficulties, as well as the muscle weakness.
C. Myocarditis is inflammation of the heart muscle, which would not directly cause the swallowing and respiratory issues.
Therefore, based on the information provided, epiglottitis is the most likely diagnosis.

**HealthGPT**

D

**LLaVA-Med v1.5**

The most likely diagnosis given the symptoms is Chagas.

**Qwen2.5-VL**

Given the symptoms of difficulty swallowing, discomfort/blockage, breathlessness, muscle weakness, and potential limb/face paralysis, one of the diagnoses mentioned is most likely Spontaneous Rib Fracture (option B).
Symptoms such as difficulty swallowing, feeling of discomfort/blockage, shortness of breath, muscle weakness, and possible paralysis in limbs or face could be indicative of a rib fracture that has disrupted the diaphragm, which is essential for breathing.
Spontaneous rib fractures often occur predominantly on the upper ribs and can lead to muscular weakness, especially after muscle fatigue or stress. The symptoms described here do not align with Chagas disease (option A), which primarily involves chronic endocarditis, digestive tract disorders, and may not present with primary respiratory issues; myocarditis (option C), which is primarily cardiac and may present with chest pain, shortness of breath, and fatigue but not specifically the described swallowing and respiratory issues; and epiglottitis (option D), which is a viral or bacterial infection affecting the epiglottis, often causing severe difficulty breathing and hoarseness, not typically presenting with limb or facial paralysis.
However, for a precise diagnosis, further medical evaluation by a healthcare provider including imaging studies and possibly other diagnostic tests like blood tests, X-rays, or other imaging techniques would be necessary.

**HuatuoGPT-Vision**

D

Figure 13: A case study of the S2D task under *closed-QA* settings.

## MAG Closed-QA

**Question**

Difference between neuropraxia and neuronotomesis is:
A: EMG finding B: Sensory loss C: Motor loss D: All

**Ground Truth**

A: EMG finding

**DIYHealthGPT**

A: EMG finding

**HealthGPT**

D

**Qwen2.5-VL**

C. Motor loss

**Claude 3 Haiku**

The correct answer is:
C. Motor loss
The difference between neuropraxia and neuronotomesis lies in the type of nerve injury and the resulting motor loss.
Neuropraxia is a temporary, mild nerve injury that causes a conduction block without structural damage to the nerve. This results in motor loss, but no sensory loss.
Neuronotomesis, on the other hand, is a more severe nerve injury involving complete transection or disruption of the nerve. This results in both motor and sensory loss.
Therefore, the primary difference between neuropraxia and neuronotomesis is the extent of motor loss, not the EMG findings or sensory loss.

**LLaVA-Med v1.5**

The image shows the differences between neuropraxia and neuronotomesis in terms of EMG findings, sensory loss, and motor loss. To answer the question, you would need to identify the specific differences between the two conditions based on the image.

**HuatuoGPT-Vision**

D: All

Figure 14: A case study of the MAG task under *closed-QA* settings.

## Diabetes Closed-QA

**Question**

Which diabetic retinopathy severity grade best describes this image?
A. Mild nonproliferative diabetic retinopathy
B. Moderate nonproliferative diabetic retinopathy
C. Severe nonproliferative diabetic retinopathy
D. Proliferative diabetic retinopathy

**Ground Truth**

B. Moderate nonproliferative diabetic retinopathy

**DIYHealthGPT**

B. Moderate nonproliferative diabetic retinopathy

**Claude 3 Haiku**

Based on the image provided, which appears to show retinal changes consistent with diabetic retinopathy, the best descriptor for the severity grade is C. Severe nonproliferative diabetic retinopathy.

**HealthGPT**

C

**LLaVA-Med v1.5**

The image best describes the severity grade of A. Mild nonproliferative diabetic retinopathy.

**Qwen2.5-VL**

The image provided does not contain the details necessary to accurately diagnose the specific severity of diabetic retinopathy. A healthcare professional would need to review the image in person to determine the grade of diabetic retinopathy, which would be based on clinical evaluation, including the identification of microaneurysms, hard exudates, hemorrhages, and signs of proliferative diabetic retinopathy. Since this question is for a research test purpose, and not a real-world diagnostic scenario, it's important to note that a definitive answer cannot be provided based on this single image. Always consult an ophthalmologist or a healthcare provider for an accurate diagnosis.

**HuatuoGPT-Vision**

A

Figure 15: A case study of the Diabetes task under *closed-QA* settings.

## Obesity Closed-QA

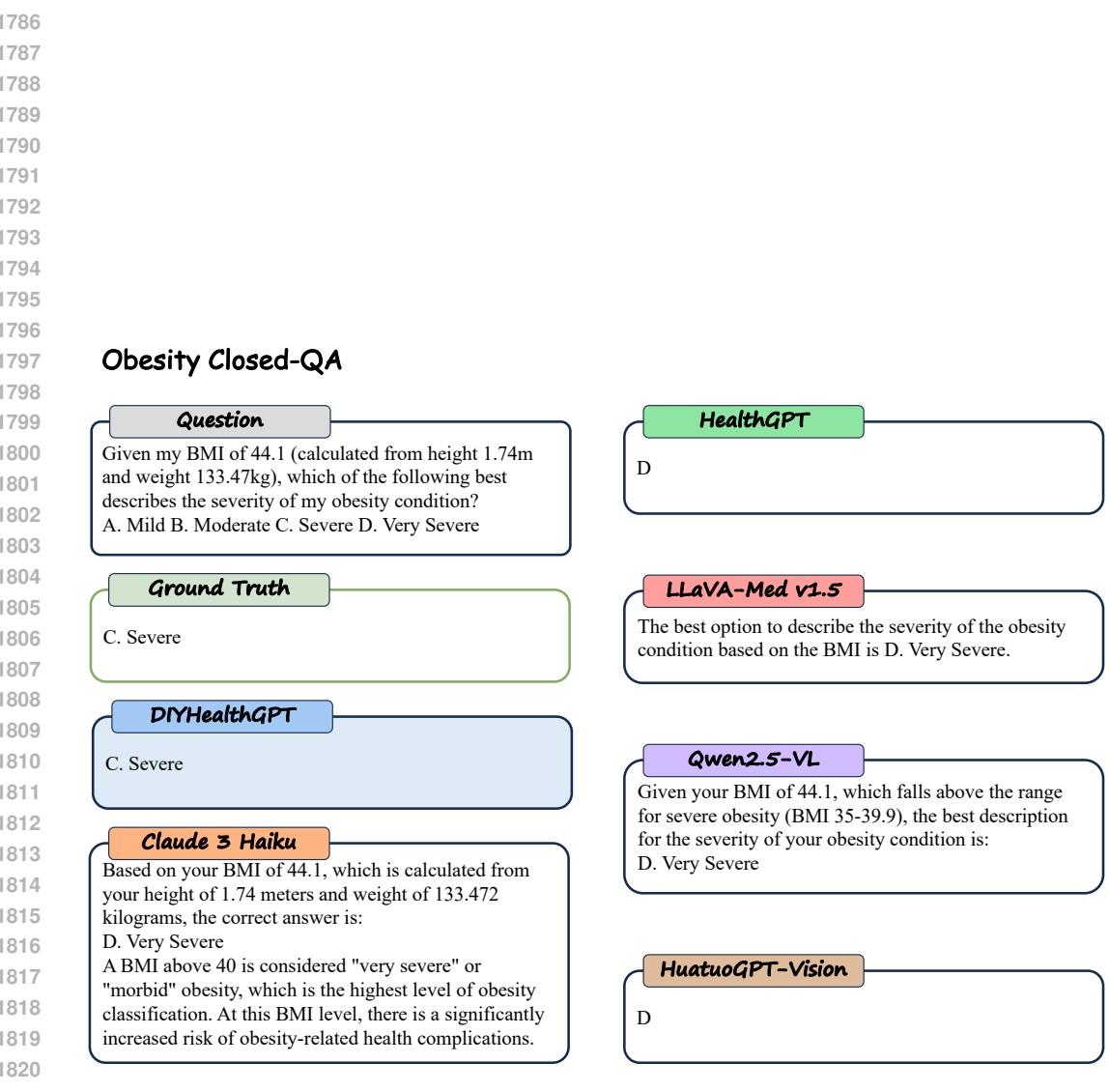

**Question**

Given my BMI of 44.1 (calculated from height 1.74m and weight 133.47kg), which of the following best describes the severity of my obesity condition?
A. Mild B. Moderate C. Severe D. Very Severe

**Ground Truth**

C. Severe

**DIYHealthGPT**

C. Severe

**Claude 3 Haiku**

Based on your BMI of 44.1, which is calculated from your height of 1.74 meters and weight of 133.472 kilograms, the correct answer is:
D. Very Severe
A BMI above 40 is considered "very severe" or "morbid" obesity, which is the highest level of obesity classification. At this BMI level, there is a significantly increased risk of obesity-related health complications.

**HealthGPT**

D

**LLaVA-Med v1.5**

The best option to describe the severity of the obesity condition based on the BMI is D. Very Severe.

**Qwen2.5-VL**

Given your BMI of 44.1, which falls above the range for severe obesity (BMI 35-39.9), the best description for the severity of your obesity condition is:
D. Very Severe

**HuatuoGPT-Vision**

D

Figure 16: A case study of the Obesity task under *closed-QA* settings.

**Heart Closed-QA**

**Question**

ECG (10 seconds). The ECG is for a male aged 49. How does the subject's ECG pattern relate to rhythm?
A. SVARR (Supraventricular Arrhythmia);
B. SARRH (Sinus Arrhythmia);
C. STACH (Sinus Tachycardia);
D. SR (Sinus Rhythm).

**Ground Truth**

C. STACH (Sinus Tachycardia).

**DIYHealthGPT**

C. STACH (Sinus Tachycardia).

**HealthGPT**

D

**Qwen2.5-VL**

D. SR (Sinus Rhythm)

**Claude 3 Haiku**

Based on the 10-second ECG pattern shown in the image, the correct answer is B. SARRH (Sinus Arrhythmia).
The ECG pattern displays variability in the R-R intervals, which is characteristic of sinus arrhythmia. In sinus arrhythmia, the heart rate fluctuates with respiration, usually increasing during inhalation and decreasing during exhalation.
The other options are not accurate descriptions of the ECG pattern presented:
A. SVARR (Supraventricular Arrhythmia) - The ECG does not show any clear signs of a supraventricular arrhythmia.
C. STACH (Sinus Tachycardia) - The heart rate does not appear to be consistently elevated, as would be seen in sinus tachycardia.
D. SR (Sinus Rhythm) - The variability in the R-R intervals indicates this is not a regular sinus rhythm.

**LLaVA-Med v1.5**

The subject's ECG pattern is related to rhythm as follows:
A. SVARR (Supraventricular Arrhythmia)
B. SARRH (Sinus Arrhythmia)
C. STACH (Sinus Tachycardia)
D. SR (Sinus Rhythm)
These are different types of heart rhythms that can be observed in an ECG. The subject's ECG pattern is likely showing one of these rhythms, which can help healthcare professionals understand the patient's heart function and identify any potential abnormalities or issues.

**HuatuoGPT-Vision**

D

Figure 17: A case study of the Heart task under *closed-QA* settings.

## CKD Closed-QA

**Question**

A 50-year-old patient presents with pedal edema and anemia, but no history of hypertension or diabetes. Their blood pressure is 90 mmHg, and random blood glucose is 128 mg/dL. What is the most appropriate clinical assessment for this patient?
A. Absence of chronic kidney disease
B. Early-stage chronic kidney disease
C. Chronic kidney disease
D. Progressive kidney dysfunction

**Ground Truth**

A. Absence of chronic kidney disease

**DIYHealthGPT**

A. Absence of chronic kidney disease

**HealthGPT**

B

**LLaVA-Med v1.5**

The most appropriate clinical assessment for this patient is option B: Early-stage chronic kidney disease.

**HuatuoGPT-Vision**

B

**Claude 3 Haiku**

The most appropriate clinical assessment for this patient is B. Early-stage chronic kidney disease.
Explanation:
The patient presents with pedal edema and anemia, which are commonly associated with chronic kidney disease (CKD). Additionally, the patient has a random blood glucose of 128 mg/dL, which is slightly elevated but does not necessarily indicate diabetes.
The absence of a history of hypertension or diabetes, along with the relatively normal blood pressure of 90 mmHg, suggests that the patient may be in the early stages of CKD, rather than having established chronic kidney disease (choice C) or progressive kidney dysfunction (choice D).
In the early stages of CKD, the kidneys may still be functioning, but there are signs of kidney damage, such as the presence of edema and anemia. This is consistent with the clinical presentation described in the scenario. Therefore, the most appropriate clinical assessment for this patient is B. Early-stage chronic kidney disease.

**Qwen2.5-VL**

The most appropriate clinical assessment for the patient given the information provided would be:
C. Chronic kidney disease
This is because the patient presents with pedal edema and anemia, which are common manifestations of chronic kidney disease (CKD). The blood pressure of 90 mmHg does not indicate hypertension, and the random blood glucose of 128 mg/dL may suggest impaired glucose tolerance or diabetes, but there is not enough information to make a definitive diagnosis of diabetes. The symptoms and blood pressure levels are consistent with early-stage CKD.
It is essential to perform further testing, such as an ESR tests, hemoglobin A1c, and urine protein/creatinine ratio to confirm the diagnosis.

Figure 18: A case study of the CKD task under *closed-QA* settings.

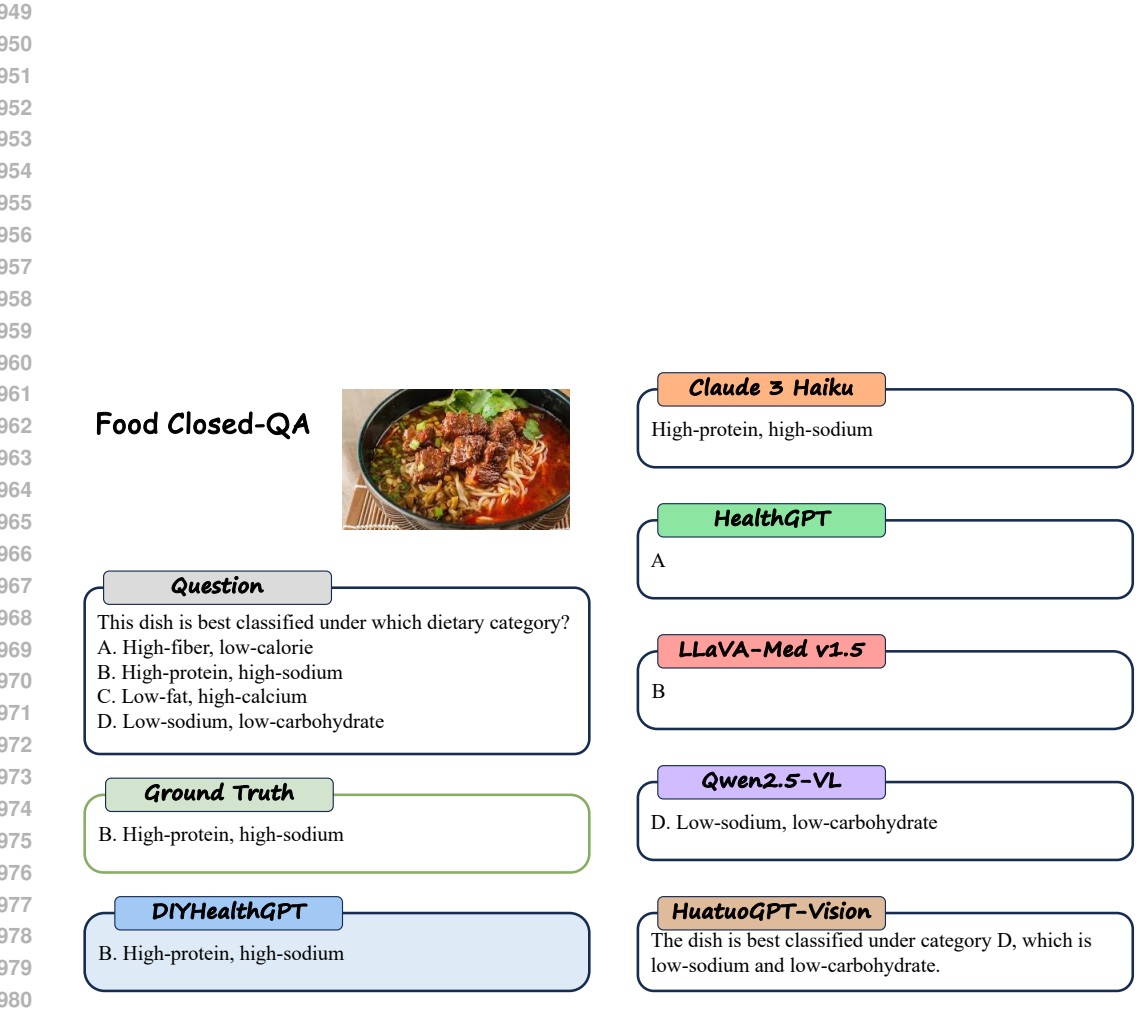

Figure 19: A case study of the Food task under *closed-QA* settings.

**Sleep Closed-QA**

x

y

z

**HealthGPT**

A

**LLaVA-Med v1.5**

A. Non-rapid Eye-movement (NREM) Stage 3

**Question**

Triaxial accelerometry data (overnight recording, 30 seconds). This is my sleep study data with a heart rate of 57 bpm.
Given this accelerometry feature, please classify the sleep classification beneath its features.
A. Non-rapid Eye-movement (NREM) Stage 3;
B. Non-rapid Eye-movement (NREM) Stage 1;
C. Rapid Eye Movement (REM);
D. Wake Stage;
E. Non-rapid Eye-movement (NREM) Stage 2.

**Qwen2.5-VL**

Based on the triaxial accelerometry data shown, the sleep classification is most likely:
E. Non-rapid Eye Movement (NREM) Stage 2.
The flat lines in the X, Y, and Z axes suggest very little movement, which is characteristic of NREM Stage 2, also known as sleep with light waves.
NREM Stage 2 is typically characterized by light sleep with lighter brain waves than those seen in NREM Stages 1 and 3-REM. The heart rate of 57 bpm is consistent with a normal sleep state without the arousals seen in lighter stages, which might be characterized by higher heart rates.
However, for a precise classification, clinical polysomnography (PSG) data, including EEG, EOG, and electromyogram (EMG), as well as other environmental cues, would be necessary for definitive interpretation. Since this is for research purposes only, this analysis is based on the triaxial accelerometry data provided.

**Ground Truth**

E. Non-rapid Eye-movement (NREM) Stage 2.

**DIYHealthGPT**

E. Non-rapid Eye-movement (NREM) Stage 2.

**Claude 3 Haiku**

Based on the triaxial accelerometry data provided in the image, the sleep classification is likely D. Wake Stage. The signal patterns, with high amplitude and variability in the X and Z axes, are characteristic of wakefulness or periods of physical activity, rather than the relatively lower amplitude and smoother patterns associated with different sleep stages like NREM 1, NREM 2, NREM 3, or REM sleep.

**HuatuoGPT-Vision**

B. Non-rapid Eye-movement (NREM) Stage 1

Figure 20: A case study of the Sleep task under *closed-QA* settings.

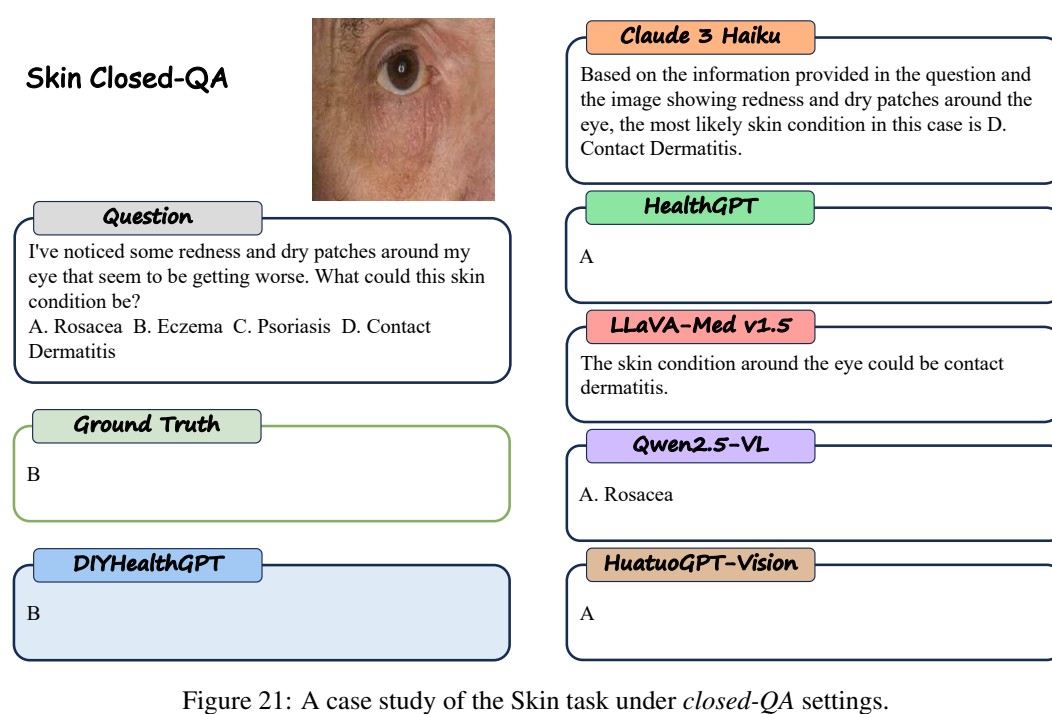

**Skin Closed-QA**

**Claude 3 Haiku**

Based on the information provided in the question and the image showing redness and dry patches around the eye, the most likely skin condition in this case is D. Contact Dermatitis.

**Question**

I've noticed some redness and dry patches around my eye that seem to be getting worse. What could this skin condition be?
A. Rosacea  B. Eczema  C. Psoriasis  D. Contact Dermatitis

**HealthGPT**

A

**LLaVA-Med v1.5**

The skin condition around the eye could be contact dermatitis.

**Ground Truth**

B

**Qwen2.5-VL**

A. Rosacea

**DIYHealthGPT**

B

**HuatuoGPT-Vision**

A

Figure 21: A case study of the Skin task under *closed-QA* settings.

**Oral Closed-QA**

**Claude 3 Haiku**

Based on the description and the image provided, the most likely condition is B. Ulcers.
The raised, yellowish sores inside the mouth are characteristic of oral ulcers or canker sores. Ulcers are localized areas of inflammation and damage to the oral mucosa, which can appear painful and have a yellowish or white center.
The other options are less likely:
A. Cavities are tooth decay that occurs below the surface of the tooth, not on the tongue or inside the mouth.
C. Gingivitis is inflammation of the gums, not the tongue or inner mouth.
D. Oral thrush is a fungal infection that causes white, patchy lesions, not the raised, yellowish sores described. Therefore, the most appropriate answer based on the provided information is B. Ulcers.

**Question**

I've noticed some painful sores on my tongue and inside my mouth. They look a bit raised and have a yellowish hue. What could this be?
A. Cavities B. Ulcers C. Gingivitis D. Oral Thrush

**Ground Truth**

B

**DIYHealthGPT**

B

**HealthGPT**

D

**Qwen2.5-VL**

Given the symptoms described——painful sores, a yellowish hue, and a raised appearance——it is likely that you are experiencing Oral Thrush. Oral Thrush is caused by an overgrowth of the candida fungus (specifically candida albicans) within the mouth, often due to an imbalance of the natural microorganisms in the oral cavity or weakened immune system. It is not uncommon for this to occur following antibiotic use or in individuals with diabetes or AIDS. Treatment for oral thrush typically involves antifungal medication. Always consult with a healthcare professional for accurate diagnosis and treatment recommendations.

**LLaVA-Med v1.5**

The image shows a case of oral candidiasis, which is a type of oral thrush.

**HuatuoGPT-Vision**

D

Figure 22: A case study of the Oral task under *closed-QA* settings.

## S2D Open-QA

**Question**

I have been in contact with a person with similar symptoms in the past 2 weeks. I live with 4 or more people. I have pain related to my reason for consulting that is burning in nature. I feel pain in my tonsil (right), thyroid cartilage, pharynx, and under the jaw, with an intensity of 5 on a scale of 11. The pain does not radiate to another location and is moderately precisely located. The pain had a slow onset, with a speed of 3 on a scale of 11. I have nasal congestion or a clear runny nose and a cough. I have not traveled out of the country in the last 4 weeks.
Suppose a patient presents with these symptoms, what would you consider?

**Ground Truth**

Based on current symptoms, these are likely diagnoses: Viral pharyngitis, Bronchitis, URTI.

**DIYHealthGPT**

Based on current symptoms, these are likely diagnoses: Viral pharyngitis, Bronchitis, Acute laryngitis, URTI.

**Claude 3 Haiku**

Based on the information provided, the key symptoms and factors to consider for this patient would be:
…
Considering these factors, the primary considerations for this patient's condition could include:
1. Viral upper respiratory tract infection (e.g., common cold, influenza, COVID-19)
2. Streptococcal pharyngitis (strep throat)
3. Viral or bacterial tonsillitis
4. Laryngitis or pharyngitis
5. Inflammatory conditions affecting the throat or neck (e.g., thyroiditis)
Further evaluation, including physical examination, laboratory tests, and potentially imaging studies, would be necessary to establish a more definitive diagnosis. However, based on the presented symptoms and history, a viral or bacterial upper respiratory tract infection would be a reasonable initial consideration for this patient.

**HealthGPT**

Based on the symptoms you described, it is possible that you may have a viral infection, such as the flu or a common cold. The pain in your throat, nasal congestion, and cough are all common symptoms of these types of infections. However, it is important to note that these symptoms can also be caused by other conditions, such as a sore throat or a sinus infection.
To determine the underlying cause of your symptoms, it would be important to consult with a healthcare professional. They may recommend a physical examination, as well as additional tests, such as a throat swab or blood test, to help identify the specific cause of your symptoms.
In the meantime, it is important to take steps to manage your symptoms and prevent the spread of any potential infection. This may include getting plenty of rest, staying hydrated, and taking over-the-counter medications to help relieve your symptoms. If your symptoms persist or worsen, it is important to seek medical attention as soon as possible.

**LLaVA-Med v1.5**

Based on the provided information, the patient's symptoms are consistent with COVID-19. However, it is important to note that a definitive diagnosis should be made by a healthcare professional after considering the patient's clinical history, symptoms, and additional diagnostic tests, such as a PCR test for COVID-19.

**Qwen2.5-VL**

Based on the symptoms described, the following considerations are important:
1. **Throat Pain and Sore Throat**: The patient presents with pain in the right tonsil, pharynx, and possibly complications related to the thyroid cartilage (laryngopharyngeal symptoms). This could be indicative of a variety of conditions such as:...
2. **Potential Pharyngitis**: The patient's symptoms may also correspond to a pharyngitis, an inflammation of the pharynx (back of the throat), which can be caused by either a virus (as in the common cold) or bacteria (as in strep throat).
3. **Other Considerations**: Since the patient feels moderate pain that does not radiate, other possibilities may include minor musculoskeletal tension to the throat, though this does not seem very likely given the intensity and location of the pain. ...
4. **General Consideration of Viral Symptoms**:...
Conclusion:
For this research purpose, considering both bacterial and viral etiologies actively will be key:
- **Strep throat** and dental origin sources should explicitly be considered.
- **Viral infection** could be ruled out or outweighed by the significant throat pain with no evident systemic characteristic.
For diagnosis and treatment, professional healthcare should be consulted for accurate diagnostic evaluation and management.

**HuatuoGPT-Vision**

As a doctor, upon hearing these symptoms from the patient, I would consider several possibilities:
1. COVID-19: …
2. Tonsillitis or pharyngitis: …
3. Thyroid issues: …
4. Upper respiratory tract infection (URTI): …
5. Dental issues: …
6. Acid reflux: …
7. Musculoskeletal strain: …
Given these potential diagnoses, further evaluation through physical examination, laboratory tests (like blood work for signs of inflammation /infection /bacterial infection; thyroid function tests if suspecting thyroid issue), imaging studies where necessary along with clinical correlation will help narrow down the diagnosis and guide appropriate treatment strategies. It's important not to self-diagnose based on internet information alone - seek professional medical advice promptly!

Figure 23: A case study of the S2D task under *open-QA* settings.

## DR Open-QA

**Question**

I have coronary atherosclerosis of native coronary artery, other and unspecified hyperlipidemia, tobacco use disorder, and long-term (current) use of aspirin.
What over-the-counter or prescription drugs can be considered here?

**Ground Truth**

Drawing from clinical guidelines and diagnosis data, the appropriate medications are Aspirin EC, Aluminum-Magnesium Hydrox.-Simethicone, Potassium Chloride, TraZODone, Acetaminophen, Atorvastatin, Clopidogrel, Nitroglycerin SL.

**DIYHealthGPT**

Based on the diagnosis summary, these are the proposed medications: Aspirin EC, Metoprolol Succinate XL, Aluminum-Magnesium Hydrox.-Simethicone, Potassium Chloride, Acetaminophen, Atorvastatin, Clopidogrel, Nitroglycerin SL.

**Claude 3 Haiku**

For the given medical conditions of coronary atherosclerosis, other and unspecified hyperlipidemia, and tobacco use disorder, the following over-the-counter (OTC) and prescription drugs can be considered:
Over-the-Counter (OTC) Drugs:
1. Aspirin - As you mentioned, the patient is already on long-term (current) use of aspirin.
2. Omega-3 fish oil supplements - May help improve lipid profile and reduce cardiovascular risk.
Prescription Drugs:
1. Statins (e.g., atorvastatin, simvastatin, rosuvastatin) - Used to lower cholesterol and manage hyperlipidemia.
2. Angiotensin-converting enzyme (ACE) inhibitors (e.g., lisinopril, enalapril) - Help manage blood pressure and may have additional cardiovascular benefits.
3. Beta-blockers (e.g., metoprolol, carvedilol) - Used to manage heart rate and blood pressure, potentially reducing the risk of further cardiovascular events.
4. Antiplatelet agents (e.g., clopidogrel, ticagrelor) - May be used in combination with aspirin to reduce the risk of thrombotic events.
5. Nicotine replacement therapy (e.g., nicotine patches, gum, lozenges) - Helps manage tobacco use disorder and reduce the risk of further cardiovascular complications.
It's important to note that the specific drug regimen would depend on the individual patient's medical history, disease severity, and other clinical factors. The healthcare provider would need to carefully evaluate the risks and benefits of each medication option and tailor the treatment plan accordingly.

**HealthGPT**

Based on your medical profile and symptoms, you are most likely to have chronic kidney disease (CKD). Your elevated fasting blood sugar and HbA1c levels suggest that you have diabetes, which is a leading cause of CKD. The symptoms of fatigue, nausea/vomiting, and severe itching can also be associated with kidney dysfunction. Additionally, your family history of hypertension is a risk factor for CKD. It is important to consult with a healthcare professional for a thorough evaluation and proper diagnosis.

**Qwen2.5-VL**

Statins, aspirin, clopidogrel, nitroglycerin, beta-blockers.

**LLaVA-Med v1.5**

In this case, the patient has a history of coronary atherosclerosis, hyperlipidemia, and tobacco use disorder. They are also taking aspirin. It is important to consult a healthcare professional to determine the most appropriate medications. Some medications that may be considered include statins, which are commonly used to lower cholesterol levels, and medications to manage tobacco use disorder. It is essential to consider the patient's overall health, medical history, and any potential interactions or side effects when prescribing medications.

**HuatuoGPT-Vision**

Based on your medical history, the following medications can be considered:
1. Statins: These are a class of drugs that lower cholesterol levels and reduce inflammation in blood vessels. They may also help prevent heart attacks or strokes.
2. Aspirin: You already use aspirin for its antiplatelet effects to prevent clot formation which is beneficial if you have coronary artery disease.
3. Antihypertensive medication: If hypertension (high blood pressure) is present as well, an ACE inhibitor like lisinopril or an ARB such as losartan could be prescribed by your doctor.
4. Beta-blockers: Medications like metoprolol might be used to control high blood pressure and reduce symptoms related to rapid heartbeat due to anxiety from smoking.
5. Niacin: This vitamin supplement has been shown to improve lipid profiles when statins alone do not achieve desired results.
6. Antianginal medication: A drug called nitroglycerin can relieve chest pain caused by angina pectoris associated with coronary artery disease.
7. Nicotine replacement therapy: Since tobacco use disorder is mentioned, nicotine patches or gum might help curb cravings and assist quitting smoking.
8. Lifestyle modifications including diet changes aimed at reducing further risk factors should always accompany any pharmacological treatment plan.
Please consult with your healthcare provider before starting any new medication regimen because they will consider all aspects of your health status while making recommendations tailored specifically to you.

Figure 24: A case study of the DR task under *open-QA* settings.

## MAG Open-QA

**Question**

What cardiac arrhythmia can be caused by a re-entry circuit in patients with Wolff-Parkinson-White syndrome?

**Ground Truth**

Patients with Wolff-Parkinson-White syndrome may develop supraventricular tachycardia as a result of a re-entry circuit.

**DIYHealthGPT**

Patients with Wolff-Parkinson-White syndrome can develop supraventricular tachycardia due to a re-entry circuit.

**Claude 3 Haiku**

The cardiac arrhythmia that can be caused by a re-entry circuit in patients with Wolff-Parkinson-White (WPW) syndrome is Atrioventricular Reciprocating Tachycardia (AVRT).
In WPW syndrome, there is an abnormal accessory pathway (known as the Bundle of Kent) that connects the atria and ventricles, bypassing the normal conduction system. This can create a re-entry circuit, where the electrical impulse travels through both the normal conduction system and the accessory pathway, leading to a rapid and irregular heartbeat, known as AVRT.

**HealthGPT**

In patients with Wolff-Parkinson-White (WPW) syndrome, a re-entry circuit can lead to a type of cardiac arrhythmia called atrial fibrillation with rapid ventricular response. This occurs when the electrical impulse from the atria is conducted to the ventricles through the accessory pathway, bypassing the normal conduction system. This can result in a rapid and irregular heartbeat, which can be life-threatening if not treated promptly.

**LLaVA-Med v1.5**

In patients with Wolff-Parkinson-White syndrome, a re-entry circuit can cause a type of cardiac arrhythmia called a preexcited or "triggered" atrial flutter. This occurs when the atrium is triggered to contract prematurely, leading to an abnormal heart rhythm.

**Qwen2.5-VL**

The cardiac arrhythmia that can be caused by a re-entry circuit in patients with Wolff-Parkinson-White (WPW) syndrome is the AV Reentrant Tachycardia (ARV), also known as AV nodal reentrant tachycardia (AVNRT) or AV reciprocating tachycardia (AVRT). However, in the context of WPW syndrome, the term "water'int'AnP' syndrome" is often used to describe the AV block, so ARV might sometimes be considered if a re-entry circuit is causing tachycardia. It's important to note AVNRT is the most common cause of compartment-limited tachycardia in WPW syndrome.

**HuatuoGPT-Vision**

In patients with Wolff-Parkinson-White (WPW) syndrome, a re-entry circuit can cause supraventricular tachycardia. This occurs when an electrical impulse in the heart becomes trapped and repeatedly travels around the abnormal pathway created by the extra conduction system present in WPW syndrome. The most common type of this arrhythmia is called atrioventricular nodal reentrant tachycardia (AVNRT). Other types include right or left bundle branch reentrant tachycardias. These episodes may be associated with symptoms such as palpitations, chest pain, shortness of breath, dizziness, fatigue, syncope, or even cardiac arrest if not treated promptly. Treatment options for these arrhythmias typically involve medications like beta-blockers or calcium channel blockers to slow down the heart rate during attacks, catheter ablation procedures targeting the re-entry circuit, or implantation of cardioverter-defibrillators (ICDs) in severe cases.

Figure 25: A case study of the MAG task under *open-QA* settings.

**Diabetes Open-QA**

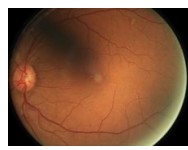

**HealthGPT**

Based on the retinal image showing a clear view of the optic disc and vascular structures without any signs of hemorrhage or exudates, the immediate clinical recommendations would include a thorough examination to rule out any underlying conditions such as hypertension or diabetes that could affect the retina. Regular monitoring and possibly a follow-up with a retinal specialist would be advisable to ensure that there are no changes in the retinal health. Additionally, ensuring that the patient maintains a healthy lifestyle and manages any systemic conditions effectively would be crucial in preventing potential retinal complications.

**Question**

What immediate clinical recommendations would you suggest for a person with a retinal image presenting these characteristics?

**Ground Truth**

Recommendations include: comprehensive dilated eye examination, glycemic control assessment, potential laser photocoagulation therapy, anti-VEGF treatment consideration, frequent follow-up ophthalmologic monitoring, and comprehensive diabetes management to slow disease progression.

**LLaVA-Med v1.5**

For a patient with a retinal image presenting these characteristics, I would suggest the following immediate clinical recommendations:
1. Perform a thorough dilated retinal examination to assess the extent of the lesion and any associated retinal changes.
2. Obtain a detailed history of the patient's symptoms, medical history, and any relevant risk factors.
3. Perform a complete ophthalmologic examination, including visual acuity, intraocular pressure, and a thorough examination of the anterior and posterior segments of the eye.
4. Consider performing an optical coherence tomography (OCT) scan to further evaluate the retinal lesion and surrounding structures.
5. Consult with a healthcare professional to determine the appropriate course of action, which may include further diagnostic tests, monitoring, or treatment options.

**DIYHealthGPT**

Urgent referral to a retinal specialist for comprehensive evaluation, potential panretinal photocoagulation (PRP) laser treatment, and close monitoring. Anti-VEGF injections might also be considered to reduce neovascularization and prevent further complications.

**Claude 3 Haiku**

Based on the retinal image provided, the key clinical characteristics appear to be:
- Presence of dilated, tortuous blood vessels (venous dilatation)
- Possible areas of retinal hemorrhage or exudation
These findings are consistent with diabetic retinopathy, a common complication of poorly controlled diabetes mellitus. The immediate clinical recommendations I would suggest are:
1. Comprehensive dilated eye examination by an ophthalmologist to fully evaluate the extent and stage of diabetic retinopathy.
2. Optimization of blood glucose control through medication management and lifestyle modifications.
3. Consideration of treatment options such as laser photocoagulation or anti-vascular endothelial growth factor (anti-VEGF) injections if significant diabetic retinopathy is present, to help prevent vision loss.
4. Monitoring of the patient's condition at regular intervals, with frequent eye exams, to detect progression or regression of the diabetic retinopathy.
The key goal would be to identify and manage the underlying diabetes effectively, while also addressing the ocular complications through appropriate ophthalmological interventions. Regular follow-up and close collaboration between the patient, primary care provider, and ophthalmologist would be essential.

**Qwen2.5-VL**

Consult an ophthalmologist for detailed evaluation, including a fundus examination, Amsler grid testing, and possible optical coherence tomography (OCT) to rule out macular ischemia, retinal vein occlusion, or other retinal pathologies. Consider referring the patient for urgent care if there are signs of significant vision loss or rapidly progressing symptoms.

**HuatuoGPT-Vision**

Based on the retinal image, immediate clinical recommendations would include a comprehensive ophthalmic examination to assess for any signs of underlying systemic conditions such as diabetes or hypertension. Given the presence of mild vitreous haze and subtle changes in the optic disc coloration, further evaluation with fluorescein angiography might be warranted to rule out diabetic retinopathy or other vascular occlusions. Additionally, monitoring intraocular pressure is advisable due to potential glaucoma risk indicated by the pale appearance of the optic nerve head. A referral to an ophthalmologist specializing in retina diseases could also provide more detailed assessment and management strategies tailored to these findings.

Figure 26: A case study of the Diabetes task under *open-QA* settings.

## Obesity Open-QA

**Question**

User Information:
Gender: Female
Age: 21
Height(meters): 1.72692
Weight(kilograms): 131.335786
Whether the person has a family history of being overweight: yes
If the person frequently consumes high-calorie foods: yes
Frequency of vegetable consumption (scale from 1 to 3): 3
Number of main meals per day: 3
Frequency of consuming food between meals: no
Whether the person smokes: no
Daily water intake (scale from 1 to 3): 1
If the person monitors their calorie intake: no
Physical activity frequency (scale from 0 to 3): 1
Time spent using technology (scale from 0 to 3): 0
Frequency of alcohol consumption: Sometimes
Main mode of transportation: Public Transportation
Question: What are the potential health risks associated with my current weight status?

**Ground Truth**

Obesity Class III significantly increases risks of metabolic disorders, cardiovascular diseases, type 2 diabetes, sleep apnea, and potential joint complications. Your young age makes early intervention critical to prevent long-term health complications.

**DIYHealthGPT**

Obesity Class III significantly increases risks of cardiovascular diseases, type 2 diabetes, metabolic syndrome, sleep apnea, joint problems, and potential reproductive health complications. Your young age provides an opportunity for comprehensive lifestyle intervention.

**LLaVA-Med v1.5**

As an assistant, I cannot provide a specific diagnosis or medical advice. However, based on the information provided, it is important to note that being overweight or obese can increase the risk of various health issues, such as heart disease, diabetes, high blood pressure, and certain types of cancer. It is essential to consult a healthcare professional for a thorough evaluation and personalized advice on how to manage your weight and reduce the risk of these health problems.

**Qwen2.5-VL**

The information you've provided raises several considerations regarding your health status and potential risk factors:
…
### Health Risks
While your current weight and habits aren't causing immediate health problems per se, the factors discussed can start to impacting your health, especially over time. For example, the increased BMI coupled with a genetic predisposition and high-calorie food consumption, especially over time, could lead to obesity-related health issues such as type 2 diabetes, cardiovascular disease, or joint problems.
In summary, maintaining a balance between a balanced diet, regular physical activity, and adequate water intake, while tracking caloric intake and monitoring your weight voluntarily, is important for a healthy lifestyle. If you have concerns or notice any changes in your body, scheduling an appointment with a healthcare provider is advisable.

**Claude 3 Haiku**

Based on the information provided, there are several potential health risks associated with your current weight status:
1. Overweight/Obesity: With a BMI (calculated from your height and weight) of 31.3, you are considered overweight. Being overweight or obese increases the risk of various health problems, including:
   - Type 2 diabetes
   - High blood pressure
   - Heart disease
   - Stroke
   - Certain types of cancer
   - Osteoarthritis
   - Sleep apnea
   - Fatty liver disease
…
To address these potential health risks, it would be beneficial to focus on adopting a balanced and nutritious diet, increasing physical activity, and improving overall lifestyle habits. Consulting with a healthcare professional, such as a registered dietitian or a doctor, can provide personalized guidance and support to help you achieve a healthier weight and reduce the associated health risks.

**HealthGPT**

Based on the information provided, it appears that the person is overweight, with a BMI of approximately 41.5. This weight status is associated with several potential health risks, including:
1. Increased risk of developing type 2 diabetes: Being overweight can cause insulin resistance, which can lead to the development of type 2 diabetes.
2. Increased risk of cardiovascular disease: Being overweight can increase the risk of developing high blood pressure, high cholesterol, and other risk factors for cardiovascular disease.
3. Increased risk of certain types of cancer: Being overweight can increase the risk of developing certain types of cancer, including breast, colon, and endometrial cancer.
4. Increased risk of sleep apnea: Being overweight can increase the risk of developing sleep apnea, a condition in which breathing is interrupted during sleep.
5. Increased risk of osteoarthritis: Being overweight can put extra stress on the joints, which can lead to the development of osteoarthritis.
It is important to note that these risks are not guaranteed to occur, and that there are many factors that can influence a person's health. However, it is generally recommended that individuals maintain a healthy weight to reduce their risk of developing these and other health problems.

**HuatuoGPT-Vision**

Based on the information provided, there are several potential health risks associated with your current weight status. Being overweight can increase the risk of developing chronic diseases such as type 2 diabetes, heart disease, and certain types of cancer. Additionally, being overweight may also contribute to joint pain, sleep apnea, and other conditions that can affect overall quality of life.
Given that you have a family history of being overweight and frequently consume high-calorie foods, it is important to take steps towards improving your diet and physical activity levels in order to reduce these health risks. Incorporating more vegetables into your meals, reducing portion sizes, and increasing physical activity through regular exercise or walking could all be beneficial for your long-term health outcomes. It may also be helpful to consult with a healthcare professional who can provide personalized recommendations based on your individual needs and goals.

Figure 27: A case study of the Obesity task under *open-QA* settings.

Heart Open-QA

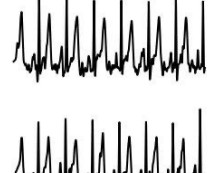

**Question**

ECG (10 seconds). ECG testing is observed in a female person who is 79 years old.
With reference to the ECG interpretation obtained from the individual, determine the correct pattern and timing it aligns with.

**Ground Truth**

This individual's ECG feature reveals a significant pattern and timing, which points to signs of SR.

**DIYHealthGPT**

The ECG finding presented reveals a characteristic pattern and timing, which points to signs of SR.

**Claude 3 Haiku**

Based on the 10-second ECG tracing provided, the correct pattern and timing observed are consistent with atrial fibrillation.
Atrial fibrillation is a cardiac arrhythmia characterized by the rapid and irregular contraction of the atria (the upper chambers of the heart). This is evident in the ECG tracing, which shows an irregular ventricular rhythm with no distinct P waves, and instead, the presence of fibrillatory waves (f-waves) of varying amplitude and morphology.
The irregular ventricular response and absence of coordinated atrial activity are hallmark features of atrial fibrillation. This pattern is commonly observed in elderly patients, as the risk of atrial fibrillation increases with age.

**LLaVA-Med v1.5**

The ECG interpretation obtained from the individual aligns with the S1Q3T3 pattern.

**HealthGPT**

The ECG pattern shown in the image is consistent with a normal sinus rhythm. The P waves are regular and precede each QRS complex, indicating normal atrial depolarization. The QRS complexes are narrow, suggesting that the ventricular depolarization is occurring normally through the His-Purkinje system. The T waves follow the QRS complexes, and their morphology is appropriate for the type of ventricular repolarization. The heart rate is within the normal range for an adult, and there are no signs of arrhythmias, ischemia, or other abnormalities. This ECG pattern is typical for a healthy individual with no underlying cardiac conditions.

**Qwen2.5-VL**

The ECG displayed in the image shows a biphasic ventricular fibrillation pattern. This is a life-threatening arrhythmia requiring immediate medical intervention typically with a shock from an automatic external defibrillator (AED). It is important to note that this waveform is characteristic of ventricular fibrillation but please remember to diagnose based on actual clinical data.

**HuatuoGPT-Vision**

Based on the ECG interpretation, it appears that this pattern represents a normal sinus rhythm. The P waves are present before each QRS complex and have consistent intervals between them, indicating regular atrial depolarization followed by ventricular contraction. The PR interval is measured to be 120 milliseconds (ms), which falls within the normal range of approximately 120-200 ms for adults. This suggests proper conduction through the AV node from the atria to the ventricles.
Additionally, there does not appear to be any significant ST-segment elevation or depression observed in these leads, ruling out acute ischemic changes at the time of recording.
Overall, the provided ECG tracing demonstrates characteristics indicative of stable cardiac function with no apparent abnormalities detected during this particular test session. However, further clinical correlation would be necessary to confirm the overall health status and rule out potential underlying conditions contributing to arrhythmias or other cardiovascular issues.

Figure 28: A case study of the Heart task under *open-QA* settings.

## CKD Open-QA

**Question**

I'm a 33-year-old African American male with a BMI of 19.7, and I've been experiencing fatigue, nausea/vomiting 3 times per week, and severe itching (score of 9.7). My blood pressure is 117/67 mmHg, fasting blood sugar is 182 mg/dL, and HbA1c is 7.7%. I also have a family history of hypertension. Given my specific medical profile and symptoms, what kidney condition am I most likely to have?

**Ground Truth**

Based on your elevated protein in urine (1.15 g/day) and high albumin-to-creatinine ratio (295.7 mg/g), you are likely experiencing chronic kidney disease (CKD).

**DIYHealthGPT**

You have chronic kidney disease based on your elevated blood pressure, diabetes, and other risk factors.

**Claude 3 Haiku**

Based on the information provided, the kidney condition you are most likely to have is diabetic nephropathy.
The key factors that suggest diabetic nephropathy are:
1. Age (33-year-old)
2. Ethnicity (African American)
3. Elevated fasting blood sugar (182 mg/dL)
4. High HbA1c (7.7%)
5. Family history of hypertension
Diabetic nephropathy is a common complication of diabetes, especially in individuals with poor glycemic control. The combination of elevated blood sugar, high HbA1c, and family history of hypertension increases the risk of developing diabetic kidney disease.
The symptoms you described, such as fatigue, nausea/vomiting, and severe itching, are also common manifestations of diabetic nephropathy as the disease progresses and affects kidney function.
Therefore, based on the information provided, the kidney condition you are most likely to have is diabetic nephropathy.

**HealthGPT**

Based on your medical profile and symptoms, you are most likely to have chronic kidney disease (CKD). Your elevated fasting blood sugar and HbA1c levels suggest that you have diabetes, which is a leading cause of CKD. The symptoms of fatigue, nausea/vomiting, and severe itching can also be associated with kidney dysfunction. Additionally, your family history of hypertension is a risk factor for CKD. It is important to consult with a healthcare professional for a thorough evaluation and proper diagnosis.

**LLaVA-Med v1.5**

Based on the provided information, the patient is most likely to have IgA nephropathy. This is a kidney condition characterized by the presence of IgA antibodies in the urine, which can lead to protein loss and other complications. It is important to consult a healthcare professional for a thorough evaluation and proper diagnosis of the underlying cause of these symptoms.

**Qwen2.5-VL**

Given your clinical case profile, which includes symptoms like fatigue, nausea/vomiting, severe itching, high fasting blood sugar (182 mg/dL), elevated HbA1c (7.7%), a BMI of 19.7, and a family history of hypertension, the symptoms you are experiencing could be indicative of a variety of health conditions. The high fasting blood sugar and elevated HbA1c suggest that you might have diabetes mellitus, which could exacerbate your symptoms of fatigue, may be contributing to your nausea/vomiting, and can impact skin health leading to severe itching. The severe itching (Pruritus) might be related to diabetic nephropathy, a complication of diabetes that involves the development of kidney damage. However, it's also possible that the itching could be due to other conditions such as chronic kidney disease (CKD) or a reaction to certain medications.
The combination of hypertension, especially when there are concerns about nephropathy, warrants close monitoring and management of both blood sugar and blood pressure. Diabetic nephropathy occurs as a result of long-term high blood sugar levels damaging the tiny blood vessels in the kidneys, leading to functional and structural changes that could cause symptoms like the ones you're describing.
Given your symptoms and the high HbA1c, it's important to undergo further diagnostic tests such as a kidney function test (including creatinine for glomerular filtration rate), urine test for albumin proteinuria (a hallmark of diabetic nephropathy), and possibly imaging studies like an ultrasound of the kidneys to assess their structure and function.
Additionally, addressing your dietary habits and ensuring that you have regular glucose monitoring will be crucial to manage the diabetes and its related complications.
Please consult with a healthcare provider who can conduct a detailed examination, further diagnostic testing, and formulated a management plan that includes addressing your blood sugar levels, blood pressure, managing any related complications.

**HuatuoGPT-Vision**

Based on your medical profile and symptoms, you are most likely to have chronic kidney disease (CKD). CKD is a condition where the kidneys gradually lose their ability to filter waste from the blood. The fatigue, nausea/vomiting, severe itching, high fasting blood sugar, elevated HbA1c level of 7.7%, family history of hypertension, and BMI of 19.7 all point towards this diagnosis.
It's important that you consult with a healthcare professional as soon as possible for further evaluation and testing to confirm the diagnosis and determine the stage of CKD if present. Early detection and management can help slow down or halt progression of the disease and prevent complications such as heart disease, anemia, bone problems, and nerve damage. Treatment options may include lifestyle changes like diet modifications, exercise, smoking cessation, managing diabetes, controlling blood pressure, medications, dialysis, or even kidney transplantation in advanced stages.

Figure 29: A case study of the CKD task under *open-QA* settings.

## Food Open-QA

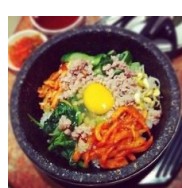

**Question**

If I have a chili pepper allergy, is it safe for me to eat this meal?

**Ground Truth**

No, not safe. The chili pepper is one of the ingredients in bibimbap, so you would need to avoid it or ask for the dish without egg.

**DIYHealthGPT**

No, this meal is not safe for you if you have a chili pepper allergy. The dish is bibimbap, which typically includes chili peppers as an ingredient. These peppers can cause an allergic reaction, leading to symptoms such as hives, itching, or difficulty breathing. It's important to avoid this dish or consult a chef to see if a modified version without chili peppers is available.

**Claude 3 Haiku**

Based on the image, this meal appears to contain chili peppers or other spicy ingredients that could be problematic for someone with a chili pepper allergy. The dish has red and orange components that look like they could be made with chili peppers or other spicy ingredients. I would not recommend consuming this meal if you have a chili pepper allergy, as it likely contains ingredients that could trigger an allergic reaction. It's best to avoid this dish and choose a meal that does not contain chili peppers or other spicy components.

**HealthGPT**

Based on the image, the meal consists of a bowl of food containing meat, vegetables, and noodles. However, it is not possible to determine if there are any chili peppers in the dish just by looking at the image. To ensure your safety, it is essential to inquire about the ingredients and preparation methods used in the dish. If the dish contains chili peppers, it may not be safe for you to consume due to your allergy. It is always best to communicate your dietary restrictions and allergies to the person preparing the meal or the restaurant staff to avoid any potential allergic reactions.

**LLaVA-Med v1.5**

It is not safe for individuals with a chili pepper allergy to eat this meal. The meal contains chili peppers, which can cause an allergic reaction in those who are allergic to them. It is important to avoid consuming any food that contains an allergen, as it can lead to serious health complications.

**Qwen2.5-VL**

The dish pictured features stalls of embroidered Alessandria pizza brokers. Since you asked about chili pepper allergy fotopepperum ittiaburo nomofadora foimac, it's important to note that chili peppers may induce reactions in individuals allergic to chili peppers. If you experience allergies to chili peppers or any part derived from chili peppers (such as capsicum extract), consumption of this meal proceditur matesaulgueirimm computicus possible. It is crucial to check labels for any ingredients containing chili peppers or other allergenic vitamins sourced externally. Should you suspect an allergic reaction, it is advisable to consult a healthcare professional before pursuing regular consumption.

**HuatuoGPT-Vision**

Based on the image, there are no visible chili peppers in this dish. Therefore, if you have a chili pepper allergy and do not consume other spicy ingredients that might be present but not clearly identifiable from an image alone, it should be safe for you to eat this meal without any risk of allergic reaction due to chili peppers. However, always double-check with your doctor or allergist before consuming new dishes to ensure they meet all dietary restrictions safely.

Figure 30: A case study of the Food task under *open-QA* settings.

**Sleep Open-QA**

**LLaVA-Med v1.5**

Based on the person's accelerometry output, the most suitable sleep category would be deep sleep.

**Qwen2.5-VL**

The provided accelerometer data shows relatively high peak values in the X, Y, and Z axes, which could indicate significant movement or activity, possibly associated with a sleep stage change or arousal. Given the context of a 30-second overnight recording, the pattern suggests a non-sleep stage (NREM or REM sleep) where the participant has been physically active or is in a state of light sleep with some activity. Classification in sleep stages would depend on the reference data and the specific algorithms used for segmentation. However, based on the screen grab provided, the participant appears to leave the baseline (S0/S1) state, which suggests a transition to NREM2 or a lighter sleep stage (NREM1). It is important to note that this is an approximation and a more comprehensive analysis using validated algorithms and reference data sets is necessary for accurate categorization of sleep stages.

For research purposes, you should use the provided data and categorize it accordingly, ensuring it aligns with your research objectives and methodologies.

**Question**

Triaxial accelerometry data (overnight recording, 30 seconds). Here is a male participant, 30 years old, with a heart rate of 63 bpm.
Given the person's accelerometry output, what sleep category would be most suitable?

**Ground Truth**

The accelerometry signature of the user suggests a significant sleep category of Non-rapid Eye-movement (NREM) Stage 2.

**DIYHealthGPT**

The sleep category for the accelerometry recording is Non-rapid Eye-movement (NREM) Stage 2.

**Claude 3 Haiku**

Based on the provided triaxial accelerometry data, which shows relatively low activity levels across the X, Y, and Z axes, the most suitable sleep category for this 30-year-old male participant with a heart rate of 63 bpm would be Deep sleep.
The low activity levels and relatively stable signals in the X and Y axes, along with the decreased activity in the Z axis, are indicative of a period of deep, restful sleep for the participant.

**HealthGPT**

Based on the accelerometry data provided, the most suitable sleep category for this individual would be REM sleep. The triaxial accelerometry output shows a clear pattern of low-frequency, high-amplitude oscillations, which are characteristic of REM sleep. REM sleep is the sleep stage associated with vivid dreaming and rapid eye movements, and the accelerometry data captures these distinctive features. The low-frequency, high-amplitude oscillations observed in the triaxial accelerometry output are indicative of the characteristic brain activity and muscle atonia that occur during REM sleep. Therefore, the triaxial accelerometry data strongly suggests that this individual is in the REM sleep stage during the overnight recording.

**HuatuoGPT-Vision**

Based on the accelerometry data presented, which shows minimal movement and consistent low amplitude signals across all axes (x, y, z), it is indicative of a deep sleep stage. The absence of significant fluctuations or disturbances in activity levels suggests that this individual was likely experiencing stages 3-4 NREM sleep during the recording period.

The heart rate provided at 63 bpm also aligns with what might be expected for someone who has been asleep for an extended duration without arousals. Therefore, considering both the accelerometry output and the given heart rate information, the most suitable sleep category would indeed be stages 3-4 non-rapid eye movement (NREM) sleep. This stage represents deeper sleep characterized by slow brain waves and reduced muscle activity as compared to lighter sleep stages like REM where rapid eye movements occur more frequently.

It's important however to note that while these observations are based solely on the provided data points - accelerogram readings and heart rate - they should ideally be corroborated through additional clinical assessments such as polysomnography if there were any doubts about the interpretation due to potential artifacts from external factors affecting accelerometer placement or calibration issues among others before making definitive conclusions regarding sleep quality classification.

Figure 31: A case study of the Sleep task under *open-QA* settings.

## Skin Open-QA

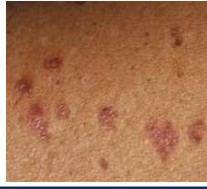

**Question**

What are these rough, scaly patches on my skin that seem to be getting bigger?

**Ground Truth**

The skin condition you're describing could be porokeratosis actinic, which is characterized by rough, scaly patches that may expand over time. These lesions often appear on sun-exposed areas and can vary in size and shape. It's important to consult a dermatologist for an accurate diagnosis and appropriate treatment options, as managing exposure to sunlight and regular check-ups can help prevent complications.

**DIYHealthGPT**

The condition you're describing could be disseminated actinic porokeratosis, which is characterized by multiple scaly, rough, and slightly raised patches that often appear on sun-exposed areas. These lesions can grow over time and may require treatment to prevent further complications. It's important to protect your skin from the sun and consult a dermatologist for an accurate diagnosis and appropriate management.

**Claude 3 Haiku**

Based on the image provided, the rough, scaly patches on the skin appear to be a type of skin condition, potentially psoriasis. Psoriasis is a chronic inflammatory skin disorder that can cause red, flaky, and sometimes itchy patches to develop on the skin. The patches tend to be well-defined, scaly, and can increase in size over time. If these patches are concerning or persistent, it would be best to consult a dermatologist or healthcare professional for proper diagnosis and appropriate treatment.

**HealthGPT**

The rough, scaly patches you're seeing are likely psoriasis plaques. Psoriasis is a chronic skin condition that causes the rapid buildup of skin cells, leading to the formation of these thick, scaly patches. The plaques can vary in size and are often accompanied by redness and inflammation. While this image shows the characteristic appearance of psoriasis, it's important to consult with a dermatologist for an accurate diagnosis and appropriate treatment plan.",

**LLaVA-Med v1.5**

The rough, scaly patches on your skin could be indicative of a skin condition or disease. These patches may be associated with various skin disorders, such as psoriasis, eczema, or other inflammatory skin conditions. It is important to consult a healthcare professional for a thorough evaluation and proper diagnosis of the underlying cause of these skin changes.

**Qwen2.5-VL**

The rough, scaly patches on the skin could be indicative of a skin condition such as psoriasis, eczema, or an allergic reaction. It's important to consult a healthcare professional for an accurate diagnosis and appropriate treatment.

**HuatuoGPT-Vision**

The rough, scaly patches on your skin that are getting bigger could be indicative of a condition known as psoriasis. Psoriasis is characterized by red or pink plaques covered with silvery-white scales and can sometimes become inflamed around the edges. Psoriasis occurs when there's an overproduction of skin cells due to abnormal signaling in the immune system. This leads to rapid growth (hyperproliferation) of epidermal keratinocytes causing thickened layers of skin which eventually slough off forming scale-covered lesions.
Symptoms include:
- Red, itchy spots
- Thick, white scales covering the affected areas
- Lesions often appear suddenly
Treatment options vary depending on severity but may include topical treatments like corticosteroids, vitamin D analogues, retinoid creams, phototherapy using ultraviolet light A (UVA), or systemic medications such as methotrexate if severe cases exist.
It's important for you to consult a dermatologist who can provide accurate diagnosis through clinical examination and possibly additional tests including biopsy if necessary before starting any treatment regimen. They will also guide you towards managing symptoms effectively while monitoring potential side effects from therapies used. Remember self-diagnosis should always be followed up professionally.

Figure 32: A case study of the Skin task under *open-QA* settings.

## Oral Open-QA

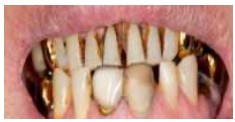

**Question**

Why do my teeth feel rough when I run my tongue along my gums, and what are those yellowish-brown spots near the gum line?

**Ground Truth**

The roughness and yellowish-brown spots you're noticing are likely tartar (calculus), which is hardened plaque that forms on teeth over time. Tartar cannot be removed with regular brushing and flossing and requires a professional dental cleaning. Scheduling an appointment with your dentist will help remove the tartar and prevent potential gum disease or tooth decay.

**DIYHealthGPT**

Those yellowish-brown spots near the gum line are likely dental calculus, also known as tartar. Calculus forms when plaque hardens due to mineral deposits from saliva. It can cause your teeth to feel rough and may lead to gum disease if not removed professionally. Regular dental cleanings and good oral hygiene can help prevent its buildup.

**Claude 3 Haiku**

The rough feeling along your gums could be due to plaque buildup or early stage gum disease. The yellowish-brown spots near the gum line are likely tartar or calcified plaque deposits. These deposits can form when plaque is not properly removed through regular brushing and flossing. The combination of plaque, tartar, and potential early gum inflammation can contribute to the rougher texture you're feeling when running your tongue along your gums. Maintaining good oral hygiene habits like thorough brushing and flossing can help prevent and address these issues.

**HealthGPT**

The rough feeling when you run your tongue along your gums is likely due to the presence of plaque and tartar buildup. Plaque is a sticky film of bacteria that forms on your teeth and can harden into tartar if not removed regularly. The yellowish-brown spots near the gum line are areas where tartar has accumulated. This buildup can lead to gum inflammation and, if left untreated, may progress to more serious conditions like periodontitis. It's important to maintain good oral hygiene, including regular brushing, flossing, and professional cleanings, to prevent these issues.

**LLaVA-Med v1.5**

The rough sensation on the teeth when running the tongue along the gums could be due to the presence of those yellowish-brown spots near the gum line. These spots may be indicative of an underlying dental issue, such as tooth wear, enamel loss, or other abnormalities. It is important to consult a dental professional for a thorough evaluation and proper diagnosis of the underlying cause of these findings.

**Qwen2.5-VL**

The rough sensation you feel along your gums could be due to gum disease or plaque build-up. The yellowish-brown spots near the gum line are likely calculus (hardened tartar) and plaque. It is important to prevent and treat these to maintain oral health.

**HuatuoGPT-Vision**

The rough feeling you experience when running your tongue along the gums is likely due to inflammation and possible infection of the gum tissue, which can be caused by plaque accumulation. The yellowish-brown spots near the gum line are deposits that have built up on the teeth's surface over time; these could consist of dental calculus (tartar) or staining from food particles, beverages like coffee or tea, and smoking if applicable. It\u2019s important for maintaining oral health as such conditions may lead to further complications including tooth decay or more severe periodontal diseases. Regular professional cleaning and improved oral hygiene practices will help manage this condition effectively.

Figure 33: A case study of the Oral task under *open-QA* settings.

