# OpenReview forum: "DIYHealth Suite: Dataset, Model, and Benchmark for Health Management at Home"
_ICLR.cc/2026/Conference — Submitted to ICLR 2026_

### Official Review · Reviewer_r2KA · 2025-10-19

**Soundness:** 2
**Presentation:** 3
**Contribution:** 2
**Rating:** 4
**Confidence:** 3

**Summary:**

This paper presents DIYHealth Suite, a comprehensive framework for AI-driven home healthcare. To address challenges such as data heterogeneity, task variability, and lack of benchmarks in non-clinical settings, the authors introduce three key components: DIYHealth-900K, a large-scale multimodal dataset of real-world home care scenarios; DIYHealthGPT, an adaptive foundation model leveraging a novel Hybrid Hyper Low-Rank Adaptation technique for balancing generalization and personalization; and DIYHealthBench, the first benchmark for evaluating home care tasks. The proposed model achieves state-of-the-art results on 11 tasks, demonstrating its effectiveness for personalized, scalable health management at home.

**Strengths:**

1. The home healthcare scenario addressed in this work is practical and useful.
2. The dataset and benchmark are comprehensive, covering a wide range of home healthcare tasks.
3. The effort in constructing the dataset—despite relying solely on public sources—is commendable.
4. The proposed adaptation method is empirically validated and demonstrates considerable effectiveness.

**Weaknesses:**

1. DIYHealth-900K is constructed entirely from existing public datasets, so no new data or tasks are introduced in this work.
2. The strong performance of DIYHealthGPT is largely attributable to training on in-distribution data, as the training and test sets are drawn from the same dataset. Comparisons against other models—whether general-purpose or medical—are therefore unfair, since those baselines are effectively evaluated in a zero-shot setting. This shifts the novelty from an "adaptive" foundation model to a "unique" dataset, which itself is not produced by this work.
3. The proposed evaluation tasks appear relatively easy, with even small models achieving strong results and the proposed method approaching near-perfect performance in many cases. This suggests that larger models may already saturate the benchmark, limiting its usefulness for measuring state-of-the-art progress.

**Questions:**

1. The paper lacks an explanation of the used evaluation metrics, such as Matthews Correlation Coefficient (MCC). Providing their definition, interpretation, and justification for use would improve the presentation.

---

> ### Author Response · Authors · 2025-11-21
> **Part one of the response to Reviewer r2KA**
>
> We are grateful for your insightful feedback. We provide detailed responses to your concerns below with the updated results and corresponding revisions in the manuscript (highlighted in blue).
>
> *Review W1: DIYHealth-900K is constructed entirely from existing public datasets, so no new data or tasks are introduced in this work.*
>
> Response: Thank you for raising this point. We appreciate the opportunity to clarify the scope and contribution of DIYHealth-900K.
> While the dataset indeed builds upon publicly available sources, it is not a direct aggregation of existing datasets.
> Instead, our main contribution lies in constructing a comprehensive DIYHealth Suite tailored specifically to home care scenarios, achieved not through raw data collection but through multimodal harmonization, task reformulation, and validation of heterogeneous medical data, going significantly beyond what the original datasets were designed to support.
>
> Firstly, we introduce 11 home care tasks, which are designed to reflect realistic home care use cases covering personalized health management, chronic disease risk assessment, and daily health monitoring.
>
> Secondly, raw data from 20 public sources are systematically filtered to remove content that would not realistically occur in home settings (e.g., advanced imaging modalities, professional terminology, or hospital-specific metadata). The remaining samples are then reformulated into human-producible questions, descriptions, and statements, enabling alignment with home care needs.
>
> Thirdly, we design a DIYHealth data engine to perform data processing, consistency checking, and medical validation. This step produces a transformed dataset that is semantically coherent, clinically grounded, and suitable for home care scenarios, the quality of which cannot be obtained from the raw public datasets.
>
> Finally, the resulting DIYHealth-900K is the first large-scale, standardized, multimodal dataset explicitly curated for home care applications, rather than clinical environments. This makes it fundamentally new in terms of task definitions, data format, and intended usage.
> Thus, while the underlying sources are public, DIYHealth-900K provides a systematically curated and reformulated resource that supports home-care specific data, tasks, and evaluation procedures that were not previously available for this domain.
>
> (continued below)

---

> ### Author Response · Authors · 2025-11-21
> **Part two of the response to Reviewer r2KA**
>
> *Review W2: The strong performance of DIYHealthGPT is largely attributable to training on in-distribution data, as the training and test sets are drawn from the same dataset. Comparisons against other models—whether general-purpose or medical—are therefore unfair, since those baselines are effectively evaluated in a zero-shot setting. This shifts the novelty from an "adaptive" foundation model to a "unique" dataset, which itself is not produced by this work.*
>
> Response: In line with established protocols in recent LLM benchmark studies [1,2,3], our evaluation assesses baseline models in their default inference settings without further adaptation to our dataset. This setup is aligned with our objective: to examine whether current generalist and medical-specific foundation models can effectively operate in realistic home-care scenarios. Our experimental results show they largely struggle in this setting, highlighting **the gap between existing models and the demands of home-based health management**.
>
> To further analyze the source of performance gain and ensure fairness, we have additionally fine-tuned two representative baselines, Gemma 3-4B and LLaVA-Med v1.5-7B, one generalist and one medical-specific model, using the same DIYHealth-900K training data.
> As shown in the table below, DIYHealthGPT-3.8B consistently outperforms Gemma 3-4B and LLaVA-Med v1.5-7B by a substantial margin on the four evaluation metrics under both closed-QA and open-QA settings, despite Gemma 3-4B and LLaVA-Med v1.5-7B  having larger model sizes.
> These results indicate that the observed improvements cannot be attributed only to the dataset. Instead, the joint effect of the H$^2$LoRA architecture, training strategy, and curated DIYHealth-900k contributes to the model's effectiveness in home-care scenarios.
>
> We have incorporated these results and further analysis in Section 6.3 (lines 486–497) to clarify this point. We hope that this expanded evaluation clarifies the experimental setting and demonstrates the necessity of adaptive modeling for home care tasks.
>
> | Model             | Closed-QA |           |  Open-QA  |           |
> | ----------------- | :-------: | :-------: | :-------: | :-------: |
> |                   |    ACC    |    MCC    |  F1-Bio   |    RL     |
> | Gemma 3-4B        |   80.96   |   74.42   |   84.72   |   41.59   |
> | LLaVA-Med v1.5-7B |   77.58   |   69.51   |   86.28   |   49.63   |
> | DIYHealthGPT-3.8B | **86.80** | **82.36** | **87.34** | **52.11** |
>
> Furthermore, DIYHealthGPT incorporates the proposed H$^2$LoRA architecture, which introduces task-level expert specialization and instance-level personalization, enabling fine-grained adaptive modeling that goes beyond standard fine-tuning and is not supported by existing LoRA or MoELoRA variants. This architectural novelty is further evidenced by our empirical comparisons in Section 6.3.
> Additionally, as discussed in our response to Comment W1, DIYHealth-900K is systematically curated and reformulated to support home-care–specific data, tasks, and evaluation procedures, establishing a research setting that was previously unavailable in this domain.
>
> &nbsp;
>
> *Review W3: The proposed evaluation tasks appear relatively easy, with even small models achieving strong results and the proposed method approaching near-perfect performance in many cases.
> This suggests that larger models may already saturate the benchmark, limiting its usefulness for measuring state-of-the-art progress.*
>
> Response: We appreciate your observation and agree that some tasks, such as S2D and CKD in the closed-QA setting, and Food and Oral in the open-QA setting, tend to be relatively easier and yield strong performance even for small models.
> This diversity aligns with our objective: to systematically evaluate both factual competency and reasoning capability across a broad spectrum of realistic home-care scenarios.
>
> Accordingly, instead of enforcing equal task difficulty, DIYHealthBench is designed to reflect the inherent diversity of real-world home-care scenarios, where simple routine queries (e.g., dietary advice or common symptom descriptions) naturally coexist with more complex tasks that demand deeper reasoning or domain-specific understanding.
> Notably, the latter category, including Sleep, Heart, and DR under the open-QA setting, remains far from saturated.
> Even state-of-the-art models still exhibit substantial performance gaps on these tasks. This indicates that the benchmark continues to expose meaningful challenges and provides ample room for future model improvements.
>
> Overall, DIYHealthBench captures a heterogeneous task landscape: some tasks reflect routine home care queries where high accuracy is expected, whereas others pose challenging reasoning problems that remain unresolved. We have revised the paper and highlighted the difficult tasks within the benchmark to guide future research in Section 7 (lines 510 - 512).
>
> (continued below)

---

> ### Author Response · Authors · 2025-11-21
> **Part three of the response to Reviewer r2KA**
>
> *Review Q1: The paper lacks an explanation of the used evaluation metrics, such as Matthews Correlation Coefficient (MCC). Providing their definition, interpretation, and justification for use would improve the presentation.*
>
> Response: As suggested, we have enhanced the explanation of the evaluation metrics and added a new section in Appendix D.5 to provide their detailed definitions and descriptions.
> Following the evaluation of previous healthcare studies on QA settings [4,5,6,7], we employ Accuracy (ACC) and Matthews Correlation Coefficient (MCC) for closed-QA tasks and use F1-BioBERT, ROUGE-L, F1-RadGraph, and BLEU for open-QA tasks.
> We summarize the evaluation metrics as follows.
>
> * ACC measures the proportion of predictions that exactly match the ground-truth labels.
> * MCC measures the correlation between the predicted and true labels across all classes by considering the full confusion matrix.
> * F1-BioBERT measures the semantic alignment between generated answers and ground truth using BioBERT embeddings. It captures domain-specific semantic similarity.
> * ROUGE-L evaluates the longest common subsequence between predicted and ground truth answers, reflecting recall-oriented similarity.
> * F1-RadGraph computes entity-level F1 scores based on RadGraph evaluation, evaluating the correctness of clinical entity extraction and relation grounding.
> * BLEU computes the precision of n-gram overlaps between predictions and ground truth to assess lexical fidelity.
>
> Together, these six metrics jointly assess correctness, clinical grounding, semantic fidelity, and textual similarity, offering a balanced and reliable evaluation across the diverse tasks in DIYHealthBench. Please refer to the added Appendix D.5 for more details about these metrics.
>
> &nbsp;
>
> We are grateful for the chance to discuss our work's potential and wish to thank you again for your valuable input. We hope to have addressed your concerns and would highly appreciate your consideration for re-evaluating your initial rating.
>
> *References:*
>
> [1] Lin, Tianwei, et al. "HealthGPT: A Medical Large Vision-Language Model for Unifying Comprehension and Generation via Heterogeneous Knowledge Adaptation." Forty-second International Conference on Machine Learning.
>
> [2] Xie, Qianqian, et al. "PIXIU: a large language model, instruction data and evaluation benchmark for finance." Proceedings of the 37th International Conference on Neural Information Processing Systems. 2023.
>
> [3] Wu, Yutong, et al. "InverseCoder: Self-improving Instruction-Tuned Code LLMs with Inverse-Instruct." Proceedings of the AAAI Conference on Artificial Intelligence. Vol. 39. No. 24. 2025.
>
> [4] Moor, Michael, et al. "Med-flamingo: a multimodal medical few-shot learner." Machine Learning for Health (ML4H). PMLR, 2023.
>
> [5] Li, Sijing, et al. "Eyecaregpt: Boosting comprehensive ophthalmology understanding with tailored dataset, benchmark and model." Proceedings of the 33rd ACM International Conference on Multimedia. 2025.
>
> [6] Xu, Weiwen, et al. "Lingshu: A Generalist Foundation Model for Unified Multimodal Medical Understanding and Reasoning." arXiv preprint arXiv:2506.07044 (2025).
>
> [7] Sellergren, Andrew, et al. "Medgemma technical report." arXiv preprint arXiv:2507.05201 (2025).

---

### Official Review · Reviewer_QoBN · 2025-10-31

**Soundness:** 3
**Presentation:** 3
**Contribution:** 3
**Rating:** 6
**Confidence:** 4

**Summary:**

This paper introduces a unified framework for AI-based home healthcare. It includes DIYHealth-900K, a large multimodal dataset, DIYHealthGPT, a foundation model using a novel Hybrid Hyper Low-Rank Adaptation (H2LoRA) for personalized reasoning, and DIYHealthBench, a benchmark for 11 home care tasks. Experiments show that DIYHealthGPT outperforms both general and medical foundation models in accuracy and biomedical fidelity, advancing personalized health management beyond clinical settings.

**Strengths:**

1. The experiments are comprehensive. Comparisons between DIYHealthGPT and baselines in both general and medical domains demonstrate the effectiveness of DIYHealthGPT, while ablation studies further showcase the correctness of rationale behind the design of each component of DIYHealthGPT.

2. Abundant content regarding DIYHealthBench notably enhances the readability of this study, facilating readers' understanding on the data.

3.Repo has been provided with detailed instructions and clear codes, ensuring decent reproducibility.

**Weaknesses:**

1. Technically, the proposed DIYHealthGPT is not considerably novel. Previous works in both general [1] and medical [2] domains have already applied similar MoE-based architecture to benefit the training of foundation model. In my perspective, the major reason that DIYHealthGPT outperforms other baselines is that it has been further trained with data in DIYHealth-900K, thus customized to this task, rather than that it has apparently more advanced architecture or training strategy. It will be beneficial if the authors could involve more discussions regarding the performance in this perspective.

[1]. Chen, Zeren, et al. "Octavius: Mitigating Task Interference in MLLMs via LoRA-MoE." The Twelfth International Conference on Learning Representations.
[2]. Wang, Xiaochen, et al. "FEDKIM: Adaptive Federated Knowledge Injection into Medical Foundation Models." Proceedings of the 2024 Conference on Empirical Methods in Natural Language Processing. 2024.

2. It is still not fully clear to me that how the DIY healthcare scenario becomes different from other healthcare applications [3]. While paying notable attention to the method and data, the authors did not fully demonstrate the uniqueness of the setting, obscuring the motivation and undermining the evaluation of this study.

[3]. Liu, Jie, et al. "Asclepius: A spectrum evaluation benchmark for medical multi-modal large language models." Proceedings of the 63rd Annual Meeting of the Association for Computational Linguistics (Volume 1: Long Papers). 2025.

**Questions:**

I notice that although the authors have created a repo for the codes, the data are still unavailable. I am personally curious whether and how the authors would like to publicize the data.

---

> ### Author Response · Authors · 2025-11-21
> **Part one of the response to Reviewer  QoBN**
>
> We thank you for your recognition of our work and valuable comments.
> Below, we provide detailed responses to the specific concerns you raised.
>
>
> *Review W1: Technically, the proposed DIYHealthGPT is not considerably novel. Previous works in both general [1] and medical [2] domains have already applied similar MoE-based architecture to benefit the training of foundation model. In my perspective, the major reason that DIYHealthGPT outperforms other baselines is that it has been further trained with data in DIYHealth-900K, thus customized to this task, rather than that it has apparently more advanced architecture or training strategy. It will be beneficial if the authors could involve more discussions regarding the performance in this perspective.*
>
> Response: We appreciate the reviewer's comments and the relevant references, and we agree that MoE-LoRA techniques have been investigated in both general and medical domains.
> However, DIYHealthGPT goes beyond existing formulations in several technically distinct aspects, specifically motivated by the demands of home care scenarios.
>
>
> First, our H$^2$LoRA integrates a Low-Rank Expert Mixture with a Hyper LoRA Adaptation module. Unlike typical MoE-LoRA designs, our **Low-Rank Expert Mixture** trains task-specific LoRA experts while using a shared matrix $\mathbf{A}$ to maintain parameter efficiency. More importantly, the **Hyper LoRA Adaptation** module employs a hypernetwork to generate task-aware and instance-specific low-rank adaptations, enabling the model to capture personalization and fine-grained variability that frequently arise in home care settings (e.g., differences in patient demographics, device heterogeneity, or self-reported inputs).
> **Existing MoE-LoRA approaches do not incorporate such instance-level adaptive modeling.**
>
> Second, **the two components play complementary roles**: the Low-Rank Expert Mixture facilitates efficient task-specific tuning, while Hyper LoRA Adaptation provides individualized adaptation. As shown in Table 3, each mechanism improves performance on its own, and their combination achieves the strongest results, demonstrating that the full H$^2$LoRA architecture offers benefits beyond existing MoE-LoRA formulations.
>
> As further validation, our experimental comparison with existing LoRA variants (including MoE-LoRA) in Section 6.3 shows consistent performance gains, improving by 1.6% in the open-QA setting and 3.7% in the closed-QA setting, which supports the necessity of modeling both task-level and instance-level adaptations in home care scenarios.
>
>
> To rigorously analyze the **source of performance gain**, we have additionally fine-tuned two representative baselines, Gemma 3-4B and LLaVA-Med v1.5-7B, using the same DIYHealth-900K training data.
> As shown in the table below, DIYHealthGPT-3.8B consistently outperforms Gemma 3-4B and LLaVA-Med v1.5-7B by a substantial margin on the four evaluation metrics under both closed-QA and open-QA settings, despite Gemma 3-4B and LLaVA-Med v1.5-7B  having larger model sizes. These results indicate that the observed improvements cannot be attributed only to the dataset. Instead, the joint effect of the H$^2$LoRA architecture, training strategy, and curated DIYHealth-900k contributes to the model's effectiveness in home-care scenarios.
> We have added the results and further discussions in Section 6.3 (lines 486–497) to clarify this point.
>
> | Model             | Closed-QA |           |  Open-QA  |           |
> | ----------------- | :-------: | :-------: | :-------: | :-------: |
> |                   |    ACC    |    MCC    |  F1-Bio   |    RL     |
> | Gemma 3-4B        |   80.96   |   74.42   |   84.72   |   41.59   |
> | LLaVA-Med v1.5-7B |   77.58   |   69.51   |   86.28   |   49.63   |
> | DIYHealthGPT-3.8B | **86.80** | **82.36** | **87.34** | **52.11** |
>
>
> For completeness, we note that our evaluation of the main results follows the standard protocol adopted in recent LLM benchmark studies [4,5,6], where baseline models are evaluated in their default inference settings without further adaptation to our dataset.
> This setup is intentional and aligned with our objective: to examine whether current generalist and medical-specific foundation models can effectively operate in realistic home-care scenarios.
> Our experimental results demonstrate that most existing models struggle in this setting, highlighting the gap between current solutions and the demands of home-based health management, which motivates the need for our proposed dataset and model.
> We hope that the additional results and analyses help clarify this point and address the reviewer's concern.

---

> ### Author Response · Authors · 2025-11-21
> **Part two of the response to Reviewer  QoBN**
>
> *Review W2: It is still not fully clear to me that how the DIY healthcare scenario becomes different from other healthcare applications [3]. While paying notable attention to the method and data, the authors did not fully demonstrate the uniqueness of the setting, obscuring the motivation and undermining the evaluation of this study.*
>
>
> Response: Thank you for the comments and the reference.
> The suggested work [3] proposes a Med-MLLM benchmark to assess models across distinct medical specialties
> While DIYHealth (i.e., home care) settings share the overarching goal of supporting health management, they operate under a different set of characteristics compared with conventional clinical scenarios.
>
> In DIYHealth settings, health management occurs in non-clinical, user-driven contexts, where consumer-grade data sources, personalization modeling needs, and heterogeneous task formulations introduce distinct constraints that are less represented in existing healthcare benchmarks.
> Rather than contrasting with clinical paradigms, DIYHealth complements them by addressing everyday health management needs that arise outside medical institutions and remain underexplored in current research.
>
>
> **Non-clinical data sources.**
> Home care data are primarily collected via consumer-grade smartphones, wearables, and home sensors, or through self-reported information. Compared to clinical imaging or EHR data, these signals are considerably more heterogeneous and noisy. This difference motivates the need for home care benchmarks.
>
> **Lack of existing large-scale datasets.**
> Prior works focus primarily on clinical imaging or structured EHR datasets. To our knowledge, large-scale, multimodal, user-centric datasets specifically curated for home care scenarios remain largely unexplored, which is precisely what our dataset and benchmark contribute. Although DIYHealth-900K is constructed from publicly available sources that may include data collected in controlled environments, we have systematically processed and reorganized it to reflect home care settings, and we have further developed a healthcare data engine to ensure improved reliability and safety.
>
>
>
> **Strong personalization requirement.**
> In clinical applications, models typically operate at the population level. In contrast, the home care setting demands instance-level personalization because individual health baselines vary widely. This need for continual personalization is a core driver of our design of H$^2$LoRA.
>
> **Task breadth and fragmentation.**
> In home care contexts, users frequently face diverse everyday health concerns (e.g., skin conditions, sleep issues, food choices, symptom triage, chronic disease monitoring), yet no unified benchmark exists to evaluate model performance across this wide task range. Our DIYHealth Suite explicitly addresses this gap by providing a large, standardized, multimodal benchmark covering 11 common home care scenarios.
>
>
> *Review Q1: I notice that although the authors have created a repo for the codes, the data are still unavailable. I am personally curious whether and how the authors would like to publicize the data.*
>
> Response: We appreciate the reviewer's interest in the data and fully recognize the importance of open-sourcing for community use. We have provided a representative subset of DIYHealth-900K, in the anonymous GitHub repository at https://anonymous.4open.science/r/DIYHealthGPT-codes-E71A for review.
> Due to the large scale of the dataset and storage limits in the anonymous review repository setting, uploading the full dataset was impractical at this stage.
> Upon acceptance, we will release the full dataset publicly on Hugging Face, together with detailed documentation and data cards to support community use.
>
> Thank you again for your positive feedback and insightful comments. We hope our clarifications and new results have addressed your concerns.

---

> ### Author Response · Authors · 2025-11-21
> **Part three of the response to Reviewer QoBN**
>
> *References:*
>
> [1]. Chen, Zeren, et al. "Octavius: Mitigating Task Interference in MLLMs via LoRA-MoE." The Twelfth International Conference on Learning Representations.
>
> [2]. Wang, Xiaochen, et al. "FEDKIM: Adaptive Federated Knowledge Injection into Medical Foundation Models." Proceedings of the 2024 Conference on Empirical Methods in Natural Language Processing. 2024.
>
> [3]. Liu, Jie, et al. "Asclepius: A spectrum evaluation benchmark for medical multi-modal large language models." Proceedings of the 63rd Annual Meeting of the Association for Computational Linguistics (Volume 1: Long Papers). 2025.
>
> [4] Lin, Tianwei, et al. "HealthGPT: A Medical Large Vision-Language Model for Unifying Comprehension and Generation via Heterogeneous Knowledge Adaptation." Forty-second International Conference on Machine Learning.
>
> [5] Xie, Qianqian, et al. "PIXIU: a large language model, instruction data and evaluation benchmark for finance." Proceedings of the 37th International Conference on Neural Information Processing Systems. 2023.
>
> [6] Wu, Yutong, et al. "InverseCoder: Self-improving Instruction-Tuned Code LLMs with Inverse-Instruct." Proceedings of the AAAI Conference on Artificial Intelligence. Vol. 39. No. 24. 2025.

---

### Official Review · Reviewer_4ZVG · 2025-11-01

**Soundness:** 3
**Presentation:** 3
**Contribution:** 3
**Rating:** 6
**Confidence:** 3

**Summary:**

To solve the problem that most medical multimodels are built for hospitals. The paper introduces DIYHealth Suite, which includes DIYHealth-900K, DIYHealthGPT, and DIYBench. Research shows an average of 22.7% improvement in ACC under closed-QA and 16.7% in F1-Bio under open-QA. In addition, DIY HealthGPT receives the most Rank-1 preferences for conciseness, correctness, relevance, and actionability.

**Strengths:**

1. Colorful and clear diagrams that are helpful for comprehension.
2. This paper is well written.
3. It achieves state-of-the-art performance across all home care tasks, exceeding the robust general and medical LVLM baseline by an average of 22.7% in closed-ended question accuracy and by 16.7% in open-ended question F1-Bio accuracy.
4. This research is interesting and has clear application potential. It is built for real home settings using consumer-grade phones and wearables, covers 11 diverse tasks, delivers personalized and actionable guidance, and is clinician-validated, making it directly usable for at-home health management.

**Weaknesses:**

1. Most training labels are machine-generated. Only 10% of labels are audited by humans, which could lead to the model learning the wrong things.
2. The paper does not provide real user feedback for assessing usability.
3. The paper does not compare performance across subgroups such as gender, sex, race or ethnicity, age, skin tone, primary language, device type, and lighting conditions. Without subgroup checks, hidden disparities may persist, and some users could receive less accurate or less safe guidance. This weakens claims of real-world reliability and raises deployment risk.

**Questions:**

See weaknesses.

---

> ### Author Response · Authors · 2025-11-21
> **Part one of the response to Reviewer 4ZVG**
>
> We thank you for your positive review and constructive feedback.
> We address your concerns and provide the new results below as well as in the revised manuscript.
>
> *Review W1: Most training labels are machine-generated. Only 10\% of labels are audited by humans, which could lead to the model learning the wrong things.*
>
> Response: We clarify that all labels in DIYHealth-900K are derived from the original datasets and are strictly preserved without alteration.
> Importantly, LLMs were not used to generate labels in DIYHealth-900K. Their role is purely linguistic refinement, such as rewriting questionnaires into natural human languages, which is similar to recent medical LLMs practices [1].
>
> Furthermore, we design a Health Data Engine with human-in-the-loop verification to ensure reliability, where 10\% of the generated QA pairs are inspected by human reviewers focusing on **semantic validity, medical consistency, and format standardization**.
> Each entry undergoes two rounds of independent review, providing fine-grained data quality control and ensuring reliability [2].
> Therefore, although only a subset was explicitly audited, the core annotations are fully preserved from the original datasets, and the human audit demonstrates the robustness of our processing pipeline.
>
> *Review W2: The paper does not provide real user feedback for assessing usability.*
>
> Response: Thank you for raising this important point.
> We have conducted a human-centered user feedback study with 36 participants spanning diverse characteristics, such as age, gender, and self-rated health status.
> This study comprises two components: answer-level evaluation and system-level evaluation. All items are assessed using a
> 5-point scale, where 1 indicates strongly disagree and 5 indicates strongly agree.
> We summarize the results in the Tables below. The results are also presented in Figure 11 of Appendix E.4.
>
>
> **Answer-level Evaluation**
>
> | Metric        | Mean Score | Std. Error |
> |---------------|------------|------------|
> | Clarity       | 4.39       | 0.19       |
> | Usefulness    | 4.09       | 0.32       |
> | Conciseness   | 4.08       | 0.21       |
> | Safety        | 4.05       | 0.34       |
> | Willingness   | 3.96       | 0.34       |
> | Trust         | 3.94       | 0.34       |
>
> **System-level Evaluation**
>
> | Question | Mean Score | Std. Error |
> |----------|------------|------------|
> | Q1       | 4.31       | 0.70       |
> | Q2       | 4.63       | 0.62       |
> | Q3       | 4.75       | 0.45       |
> | Q4       | 4.31       | 0.79       |
> | Q5       | 4.31       | 0.48       |
> | Q6       | 4.56       | 0.51       |
> | Q7       | 4.63       | 0.50       |
> | Q8       | 4.69       | 0.60       |
> | Q9       | 4.06       | 0.68       |
> | Q10      | 4.56       | 0.51       |
>
> For **answer-level evaluation**, participants are asked to rate the quality of the model-generated answers from DIYHealthGPT across six dimensions: clarity, usefulness, conciseness, safety, willingness to follow, and trustworthiness. All six dimensions receive average scores close to or above 4.0, suggesting that the generated answers are not only considered reliable and contextually appropriate for practical use but also well aligned with users' expectations in real-world home care scenarios.
>
> For **system-level evaluation**, we employ the System Usability Scale [3], a widely used standardized evaluation for assessing system usability. DIYHealthGPT achieves a SUS score of 87.03. According to established interpretation guidelines [1,4], this score falls in the excellent usability, indicating that DIYHealthGPT is not only effective in generating answers but also user-friendly and has strong potential for practical deployment.
>
> We have incorporated the corresponding results and analysis in Appendix E.4 of the revised paper.

---

> ### Author Response · Authors · 2025-11-21
> **Part two of the response to Reviewer 4ZVG**
>
> *Review W3: The paper does not compare performance across subgroups such as gender, sex, race or ethnicity, age, skin tone, primary language, device type, and lighting conditions. Without subgroup checks, hidden disparities may persist, and some users could receive less accurate or less safe guidance. This weakens claims of real-world reliability and raises deployment risk.*
>
> Response: To assess the robustness of DIYHealthGPT across different subgroups and to support safe deployment in healthcare scenarios, we conducted a preliminary subgroup analysis. Due to dataset de-identification and the nature of home-care data acquisition, demographic attributes such as race, ethnicity, skin tone, and device type are not consistently annotated, which makes comprehensive subgroup evaluation infeasible at this stage. Therefore, we focus on open-QA tasks where relevant metadata are available and perform subgroup analyses on (i) **age and gender subgroup analysis** for the CKD and Heart tasks, and (ii) **BMI and vegetable-consumption level subgroup analysis** for the Obesity task. The results are presented in the table below.
>
>
> **Subgroup results for CKD and heart tasks.**
>
> | Subgroup         | Size |   CKD  |       |  Heart |       |
> |------------------|:----:|:------:|:-----:|:------:|:-----:|
> |                  |      | F1-Bio |   RL  | F1-Bio |   RL  |
> | Age $<$ 18       |   7  |  93.08 | 72.45 |  81.50 | 29.67 |
> | 18$\leq$Age$<$40 |  134 |  90.51 | 65.24 |  83.05 | 34.38 |
> | 41$\leq$Age$<$65 |  212 |  91.64 | 68.32 |  82.54 | 32.95 |
> | Age $\geq$ 65    |  158 |  90.06 | 62.13 |  82.47 | 33.42 |
> | Gender: Male     |  193 |  89.64 | 61.49 |  82.47 | 32.90 |
> | Gender: Female   |  205 |  89.93 | 62.18 |  82.67 | 33.62 |
>
>
> **Subgroup Results for the Obesity Task.**
>
> | Subgroup            | Size | Obesity |       |        |       |
> |---------------------|:----:|:-------:|:-----:|:------:|:-----:|
> |                     |      |  F1-Bio |   RL  | F1-Rad |  BLEU |
> | BMI $<$ 18.5        |  36  |  85.66  | 43.11 |  34.62 | 52.67 |
> | 18.5$\leq$BMI$<$ 25 |  57  |  84.77  | 41.37 |  35.64 | 53.97 |
> | 25$\leq$BMI$<$30    |  98  |  84.34  | 45.81 |  26.95 | 52.13 |
> | BMI$\geq$30        |  166 |  85.20  | 47.15 |  32.89 | 55.14 |
> | Vegetable: 1        |  44  |  85.65  | 49.07 |  32.23 | 55.68 |
> | Vegetable: 2        |  201 |  84.84  | 45.73 |  31.92 | 54.31 |
> | Vegetable: 3        |  112 |  84.85  | 43.41 |  31.64 | 52.97 |
>
>
> Across all subgroups, **the performance remains stable, with fluctuations within a small margin.** For instance, the variation in F1-Bio across different age subgroups is within 3%, and the differences between male and female subgroups are below 1%.
>
> However, there are two exceptions. Regarding the large variation in the RL metric for the Age $<18$ subgroup and the Age $\ge$ 65 subgroup (72.45 vs. 62.13), we note that the Age $<18$ subgroup contains only seven samples, which makes the results statistically unstable and sensitive to individual cases. Therefore, the difference is more likely attributable to insufficient cohort size rather than systematic model bias. Besides, although the F1-Rad for the 25 $\le$ BMI $< 30$ subgroup is much lower than that for the 18.5 $\le$ BMI $< 25$ subgroup (26.95 vs. 35.64), this variation is consistent with clinical expectations rather than indicating model bias: individuals in the overweight subgroup (25~$\le$~BMI~$<$~30) often present subtle or non-specific symptoms, making their cases harder to detect. Since F1-Rad is highly sensitive to biomedical terminology, such cases naturally require stronger reasoning and interpretation ability. This suggests an opportunity for future personalized adaptation, rather than revealing a systematic risk.
>
>
> Overall, the subgroup results indicate that DIYHealthGPT generally maintains consistent performance across different user populations, supporting its potential for real-world deployment. The results and analysis have been added in Appendix E.5.
>
> Thank you again for your encouraging feedback and thoughtful suggestions.
> We hope our responses have addressed your concerns and helped further clarify the design and potential of our work.
>
>
> *References:*
>
> [1] Luo, Renqian, et al. "BioGPT: generative pre-trained transformer for biomedical text generation and mining." Briefings in bioinformatics 23.6 (2022): bbac409.
>
> [2] Rashkin, Hannah, et al. "Measuring attribution in natural language generation models." Computational Linguistics 49.4 (2023): 777-840.
>
> [3] Brooke, John. "SUS-A quick and dirty usability scale." Usability evaluation in industry 189.194 (1996): 4-7.
>
> [4] Bangor, Aaron, Philip Kortum, and James Miller. "Determining what individual SUS scores mean: Adding an adjective rating scale." Journal of usability studies 4.3 (2009): 114-123.

---

> > ### Comment · Reviewer_4ZVG · 2025-11-26
> >
> > Thank you for the clarification. I will maintain my positive scores of six.

---

> > > ### Author Response · Authors · 2025-11-27
> > >
> > > Thank you for your time and constructive feedback. We appreciate your positive evaluation and will continue to advance this direction toward meaningful impact in healthcare.

---

### Official Review · Reviewer_srqP · 2025-11-09

**Soundness:** 3
**Presentation:** 3
**Contribution:** 4
**Rating:** 4
**Confidence:** 4

**Summary:**

The paper investigates the problem of home care and medicine access through a multimodal question-answering framework that integrates both textual and imaging data. It presents three primary contributions: (1) the creation of a large-scale dataset, (2) the development of a GPT-based model designed for this domain, and (3) benchmark results evaluated on a curated test set.

**Strengths:**

- The paper addresses an important gap in current research related to automated medical assistance in home care settings.
- It is comprehensive in scope, combining dataset development, model design, and evaluation.
- The proposed model is distinctive in its integration of task-level and instance-level cross-information, which enhances multimodal reasoning capabilities.

**Weaknesses:**

- Figure 3 does not accurately reflect the corresponding mathematical formulation presented in the text.
- The description of the publicly available datasets—collected from 20 open sources—is missing from the main paper.
- It is unclear whether the baseline models were evaluated in a zero-shot setting. If so, this would constitute an unfair comparison with the proposed model, which is fine-tuned with task-specific data.

**Questions:**

- How can matrix A be described as a shared projection if a separate one exists for each task?
- There is ambiguity around Equation (4) and the explanation on line 286; these should be clarified.
- The statement “We sample 10% of training data per task”—is this sampling performed in each epoch or only once during training?
- In Stage 3, a dedicated H2LoRA block per task is mentioned, please elaborate on its implementation.
- DIYHealthBench is derived from the designated DIYHealth-900K test set and reportedly includes 12,167 carefully sampled examples. How were these examples selected, and according to what criteria?
- Were hyperparameter tuning procedures conducted, and if so, what methodology was followed?
- What is the loss function used during training?
- Please specify the model size and architectural parameters.
- How were prompts designed for both training and evaluation?
- What temperature settings were used for the baseline models during inference?
- How was interrater agreement assessed, and what was the resulting Cohen’s kappa score?

---

> ### Author Response · Authors · 2025-11-21
> **Part one of the response to Reviewer srqP**
>
> We sincerely appreciate your insightful and detailed comments. Below, we provide a point-by-point response to each comment, together with the corresponding revisions in the paper (highlighted in blue).
>
> *Review W1: Figure 3 does not accurately reflect the corresponding mathematical formulation presented in the text.*
>
> Response: Thank you for pointing this out.
> We have revised Figure 3 to include consistent notation and clearer annotations, ensuring precise alignment with the mathematical formulation in the text.
> We also refined the corresponding description to maintain full coherence between the figure and the formulation.
>
> To further improve clarity, we briefly highlight the core data flow in the revised figure: images are encoded by the visual encoder $\mathcal{E}_v$, user queries are processed by the textual encoder $\mathcal{E}_t$, and a learnable projector $\mathcal{P}_v$ maps visual features into the textual embedding space.
>
> The concatenated embeddings $\mathcal{Z}$ are then fed into the backbone model $\mathcal{M}_{\text{LLM}}$, where H$^2$LoRA enables efficient parameter sharing and task-specific personalization through two complementary mechanisms:
> (i) Low-Rank Expert Mixture, which employs a shared matrix $\mathbf{A}^t$ with a set of $K$ expert matrices $\{\mathbf{B}^t_1,\ldots,\mathbf{B}^t_K\}$ for parameter efficiency; and
> (ii) Hyper LoRA Adaptation, which adopts a similar layout but generates its parameter offsets $\Delta \mathbf{A}^t$ and $\Delta \mathbf{B}^t_k$ through hypernetworks $\mathcal{H}_A$ and $\mathcal{H}_B$ to model task-aware, instance-specific variations for personalization.
>
> &nbsp;
>
> *Review W2: The description of the publicly available datasets—collected from 20 open sources—is missing from the main paper.*
>
> Response: As suggested, we have incorporated a clear description of the data sources into Section 3.1 (lines 152 - 155) and further refined the details in Appendix C.2 for completeness and accessibility.
>
> Specifically, we construct DIYHealth-900K through a systematic curation and adaptation process based on multimodal data from 20 publicly available data sources across Kaggle, PhysioNet, and Figshare. These sources cover a broad spectrum of data types, ranging from patient demographics, vital signs, laboratory test results, medical notes, and discharge summaries to medical conversations, patient questionnaires, commonsense knowledge, and imaging collected from diverse devices.
> To ensure applicability to realistic home-care settings, the heterogeneous datasets are carefully aligned and adapted with an emphasis on multimodal inputs, real-world variability, and non-clinical supervision.
> Full details are provided in Appendix C.2.

---

> ### Author Response · Authors · 2025-11-21
> **Part two of the response to Reviewer srqP**
>
> *Review W3: It is unclear whether the baseline models were evaluated in a zero-shot setting. If so, this would constitute an unfair comparison with the proposed model, which is fine-tuned with task-specific data.*
>
> Response: Our evaluation follows the standard protocol adopted in recent LLM benchmark studies [1,2,3], where baseline models are evaluated in their default inference settings without further adaptation to our dataset.
> This setup is intentional and aligned with our objective: to examine whether current generalist and medical-specific foundation models can effectively operate in realistic home-care scenarios. Our experimental results show they largely struggle in this setting.
> This highlights **the gap between existing models and the demands of home-based health management**, motivating the need for our proposed dataset and model.
>
> To further analyze the source of performance gain and ensure fairness, we have additionally fine-tuned two representative baselines, Gemma 3-4B and LLaVA-Med v1.5-7B, one generalist and one medical-specific model, using the same DIYHealth-900K training data. As shown in the table below, DIYHealthGPT-3.8B consistently outperforms Gemma 3-4B and LLaVA-Med v1.5-7B by a substantial margin on the four evaluation metrics under both closed-QA and open-QA settings, despite Gemma 3-4B and LLaVA-Med v1.5-7B  having larger model sizes. These results indicate that the observed improvements cannot be attributed only to the dataset. Instead, the joint effect of the H$^2$LoRA architecture, training strategy, and curated DIYHealth-900k contributes to the model's effectiveness in home-care scenarios.
>
> | Model             | Closed-QA |           |  Open-QA  |           |
> | ----------------- | :-------: | :-------: | :-------: | :-------: |
> |                   |    ACC    |    MCC    |  F1-Bio   |    RL     |
> | Gemma 3-4B        |   80.96   |   74.42   |   84.72   |   41.59   |
> | LLaVA-Med v1.5-7B |   77.58   |   69.51   |   86.28   |   49.63   |
> | DIYHealthGPT-3.8B | **86.80** | **82.36** | **87.34** | **52.11** |
>
> We have added the results and further discussions in Section 6.3 (lines 486–497) of the main paper to clarify this point.
> We believe this expanded evaluation clarifies the experimental setting and provides deeper insight into the challenges of adapting existing models to home-based health management tasks.
>
> &nbsp;
>
> *Review Q1: How can matrix A be described as a shared projection if a separate one exists for each task?*
>
> Response: We clarify that the matrix $\mathbf{A}^t$ is not shared across tasks. Instead, it is shared across a set of $K$ expert matrices within a single task $t$, i.e., $\{\mathbf{B}^t_1,\ldots,\mathbf{B}^t_K\}$, serving as a common projection for that task.
> In standard LoRA, each low-rank adapter consists of one pair of matrices ($\mathbf{A}$, $\mathbf{B}$).
> In our design, using one shared matrix $\mathbf{A}^t$ with multiple $\mathbf{B}^t_k$ matrices within the same task provides implicit regularization, enforces a consistent low-dimensional subspace, and improves both memory and computational efficiency.
>
> During the training of Stage 4, where soft-MoE is employed for multi-task training, $\mathbf{A}^t$ acts as a task-specific anchor to encourage cross-task knowledge transfer and $\mathbf{B}^t_k$ maintains the flexibility for task-level specialization. We have enhanced the corresponding descriptions in Section 4.2 (lines 262 - 266) for clarity.
>
> &nbsp;
>
> *Review Q2: There is ambiguity around Equation (4) and the explanation on line 286; these should be clarified.*
>
> Response: We have revised Equation (4) and refined the accompanying explanations in Section 4.2 (lines 244 - 248 and lines 274 - 275) to eliminate ambiguity and ensure clarity and notation consistency.
> Specifically, given the unified multimodal embedding $\mathcal{Z}$, the backbone model produces its output conditioned on the frozen backbone parameters $\Theta$.
> H$^2$LoRA has task-adaptive parameters denoted as $\Theta_{H^2L}$, where the core components include the task-specific low-rank parameters $\mathcal{A} = \{\mathbf{A} ^ t,\Delta\mathbf{A} ^ t\}_{t=1} ^ N$ and $\mathcal{B} = \{\mathbf{B} ^ t,\Delta\mathbf{B} ^ t\} _ {t=1} ^ N$, and the routing parameters $\mathcal{R}$ for inter-task knowledge fusion.
>
> For each task $t$, a H$^2$LoRA block generates an adaptation output $\mathcal{O} ^ t _ {H ^ 2LoRA}=\mathcal{Z}\mathbf{A} ^ {t}\mathbf{B} ^ {t}+\mathcal{Z}\Delta\mathbf{A} ^ {t}\Delta\mathbf{B} ^ {t}$.
> The routing module assigns weights $(\beta ^ 1, \ldots, \beta ^ N)=\mathcal{R}(\mathcal{Z})$ to each task-specific block, and the final model output is given by: $\mathcal{O} _ {H ^ 2LoRA}=\mathcal{M} _ {\text{LLM}}(\mathcal{Z}; \Theta, \Theta _ {H ^ 2L})=\mathcal{Z}\Theta+\sum _ {t=1} ^ N\beta ^ t\mathcal{O} ^ t _ {H ^ 2LoRA}$.

---

> > ### Author Response · Authors · 2025-11-21
> > **Part three of the response to Reviewer srqP**
> >
> > *Review Q3: The statement “We sample 10\% of training data per task”—is this sampling performed in each epoch or only once during training?*
> >
> > Response: The sampling is performed once prior to training. We randomly sample 10\% of the training data per task to construct a fixed subset, which remains unchanged throughout the training process. This design ensures consistent supervision across epochs.
> > To enhance clarity and avoid misunderstanding, we have refined the training setting in Section 4.4 (lines 313 - 315).
> >
> > &nbsp;
> >
> > *Review Q4: In Stage 3, a dedicated H2LoRA block per task is mentioned, please elaborate on its implementation.*
> >
> > Response: We have further clarified the implementation of training Stage 3 in Appendix D.1 (lines 1179 - 1190).
> > Specifically, during task-specific expert training in Stage 3, each task $t$ is assigned an H$^2$LoRA block with parameters $(\mathbf{A}^t,\Delta\mathbf{A}^t,\mathbf{B}^t,\Delta\mathbf{B}^t)$,
> > which are trained independently using task-specific data, while other parameters remain frozen. Each block is optimized using the same hyperparameter settings across tasks.
> > The Low-Rank
> > Expert Mixture and Hyper LoRA Adaptation modules within the H$^2$LoRA block are inserted into the linear layers of the backbone LLM as lightweight training layers. Low-Rank Expert Mixture is implemented by a shared matrix $\mathbf{A}^t$ and multiple matrices $\mathbf{B}^t_k$, with a router that aggregates the output of matrices $\mathbf{B}^t_k$. Hyper LoRA follows a similar structure, but its parameters are dynamically generated by hypernetworks $\mathcal{H}_A$ and $\mathcal{H}_B$ rather than learned directly through backward propagation. The hypernetworks learn how to generate parameters conditioned on the instance input.
> > Overall, this implementation allows the model to acquire task-level specialization and parameter-efficient adaptation before cross-task fusion in Stage 4.
> >
> > &nbsp;
> >
> > *Review Q5: DIYHealthBench is derived from the designated DIYHealth-900K test set and reportedly includes 12,167 carefully sampled examples. How were these examples selected, and according to what criteria?*
> >
> > Response: We adopt a stratified sampling strategy that randomly samples 1\% of the data from each task for evaluation, following prior large-scale benchmark setups [4]. To ensure fair task representation and statistical reliability, we apply a lower bound of 1,000 examples for tasks with relatively small data volume (e.g., Drug Recommendation and Kidney Health). This leads to a total of 12,167 test examples across 11 tasks, providing a consistent and computationally feasible benchmark for evaluating multi-task performance.
> > We have supplemented the corresponding details in Section 5 (lines 332 - 335).
> >
> > &nbsp;
> >
> > *Review Q6: Were hyperparameter tuning procedures conducted, and if so, what methodology was followed?*
> >
> > Response: To ensure fairness and consistency, the hyperparameters of DIYHealthGPT (e.g., learning rate, batch size, max sequence length) are set up according to prior studies [1,5,6] without additional hyperparameter tuning, and the detailed configurations are provided in Appendix D.3.
> > All baseline models are evaluated under the configurations provided by the respective authors.
> > Following the baseline model HealthGPT, we further conduct a sensitivity analysis on the core hyperparameter: the number of experts in H$^2$LoRA, with the results presented in Appendix E.3.
> >
> > &nbsp;
> >
> > *Review Q7: What is the loss function used during training?*
> >
> > Response: DIYHealthGPT is optimized using the next-token prediction objective with the cross-entropy loss function [5]. Specifically, the model predicts the next token in the output sequence conditioned on the multimodal input and preceding tokens.
> > As recommended, we have clarified the loss function used during training in Appendix D.1 (lines 1177 - 1178).

---

> ### Author Response · Authors · 2025-11-21
> **Part four of the response to Reviewer srqP**
>
> *Review Q8: Please specify the model size and architectural parameters.*
>
> Response: Our model consists of a 3.8B backbone, a 0.3B CLIP vision encoder, and a 0.7B H$^2$LoRA adaptation module. Following prior multimodal LVLM studies [1,7], we report the backbone size in the main experiments for consistency.
>
> As suggested, we have added Table 9 in Appendix D.3 to display the architectural parameters, and we also include it below for reference.
>
> | Module                      | Parameter                              | Value / Shape                                 |
> | --------------------------- | -------------------------------------- | --------------------------------------------- |
> | Tokenizer                   | Vocabulary size                        | 32,064                                        |
> | Backbone                    | Hidden size                            | 3,072                                         |
> |                             | \# Transformer blocks                  | 32                                            |
> |                             | Self-attn projections                  | qkv: (3072, 9216), o: (3072, 3072 )           |
> |                             | MLP                                    | gate up: (3072, 16384); down: (8192, 3072) |
> | CLIP                        | Input resolution                       | $336 \times 336$                              |
> |                             | Patch size                             | $14 \times 14$                                |
> |                             | Embedding dim                          | 1,024                                       |
> |                             | \# Transformer blocks                  | 24                                            |
> | Expert Mixture of H$^2$LoRA | LoRA rank                              | 16                                            |
> |                             | Shared matrix $\mathbf{A}$ (up / down) | (3072,16) / (8192,16)                     |
> |                             | Matrix $\mathbf{B}$ (up / down)        | (16,16384) / (16,3072)                     |
> | Hyper LoRA of H$^2$LoRA     | Rank                                   | 8                                             |
> |                             | Input                                  | (3072, 8)                                     |
> |                             | Up-path generator (down / up)          | (8, 24576) / (8, 131072)                  |
> |                             | Down-path generator (down / up)        | (8, 65536) / (8, 24576)                   |
>
> The tokenizer supports 32,064 words, while the hidden size of the backbone is 3,072 and self-attention projections of dimensions $(3072,9216)$ and $(3072,3072)$ for query–key–value and output embeddings, respectively. Each transformer block contains a feed-forward module with expansion and compression layers of $(3072,16384)$ and $(8192,3072)$, providing strong representational capacity. The CLIP encoder processes images of $336\times336$ resolution with a patch size of $14\times14$, yielding 1,024-dimensional visual embeddings through 24 transformer blocks.
>
> On top of the backbone, we introduce H$^2$LoRA adapters to enable parameter-efficient and task-adaptive learning. In the LoRA Expert Mixture of H$^2$LoRA, the shape of the shared matrix $\mathbf{A}^t$ for the up-projection is $(3072,16)$ and $(8192,16)$ for the down-projection, where up-projection and down-projection are two linear layers of the transformer block. The corresponding $\mathbf{B}^t_k$ matrices have shapes (16, 16384) and (16, 3072), respectively.
> The Hyper LoRA component further improves adaptability through dynamic parameter generation: a hypernetwork first encodes input representations via a $(3072,8)$ projection, then produces instance-specific weights through generators with $(8,24576)$ / $(8,131072)$ for the up-projection and $(8,65536)$ / $(8,24576)$ for the down-projection.
>
> &nbsp;
>
> *Review Q9: How were prompts designed for both training and evaluation?*
>
> Response: We use an identical prompt format for both training and evaluation to ensure consistency and fair comparisons.
> For each task, the input query is used directly as the input prompt.
> We present the example input query for each home care task in Figures 13-33 of the appendix.
> For example, in the Diabetes task under the open-QA setting (Figure 26), the input prompt is:
> "What immediate clinical recommendations would you suggest for a person with a retinal image presenting these characteristics?"
>
> To maintain neutrality and reproducibility, we adopt the standard system prompt used in prior work [6]:
> "You are a helpful AI assistant."
> This system prompt is kept identical across all tasks and stages of training and evaluation.
>
> This unified prompt strategy ensures reproducibility and avoids introducing task-specific prompt engineering that might bias the evaluation.

---

> > ### Author Response · Authors · 2025-11-21
> > **Part five of the response to Reviewer srqP**
> >
> > *Review Q10: What temperature settings were used for the baseline models during inference?*
> >
> > Response: For all baseline models, we set the temperature during inference to $0$, ensuring deterministic and fully reproducible outputs. This follows common practice in prior studies [1,7], and allows a fair comparison by eliminating variability introduced by stochastic decoding.
> >
> > &nbsp;
> >
> > *Review Q11: How was interrater agreement assessed, and what was the resulting Cohen’s kappa score?*
> >
> > Response: We appreciate the reviewer's focus on the interrater agreement.
> > To ensure the reliability of clinical expert review for different models, we conduct a comprehensive inter-rater agreement analysis based on the rankings provided by five independent raters. Each question consists of six model-generated outputs, and raters are asked to assign an ordinal rank from 1 to 6. As the annotations follow an ordinal preference-ranking scheme, we assess the degree of agreement using the soft agreement criterion [8,9] rather than strict rank matching. Two raters are considered to be in agreement on a given answer if the absolute difference between their assigned ranks is no larger than one, i.e.,$\lvert r_1$ – $r_2 \rvert$$\le$ 1.
> >
> > Under this soft agreement criterion, we compute Cohen’s kappa score to quantify inter-rater agreement.
> > We summarize the range of Cohen’s kappa scores in the table below, and present the detailed results in Figure 12 in Appendix E.7.
> >
> > | Task                | Personalized Health Management | Chronic Disease Risk Assessment | Daily Health Monitoring |     Overall    |
> > |---------------------|:------------------------------:|:-------------------------------:|:-----------------------:|:--------------:|
> > | Cohen’s kappa score |         [0.692, 0.846]         |          [0.663, 0.839]         |      [0.585, 0.830]     | [0.645, 0.808] |
> >
> > Overall, according to the widely adopted interpretation guideline by Landis and Koch [10], the majority of kappa values fall within the substantial agreement range [0.61, 0.80], indicating a strong level of reliability in expert judgments.
> > We have included the detailed inter-rater agreement analysis in Appendix E.7.
> >
> > &nbsp;
> >
> > We are grateful for the chance to discuss our work's potential and wish to thank you again for your valuable input.
> > We hope our clarifications and new results have addressed your concerns and can improve your evaluation of our work.
> >
> > *References:*
> >
> > [1] Lin, Tianwei, et al. "HealthGPT: A Medical Large Vision-Language Model for Unifying Comprehension and Generation via Heterogeneous Knowledge Adaptation." Forty-second International Conference on Machine Learning.
> >
> > [2] Xie, Qianqian, et al. "PIXIU: a large language model, instruction data and evaluation benchmark for finance." Proceedings of the 37th International Conference on Neural Information Processing Systems. 2023.
> >
> > [3] Wu, Yutong, et al. "InverseCoder: Self-improving Instruction-Tuned Code LLMs with Inverse-Instruct." Proceedings of the AAAI Conference on Artificial Intelligence. Vol. 39. No. 24. 2025.
> >
> > [4] Wang, Alex, et al. "GLUE: A multi-task benchmark and analysis platform for natural language understanding." Proceedings of the 2018 EMNLP workshop BlackboxNLP: Analyzing and interpreting neural networks for NLP. 2018.
> >
> > [5] Liu, Haotian, et al. "Visual instruction tuning." Advances in neural information processing systems 36 (2023): 34892-34916.
> >
> > [6] Li, Chunyuan, et al. "Llava-med: Training a large language-and-vision assistant for biomedicine in one day." Advances in Neural Information Processing Systems 36 (2023): 28541-28564.
> >
> > [7] Li, Sijing, et al. "Eyecaregpt: Boosting comprehensive ophthalmology understanding with tailored dataset, benchmark and model." Proceedings of the 33rd ACM International Conference on Multimedia. 2025.
> >
> > [8] Stemler, Steven E. "A comparison of consensus, consistency, and measurement approaches to estimating interrater reliability." Practical Assessment, Research, and Evaluation 9.1 (2004).
> >
> > [9] Fu, Lianyan, et al. "On modelling agreement and category distinguishability on an ordinal scale." Communications in Statistics-Theory and Methods 41.24 (2012): 4413-4426.
> >
> > [10] Landis, J. Richard, and Gary G. Koch. "The measurement of observer agreement for categorical data." biometrics (1977): 159-174.

---

### Author Response · Authors · 2025-12-02
**Rebuttal Summary**

Dear Reviewers, Area Chairs, Senior Area Chairs, and Program Chairs,

We sincerely thank all chairs and reviewers for their time, constructive comments, and valuable feedback. We also appreciate the opportunity to engage in the rebuttal. During this stage, we actively addressed all concerns by clarifying misunderstandings, providing detailed technical explanations, and conducting additional experiments. Below, we summarize our contributions and highlight our efforts in addressing the reviewers’ feedback.

Home-based health management holds substantial promise for enabling anytime personalized health support beyond clinical environments. However, it faces three core challenges that remain largely underexplored:
(i) The lack of standardized large-scale datasets derived from consumer-grade devices and self-reported inputs.
(ii) Highly diverse personal health baselines and continuously evolving individual conditions.
(iii) The absence of a unified benchmark for assessing model performance across a wide range of home-care tasks.

To tackle these challenges, we propose **DIYHealth Suite**, a comprehensive ecosystem comprising DIYHealth-900K, DIYHealthGPT, and DIYHealthBench. Our key contributions are summarized as follows, with recognition from reviewers.

- **DIYHealth-900K**, a large-scale multimodal dataset curated from everyday devices to capture the complexity and variability of real-world home care.

    *(Recognized by Reviewer `srqP`, `4ZVG`, `r2KA`)*
- **DIYHealthGPT**, an adaptive foundation model powered by the proposed H$^2$LoRA mechanism, enabling personalized representations while maintaining robust generalization.

    *(Recognized by Reviewer `srqP`, `4ZVG`, `r2KA`)*
- **DIYHealthBench**, the first unified benchmark for evaluating foundation models in non-clinical settings, covering tasks from daily health monitoring to chronic disease risk assessment and personalized health management.

    *(Recognized by Reviewer `srqP`, `4ZVG`, `QoBN`, `r2KA`)*
- **Extensive experiments** on DIYHealthBench demonstrate that DIYHealthGPT consistently outperforms state-of-the-art generalist and medical-specific baselines across diverse home-care tasks.

    *(Recognized by Reviewer `srqP`, `4ZVG`, `QoBN`)*

Following the reviewers’ feedback, we have rigorously addressed all concerns through **well-structured clarifications and additional experiments that further validate our contributions**. All corresponding revisions have been updated in the manuscript (highlighted in blue).

- **Clarifications**. We refine the descriptions of datasets, clarify training and experimental settings, and provide detailed implementations and architectural parameters of DIYHealthGPT.
- **Additional experiments**. We conduct additional experiments for a thorough evaluation of DIYHealthGPT.
  - **Inter-rater Agreement Analysis** to assess the reliability of clinical expert evaluations.

    *(Suggested by Reviewer `srqP`)*
  - **User Feedback Analysis** to examine usability in real-world scenarios.

    *(Suggested by Reviewer `4ZVG`)*
  - **Subgroup Study** to investigate model robustness across different populations.

    *(Suggested by Reviewer `4ZVG`)*
  - **Comparison with Fine-tuned Baselines** to validate the joint contribution of H$^2$LoRA, training strategy, and DIYHealth-900K to the overall effectiveness of the model.

    *(Suggested by Reviewer `srqP`, `QoBN`, `r2KA`)*

We have provided point-by-point responses to all the comments, and we believe the clarifications and additional experiments have fully addressed the concerns raised. We hope the revisions and comprehensive responses further demonstrate the significance and rigor of our work. Once again, we would like to thank all reviewers for their constructive feedback and valuable suggestions, which have played a key role in refining our work.

---

### Meta-Review · Area_Chair_7Hqz · 2026-01-07

**Summary:**

reviewers gave scores of 4,4,6,6, with main concerns being

DIYHealth-900K is constructed entirely from existing public datasets, so no new data or tasks are introduced in this work.

The strong performance of DIYHealthGPT is largely attributable to training on in-distribution data, as the training and test sets are drawn from the same dataset. Comparisons against other models trained on other datasets are therefore unfair (ie those would be zero-shot).

The proposed evaluation tasks appear relatively easy, with even small models achieving strong results and the proposed method approaching near-perfect performance in many cases, so the benchmark might be saturated.

Most training labels are machine-generated, only 10% of labels are audited by humans, which could lead to the model learning the wrong things.

The paper does not provide real user feedback for assessing usability, and does not compare performance across subgroups such as gender, sex, race or ethnicity, age, skin tone, primary language, device type, and lighting conditions.

Novelty concerns in the proposed methodology, since previous works have already applied similar MoE-based architectures for medical foundation models.

**Reviewer Concerns:**

DIYHealth-900K is constructed entirely from existing public datasets, so no new data or tasks are introduced in this work.

--> authors gave some explanations on unifying and processing these datasets and adding new tasks. not particularly key contribution i feel.

The strong performance of DIYHealthGPT is largely attributable to training on in-distribution data, as the training and test sets are drawn from the same dataset. Comparisons against other models trained on other datasets are therefore unfair (ie those would be zero-shot).

--> authors added experiments by fine-tuning existing models on DIYHealth-900K for fairer comparisons

The proposed evaluation tasks appear relatively easy, with even small models achieving strong results and the proposed method approaching near-perfect performance in many cases, so the benchmark might be saturated.

--> not addressed

Most training labels are machine-generated, only 10% of labels are audited by humans, which could lead to the model learning the wrong things.

--> authors addressed this, where labels are derived from the original dataset and only formatted via LLMs

The paper does not provide real user feedback for assessing usability, and does not compare performance across subgroups such as gender, sex, race or ethnicity, age, skin tone, primary language, device type, and lighting conditions.

--> authors added these experiments

Novelty concerns in the proposed methodology, since previous works have already applied similar MoE-based architectures for medical foundation models.

--> authors acknowledge their approach is based on minor modifications to existing MoE-based architectures, particularly on the LoRA methods.

**Reviewer Scores:**

i think the reviewers who gave 4 are not likely to change their scores, since their core concern about DIYHealth-900K constructed entirely from existing public datasets, so no new data or tasks are introduced in this work, and novelty concerns in the proposed methodology, since previous works have already applied similar MoE-based architectures for medical foundation models, remain mostly unaddressed.

---

### Decision · Program_Chairs · 2026-01-26

Reject